# IN-CONTEXT LEARNING THROUGH THE BAYESIAN PRISM

**Madhur Panwar**[*♡]    **Kabir Ahuja**[*†♢]    **Navin Goyal**[♡]
♡ Microsoft Research India
{t-mpanwar, navingo}@microsoft.com
♢ University of Washington
kahuja@cs.washington.edu

## ABSTRACT

In-context learning (ICL) is one of the surprising and useful features of large language models and subject of intense research. Recently, stylized meta-learning-like ICL setups have been devised that train transformers on sequences of input-output pairs $(x, f(x))$. The function $f$ comes from a function class and generalization is checked by evaluating on sequences generated from unseen functions from the same class. One of the main discoveries in this line of research has been that for several function classes, such as linear regression, transformers successfully generalize to new functions in the class. However, the inductive biases of these models resulting in this behavior are not clearly understood. A model with unlimited training data and compute is a Bayesian predictor: it learns the pretraining distribution. In this paper we empirically examine how far this Bayesian perspective can help us understand ICL. To this end, we generalize the previous meta-ICL setup to hierarchical meta-ICL setup which involve unions of multiple task families. We instantiate this setup on a diverse range of linear and nonlinear function families and find that transformers can do ICL in this setting as well. Where Bayesian inference is tractable, we find evidence that high-capacity transformers mimic the Bayesian predictor. The Bayesian perspective provides insights into the inductive bias of ICL and how transformers perform a particular task when they are trained on multiple tasks. We also find that transformers can learn to generalize to new function classes that were not seen during pretraining. This involves deviation from the Bayesian predictor. We examine these deviations in more depth offering new insights and hypotheses[1].

## 1 INTRODUCTION

In-context learning (ICL) is one of the major ingredients behind the astounding performance of large language models (LLMs) (Brown et al., 2020). Unlike traditional supervised learning, ICL is the ability to learn new functions $f$ without weight updates from input-output examples $(x, f(x))$ provided as input at test time. For instance, given the prompt up -> down, low -> high, small ->, a pretrained LLM will likely produce output big: it apparently infers that the function in the two examples is the antonym of the input and applies it on the new input. This behavior often extends to more sophisticated and novel functions unlikely to have been seen during training and has been the subject of intense study, e.g., Min et al. (2022b); Webson & Pavlick (2022); Min et al. (2022a); Liu et al. (2023); Dong et al. (2023). More broadly than its applications in NLP, ICL can also be viewed as providing a method for meta-learning (Hospedales et al., 2022) where the model learns to learn a class of functions.

Theoretical understanding of ICL is an active area of research. Since the real-world datasets used for LLM training are difficult to model theoretically and are very large, ICL has also been studied in stylized setups, e.g., Xie et al. (2022); Chan et al. (2022b); Garg et al. (2022); Wang et al. (2023);

---

[*]Equal Contribution
[†]A part of this work was done while Kabir was a Research Fellow at Microsoft Research India.
[1]We release our code at https://github.com/mdrpanwar/icl-bayesian-prism

Hahn & Goyal (2023). These setups study different facets of ICL. In this paper, we focus on the meta-learning-like framework of Garg et al. (2022). Unlike in NLP where training is done on documents for the next-token prediction task, here the training and test data look similar in the sense that the training data consists of input of the form $((\boldsymbol{x}_1, f(\boldsymbol{x}_1)), \ldots, (\boldsymbol{x}_k, f(\boldsymbol{x}_k)), \boldsymbol{x}_{k+1})$ and output is $f(\boldsymbol{x}_{k+1})$, where $\boldsymbol{x}_i \in \mathbb{R}^d$ and are chosen i.i.d. from a distribution, and $f : \mathbb{R}^d \to \mathbb{R}$ is a function from a family of functions, for example, linear functions or shallow neural networks. We call this setup **MICL**. A striking discovery in Garg et al. (2022) was that for several function families, transformer-based language models during pretraining learn to implicitly implement well-known algorithms for learning those functions in context. For example, when shown 20 examples of the form $(\boldsymbol{x}, \boldsymbol{w}^T \boldsymbol{x})$, where $\boldsymbol{x}, \boldsymbol{w} \in \mathbb{R}^{20}$, the model correctly outputs $\boldsymbol{w}_{\text{test}}^T \boldsymbol{x}$ on test input $\boldsymbol{x}_{\text{test}}$. Apart from linear regression, they show that for sparse linear regression and shallow neural networks the trained model appears to implement well-known algorithms; and for decision trees, the trained model does better than baselines. Two follow-up works Akyürek et al. (2022) and von Oswald et al. (2022) largely focused on the case of linear regression. Among other things, they showed that transformers with one attention layer learn to implement one step of gradient descent on the linear regression objective with further characterization of the higher number of layers.

**Bayesian predictor.** An ideal language model (LM) with unlimited training data and compute would learn the pretraining distribution as that results in the smallest loss. Such an LM produces the output by simply sampling from the pretraining distribution conditioned on the input prompt. Such an ideal model is often called *Bayesian predictor*. Many works make the assumption that trained LMs are Bayesian predictors, e.g. Saunshi et al. (2021); Xie et al. (2022); Wang et al. (2023). Most relevant to the present paper, Akyürek et al. (2022) show that in the MICL setup for linear regression, in the underdetermined setting, namely when the number of examples is smaller than the dimension of the input, the model learns to output the least $L_2$-norm solution which is the Bayes-optimal prediction. In this paper we empirically examine how general this behavior is across choices of tasks.

Prior work has investigated related questions but we are not aware of any extensive empirical verification. E.g., Xie et al. (2022) study a synthetic setup where the pretraining distribution is given by a mixture of hidden Markov models and show that the prediction error of ICL approaches Bayes-optimality as the number of in-context examples approach infinity. In contrast, we test the Bayesian hypothesis for ICL over a wide class of function families and show evidence for equivalence with Bayesian predictor at all prompt lengths. Also closely related, Müller et al. (2022); Hollmann et al. (2023) train transformer models by sampling data from a prior distribution (Prior Fitted Networks), so it could approximate the posterior predictive distribution at inference time. While these works focus on training models to approximate posterior distributions for solving practical tasks (tabular data), our objective is to understand how in-context learning works in transformers and to what extent we can explain it as performing Bayesian Inference on the pre-training distribution.

**Simplicity bias.** Simplicity bias, the tendency of machine learning algorithms to prefer simpler hypotheses among those consistent with the data, has been suggested as the basis of the success of neural networks. There are many notions of simplicity (Mingard et al., 2023; Goldblum et al., 2023). Does in-context learning also enjoy a simplicity bias like pretraining?

**Our contributions.** In brief, our contributions are

**1.** *A setup for studying ICL for multiple function families*: First, we extend the MICL setup from Garg et al. (2022) to include multiple families of functions. For example, the prompts could be generated from a mixture of tasks where the function $f$ is chosen to be either a linear function or a decision tree with equal probability. We call this extended setup HMICL. We experimentally study HMICL and find that high-capacity transformer models can learn in context when given such task mixtures. (We use the term "high-capacity" informally; more precisely, it means that for the task at hand there is a sufficiently large model with the desired property.)

**2.** *High-capacity transformers perform Bayesian inference during ICL:* To understand how this ability arises we investigate in depth whether high-capacity transformers simulate the Bayesian predictor. This motivates us to choose a diverse set of linear and nonlinear function families as well as prior distributions in HMICL and MICL setups. Function families we consider were chosen because either they permit efficient and explicit Bayesian inference or have strong baselines. We provide direct and indirect evidence that indeed high-capacity transformers often mimic the Bayesian predictor.

The ability to solve task mixtures arises naturally as a consequence of Bayesian prediction. In concurrent work, Bai et al. (2023) also study the multiple function classes setup for ICL like us, showing that transformers can in-context learn individual function classes from the mixture. However, there are three main differences between our works. Bai et al. (2023) interpret the multi-task ICL case as algorithm selection, where transformer based on the in-context examples selects the appropriate algorithm for the prompt and then executes it. We provide an alternate explanation that there is no need for algorithm selection and it follows naturally from the Bayesian perspective. Further, while they show that their constructions approach bayes-optimality at large prompt lengths, we show through experiments that this actually holds true for all prompt lengths. Finally, we test this phenomenon on a much larger set of mixtures. For Gaussian Mixture Models, we also compare the performance with the exact Bayesian predictor and such a comparison is missing in Bai et al. (2023).

**3.** *Link between ICL inductive bias with the pretraining data distribution:* We also investigate the inductive bias in a simple setting for learning functions given by Fourier series. If ICL is biased towards fitting functions of lower maximum frequency, this would suggest that it has a bias for lower frequencies like the spectral bias for pretraining. We find that the model mimics the Bayesian predictor; the ICL inductive bias of the model is determined by the pretraining data distribution: if during pretraining all frequencies are equally represented, then during ICL the LM shows no preference for any frequency. On the other hand, if lower frequencies are predominantly present in the pretraining data distribution then during ICL the LM prefers lower frequencies. Chan et al. (2022a;b) studies the inductive biases of transformers for ICL and the effect of pretraining data distribution on them. However, the problem setting in these papers is very different from ours and they do not consider simplicity bias.

**4.** *Generalization to new tasks not seen during training in HMICL:* In HMICL setup, we study generalization to new tasks that were not seen during pretraining. We find that when there's sufficient diversity of tasks in pretraining, transformers generalize to new tasks. Similar study was made in the concurrent work of Raventós et al. (2023) for the noisy linear regression problem within MICL.

**5.** *Study of deviations from Bayesian prediction:* Finally, we study deviations from the Bayesian predictor. These can arise either in multitask generalization problems or when the transformer is not of sufficiently high capacity for the problem at hand. For the former, we study the pretraining inductive bias and find surprising behavior of transformers where they prefer to generalize to a large set of tasks early in the pretraining which then they forget. For the latter, drawing on recent work connecting Bayesian inference with gradient-based optimization, we hypothesize that in fact transformers may be attempting to do Bayesian inference.

## 2 BACKGROUND

We first discuss the in-context learning setup for learning function classes as introduced in Garg et al. (2022), which we call **Meta-ICL** or **MICL**. Let $\mathcal{D}_\mathcal{X}$ be a probability distribution on $\mathbb{R}^d$. Let $\mathcal{F}$ be a family of functions $f : \mathbb{R}^d \to \mathbb{R}$ and let $\mathcal{D}_\mathcal{F}$ be a distribution on $\mathcal{F}$. For simplicity, we often use $f \sim \mathcal{F}$ to mean $f \sim \mathcal{D}_\mathcal{F}$. We overload the term function class to encompass both function definition as well as priors on its parameters. Hence, linear regression with a standard gaussian prior and a sparse prior will be considered different function classes based on our notation.

To construct a prompt $P = \left( \boldsymbol{x}_1, f(\boldsymbol{x}_i), \cdots, \boldsymbol{x}_p, f(\boldsymbol{x}_p), \boldsymbol{x}_{p+1} \right)$ of length $p$, we sample $f \sim \mathcal{F}$ and inputs $\boldsymbol{x}_i \sim \mathcal{D}_\mathcal{X}$ i.i.d. for $i \in \{1, \cdots p\}$. A transformer-based language model $M_\theta$ is trained to predict $f(\boldsymbol{x}_{p+1})$ given $P$, using the objective: $\min_\theta \mathbb{E}_{f, \boldsymbol{x}_{1:p}} \left[ \frac{1}{p+1} \sum_{i=0}^{p} \ell \left( M_\theta(P^i), f(\boldsymbol{x}_{i+1}) \right) \right]$, where $P^i$ denotes the sub-prompt containing the first $i$ input-output examples as well as the $(i+1)$-th input, i.e. $\left( \boldsymbol{x}_1, f(\boldsymbol{x}_1), \cdots, \boldsymbol{x}_i, f(\boldsymbol{x}_i), \boldsymbol{x}_{i+1} \right)$ and $\boldsymbol{x}_{1:p} = (\boldsymbol{x}_1, \ldots, \boldsymbol{x}_p)$. While other choices of the loss function $\ell(\cdot, \cdot)$ are possible, since we study regression problems we use the squared-error loss (i.e., $\ell(y, y') = (y - y')^2$) in accordance with Garg et al. (2022).

At test time, we present the model with prompts $P_{\text{test}}$ that were unseen during training with high probability and compute the error when provided $k$ in-context examples: $\texttt{loss@}k = \mathbb{E}_{f, P_{\text{test}}} \left[ \ell \left( M_\theta(P^k), f(\boldsymbol{x}_{k+1}) \right) \right]$, for $k \in \{1, \cdots, p\}$.

**PME.** Our work uses basic Bayesian probability as described, e.g., in Murphy (2022). We mentioned earlier that an ideal model would learn the pretraining distribution. This happens when using the cross-entropy loss. Since we use the square loss in the objective definition, the predictions of this

ideal model can be computed using the posterior mean estimator (PME) from Bayesian statistics. For each prompt length $i$, and any prompt $Q = \left(\boldsymbol{x}_1, g(\boldsymbol{x}_1), \cdots, \boldsymbol{x}_p, g(\boldsymbol{x}_p), \boldsymbol{x}_{p+1}\right)$ where $g$ is a function in the support of $\mathcal{D}_{\mathcal{F}}$, we can compute the PME by taking the corresponding summand in objective definition above, which will be given by $M_\theta(Q^i) = \mathbb{E}_f\left[f(\boldsymbol{x}_{i+1}) \,|\, P^i = Q^i\right]$ for all $i \leq p$. This is the optimal solution for prompt $Q$, which we refer to as PME. Please refer to §A.1 for technical details behind this computation.

## 2.1 HIERARCHICAL META-ICL

We generalize the MICL setup, where instead of training transformers from functions sampled from a single function class, we sample them from a mixture of function classes. Formally, we define a mixture of function classes using a set of $m$ function classes $\boldsymbol{\mathcal{F}} = \{\mathcal{F}_1, \cdots, \mathcal{F}_m\}$ and sampling probabilities $\boldsymbol{\alpha} = [\alpha_1, \cdots \alpha_m]^T$ with $\sum_{i=1}^m \alpha_i = 1$. We use $\boldsymbol{\alpha}$ to sample a function class for constructing the training prompt $P$. We assume the input distribution $\mathcal{D}_{\mathcal{X}}$ to be same for each class $\mathcal{F}_{T_i}$. More concretely, the sampling process for $P$ is defined as: i) $\mathcal{F}_i \sim \boldsymbol{\mathcal{F}}$ s.t. $\mathbb{P}(\mathcal{F} = \mathcal{F}_i) = \alpha_i$; ii) $f \sim \mathcal{F}_i$; iii) $\boldsymbol{x}_j \sim \mathcal{D}_{\mathcal{X}}, \forall j \in \{1, \cdots, p\}$; and finally, iv) $P = \left(\boldsymbol{x}_1, f(\boldsymbol{x}_1), \cdots \boldsymbol{x}_p, f(\boldsymbol{x}_p), \boldsymbol{x}_{p+1}\right)$.

We call this setup **Hierarchical Meta-ICL** or **HMICL**, as there is an additional first step for sampling the function class in the sampling procedure. Note that the MICL setup can be viewed as a special case of HMICL where $m = 1$. The HMICL setting presents a more advanced scenario to validate whether the Bayesian inference can be used to explain the behavior of in-context learning in transformers. Further, our HMICL setup is also arguably closer to the in-context learning in practical LLMs which can realize different classes of tasks (sentiment analysis, QA, summarization etc.) depending upon the inputs provided. (For additional discussion on HMICL and MICL, refer to Appendix §C.1.) The PME for the hierarchical case is given by:

$$M_{\theta, \boldsymbol{\mathcal{F}}}(P) = \beta_1 M_{\theta, \mathcal{F}_1}(P) + \ldots + \beta_m M_{\theta, \mathcal{F}_m}(P), \tag{1}$$

where $\beta_i = \alpha_i p_i(P) / p_{\boldsymbol{\mathcal{F}}}(P)$ for $i \leq m$. Probability density $p_i(\cdot)$ is induced by the function class $\mathcal{F}_i$ on the prompts in a natural way, and $p_{\boldsymbol{\mathcal{F}}}(P) = \alpha_i p_i(P) + \cdots + \alpha_m p_m(P)$. Please refer to §A.1 in the Appendix for the derivation. The models are trained with the squared error loss mentioned above and at test time we evaluate $\texttt{loss@}k$ for each task individually.

## 2.2 MODEL AND TRAINING DETAILS

We use the decoder-only transformer architecture Vaswani et al. (2017) as used in the GPT models Radford et al. (2019). Unless specified otherwise, we use 12 layers, 8 heads, and a hidden size ($d_h$) of 256 in the architecture for all of our experiments. We use a batch size of 64 and train the model for 500k steps. For encoding the inputs $\boldsymbol{x}_i$'s and $f(\boldsymbol{x}_i)$'s, we use the same scheme as Garg et al. (2022) which uses a linear map $\boldsymbol{E} \in \mathbb{R}^{d_h \times d}$ to embed the inputs $\boldsymbol{x}_i$'s as $\boldsymbol{E}\boldsymbol{x}_i$ and $f(\boldsymbol{x}_i)$'s as $\boldsymbol{E}f_{\text{pad}}(\boldsymbol{x}_i)$, where $f_{\text{pad}}(\boldsymbol{x}_i) = [f(\boldsymbol{x}_i), \boldsymbol{0}_{d-1}]^T \in \mathbb{R}^d$. In all of our experiments except the ones concerning the Fourier series, we choose $\mathcal{D}_{\mathcal{X}}$ as the standard normal distribution i.e. $\mathcal{N}(0, 1)$, unless specified otherwise. To accelerate training, we also use curriculum learning like Garg et al. (2022) for all our experiments where we start with simpler function distributions (lower values of $d$ and $p$) at the beginning of training and increase the complexity as we train the model.

## 3 TRANSFORMERS CAN IN-CONTEXT LEARN TASK MIXTURES

In this section, we provide evidence that transformers' ability to solve mixture of tasks arises naturally from the Bayesian perspective. We start with a Gaussian Mixture Models (GMMs) example where the exact Bayesian solution is tractable and later discuss results for more complex mixtures.

## 3.1 GAUSSIAN MIXTURE MODELS (GMMS)

We define a mixture of dense-linear regression classes $\boldsymbol{\mathcal{F}}_{\text{GMM}} = \{\mathcal{F}_{\text{DR}_1}, \cdots, \mathcal{F}_{\text{DR}_m}\}$, where $\mathcal{F}_{\text{DR}_i} = \left\{ f : \boldsymbol{x} \mapsto \boldsymbol{w}_i^T \boldsymbol{x} \,|\, \boldsymbol{w}_i \in \mathbb{R}^d \right\}$ and $\boldsymbol{w}_i \sim \mathcal{N}_d(\boldsymbol{\mu}_i, \boldsymbol{\Sigma}_i)$. In other words, each function class in the mixture corresponds to dense regression with Gaussian prior on weights (but different means or covariance matrices). We report experiments with $m = 2$ here, and the mean vectors are given by $\boldsymbol{\mu}_1 = (3, 0, .., 0)$ and $\boldsymbol{\mu}_2 = (-3, 0, ..., 0)$ for the two classes. The covariance matrices are equal ($\boldsymbol{\Sigma}_1 = \boldsymbol{\Sigma}_2 = \boldsymbol{\Sigma}^*$), where $\boldsymbol{\Sigma}^*$ is the identity matrix $\boldsymbol{I}_d$ with the top-left entry replaced by 0. Note that we can equivalently view this setup by considering the prior on weights as a mixture of Gaussians i.e. $p_M(\boldsymbol{w}) = \alpha_1 \mathcal{N}_d(\boldsymbol{\mu}_1, \boldsymbol{\Sigma}_1) + \alpha_2 \mathcal{N}_d(\boldsymbol{\mu}_2, \boldsymbol{\Sigma}_2)$. For brevity, we call the two function classes $T_1$ and $T_2$. We train the transformer on a uniform mixture i.e. $\alpha_1, \alpha_2$ are $\frac{1}{2}$. We use $d = 10$ and the prompt length $p \in \{10, 20\}$.

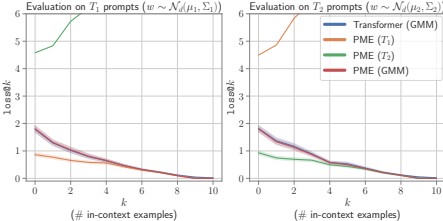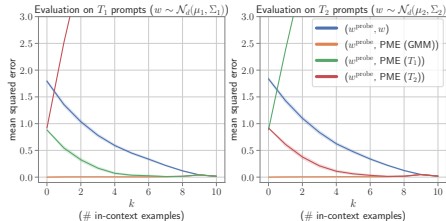

Figure 1: Transformers simulate PME when trained on dense regression task-mixture with weights having a mixture of Gaussian prior (GMM). *(left)*: Comparing the performance of the Transformer with PME of individual Gaussian components (PME ($T_1$) and PME ($T_2$)) and of the mixture PME (GMM). *(right)*: MSE between the probed weights of the Transformer and PMEs.

**Recovering implied weights.** To provide a stronger evidence for the Bayesian hypothesis, apart from the loss curves, we also extract the weights implied by transformers for solving the regression task in-context. Following Akyürek et al. (2022), we do this by generating model's predictions $\{y_i'\}$ on the test inputs $\{x_i'\}_{i=1}^{2d} \sim \mathcal{D}_\mathcal{X}$ and then solving the system of equations to recover $w^{\mathrm{probe}}$. We then compare the implied weights $w^{\mathrm{probe}}$ with the ground truth weights $w$ as well as the weights extracted from different baselines by computing the their MSE.

**Results.** In Figure 1 (*left*), we note that Transformer's errors almost exactly align with those of the PME of the mixture, PME (GMM), when prompts come from either $T_1$ or $T_2$. (For details on computation of PME, please refer to §C.2 in Appendix). For each plot, let $T_{\mathrm{prompt}}$ and $T_{\mathrm{other}}$ denote the component from which prompts are provided and the other component respectively. When $d = 10$ examples from $T_{\mathrm{prompt}}$ have been provided, the Transformer, PME ($T_{\mathrm{prompt}}$), and PME (GMM) all converge to the same minimum error of 0. This shows that Transformer is simulating PME (GMM), which converges to PME ($T_{\mathrm{prompt}}$) at $k = d$. PME ($T_{\mathrm{other}}$)'s errors keep increasing as more examples are provided. These observations are in line with Eq. 3: As more examples from the prompt are observed, the weights of individual PMEs used to compute the PME (GMM) (i.e., the $\beta$'s) evolve such that the contribution of $T_{\mathrm{prompt}}$ increases in the mixture with $k$ (Fig. 22 in the Appendix). In Figure 1 (*right*), MSE between weights from different predictors are plotted. Transformer's implied weights are almost exactly identical to PME (GMM) for all $k$. Please refer to §C.2 for additional details and results.

**More complex mixtures.** We test the generality of the phenomenon discussed above for more complex mixtures, involving mixtures of two or three different linear inverse problems (e.g. dense regression, sparse regression, sign vector regression) as well as some mixtures involving non-linear function classes like neural networks and decision trees. In all of these cases we observe that transformers trained on the mixtures are able to generalize on the new functions from the mixture of function classes and match the the performance of single-function class transformer models depending upon the distribution of input prompt. Please refer to §C.3 for details.

**Implications.** Our GMM experiments challenge the existing explanations for the multi-task case, e.g. the models first recognizes the task and then solves it. When viewed through the Bayesian lens, transformers do not need to recognize the task separately and recognition and solution are intertwined as we show in Equation 1.

## 4   SIMPLICITY BIAS IN ICL?

In this section, we explore if transformers exhibit simplicity bias in ICL. In other words, when given a prompt containing input output examples, do they prefer to fit simpler functions among those that fit the prompt? To study this behavior we consider the Fourier Series function class, where the output is a linear function of sine and cosine functions of different frequencies. By training transformers on this class, during ICL we can study if transformers prefer fitting lower-frequency functions to the prompt over higher frequencies, which can help us study the presence of a simplicity bias.

More formally, we can define Fourier series by the following expansion: $f(x) = a_0 + \sum_{n=1}^{N} a_n \cos(n\pi x/L) + \sum_{n=1}^{N} b_n \sin(n\pi x/L)$ where, $x \in [-L, L]$, and $a_0$, $a_n$'s and $b_n$'s are known as Fourier coefficients and $\cos n\pi/L$ and $\sin n\pi/L$ define the frequency $n$ components.

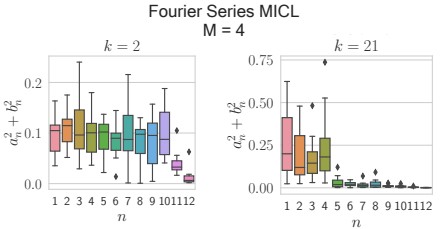
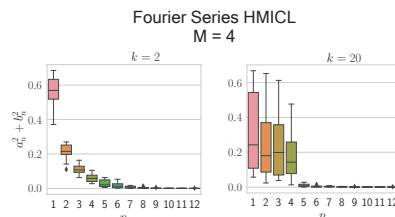

Figure 2: Measuring the frequencies of the simulated function during ICL by transformer.

**MICL Setup.** In the MICL setup we train transformer on a single function class defined as $\mathcal{F}_{\Phi_N}^{\text{fourier}} = \{f(\cdot; \Phi_N) | f(\boldsymbol{x}; \Phi) = \boldsymbol{w}^T \Phi_N(\boldsymbol{x}), \boldsymbol{w} \in \mathbb{R}^d\}$ with standard gaussian prior on weights $\boldsymbol{w}$. Note that here $\Phi_N$ as the Fourier feature map i.e. $\Phi_N(x) = [1, \cos(\pi x/L), \cdots, \cos(N\pi x/L), \sin(\pi x/L), \cdots, \sin(N\pi x/L)]^T$. For training transformers to in-context-learn $\mathcal{F}_{\Phi_N}^{\text{fourier}}$, we fix a value of $N$ and sample functions $f \in \mathcal{F}_{\Phi_N}^{\text{fourier}}$. We consider the inputs to be scalars, i.e. $x_i \in [-L, L]$ and we sample them i.i.d. from the uniform distribution on the domain: $x_i \sim \mathcal{U}(-L, L)$. In all of our experiments, we consider $N = 10$ and $L = 5$. At test time we evaluate on $\mathcal{F}_{\Phi_M}^{\text{fourier}}$ for $M \in [1, 10]$, i.e. during evaluation we also prompt the model with functions with different maximum frequency as seen during training.

**HMICL Setup.** We also consider a mixture of Fourier series function classes with different maximum frequencies, i.e. $\mathcal{F}_{\Phi_{1:N}}^{\text{fourier}} = \{\mathcal{F}_{\Phi_1}^{\text{fourier}}, \cdots, \mathcal{F}_{\Phi_N}^{\text{fourier}}\}$. We consider $N = 10$ in our experiments and train the models using a uniform mixture with normalization. During evaluation, we test individually on each $\mathcal{F}_{\Phi_M}^{\text{fourier}}$, where $M \in [1, N]$.

**Measuring inductive biases.** To study simplicity bias during ICL, we propose a method to recover implied frequency from the transformer model. We start by sampling in-context examples $(x_1, f(x_1), \cdots x_k, f(x_k))$, and given the context obtain the model's predictions on a set of $m$ test inputs $\{x_i'\}_{i=1}^m$, i.e. $y_i' = M_\theta \left( (x_1, f(x_1), \cdots x_k, f(x_k), x_i') \right)$. We can then perform Discrete Fourier Transform (DFT) on $\{y_1', \cdots, y_m'\}$ to obtain the Fourier coefficients of the function output by $M$, which we can analyze to understand the dominant frequencies.

**Results.** In both MICL and HMICL setups discussed above we observe that transformers are able to in-context learn these function classes and match the performance of the Bayesian predictor or strong baselines. Since, in this section we are primarily interested in studying the simplicity bias, here we only report the plots for frequencies recovered from transformers at different prompt lengths in Figure 2 (more details in Figures 12 and 32 of Appendix). As can be seen in Figure 2 (*left*), in the single function class case, transformers exhibit no bias towards any particular frequency. For small prompt lengths ($k = 2$), all $N$ frequencies receive similar absolute value of coefficients as implied by the transformer. As more examples are provided ($k = 21$), transformer is able to recognize the gold maximum frequency ($M = 4$) from the in-context examples, and hence coefficients are near zero for $n > M$, but as such there is no bias towards any particular frequencies. However, when we consider the mixture case in Figure 2 (*right*), the situation is different. We see a clear bias for lower frequencies at small prompt lengths; however, when given sufficiently many examples they are able to recover the gold frequencies. This simplicity bias can be traced to the training dataset for the mixture since lower frequencies are present in most of the functions of the mixture while higher frequencies will be more rare: Frequency 1 will be present in all the function classes whereas frequency $N$ will be present only in $\mathcal{F}_{\Phi_N}^{\text{fourier}}$. We perform additional experiments biasing pre-training distribution to high frequencies and observe complexity bias during ICL (Appendix §C.4.1).

**Implications.** These results suggest that the simplicity bias (or lack thereof) during ICL can be attributed to the pre-training distribution which follows naturally from the Bayesian perspective i.e. the biases in the prior are reflected in the posterior. Transformers do not add any extra inductive bias of their own as they emulate the Bayesian predictor.

## 5 MULTI-TASK GENERALIZATION

In this section we test the HMICL problems on out-of-distribution (OOD) function classes to check generalization. We work with the degree-2 monomials regression problem, $\mathcal{F}_S^{\text{mon}(2)}$ which is given by a function class where the basis is formed by a feature set $\mathcal{S}$, a subset of degree-2 monomials

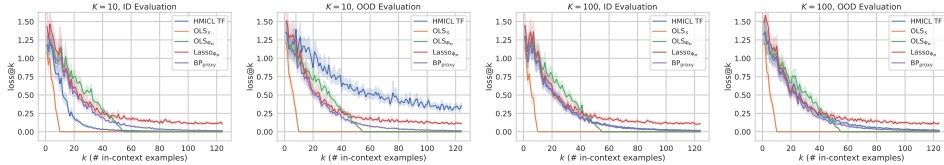

Figure 3: **Multi-task generalization results for Monomials problem**. ID and OOD evaluation for $K = 10, 100$ is presented. As task diversity ($K$) increases, the model starts behaving like $\text{Lasso}_{\Phi_M}$ and $\text{BP}_{\text{proxy}}$ and its ID and OOD losses become almost identical, i.e. it generalizes to OOD.

$\mathcal{S} \subset M = \{(i, j)| 1 \le i, j \le d\}$. We can then define the feature map $\Phi_{\mathcal{S}}(\boldsymbol{x}) = (x_i x_j)_{(i,j) \in \mathcal{S}}$ and $f(\boldsymbol{x}) = \boldsymbol{w}^T \Phi_{\mathcal{S}}(\boldsymbol{x})$ is a function of this class, where $\boldsymbol{w} \sim \mathcal{N}_{|\mathcal{S}|}(\boldsymbol{0}, \boldsymbol{I})$. We compare the performance of TFs on this class with OLS performed on the feature set $\mathcal{S}$ ($\text{OLS}_{\mathcal{S}}$) which is the Bayesian predictor in this case. We find that the error curves of the TF trained and evaluated on this class follow $\text{OLS}_{\mathcal{S}}$ baseline closely for all prompt lengths, on both in- and out-of-distribution evaluation. (Refer to §B.2.3 for a detailed description of setup and results.)

**Extending to HMICL setting.** For HMICL, we use multiple feature sets $\mathcal{S}_k$'s to define the mixture. Each $\mathcal{S}_k$ defines a function class $\mathcal{F}_{\mathcal{S}_k}^{\text{mon}(2)}$. The pretraining distribution is induced by the uniform distribution $\mathcal{U}(\boldsymbol{\mathcal{F}})$ over a collection of such function classes, $\boldsymbol{\mathcal{F}} = \{\mathcal{F}_{\mathcal{S}_1}^{\text{mon}(2)}, \cdots, \mathcal{F}_{\mathcal{S}_K}^{\text{mon}(2)}\}$, where $\mathcal{S}_k \subset M$. $K$ feature sets $\mathcal{S}_k$'s, each of size $D$, are chosen at the start of the training and remain fixed. $K$ is the task diversity of the pretraining distribution. To sample a training function for the TF, we first sample a function class $\mathcal{F}_{\mathcal{S}_k}^{\text{mon}(2)}$ with replacement from $\mathcal{U}(\boldsymbol{\mathcal{F}})$ and then sample a function from the chosen class; $f(\boldsymbol{x}) = \boldsymbol{w}^T \mathcal{S}_k(\boldsymbol{x})$, where $\boldsymbol{w} \sim \mathcal{N}_D(\boldsymbol{0}, \boldsymbol{I})$. Our aim is to check if TF trained on $\mathcal{U}(\boldsymbol{\mathcal{F}})$ can generalize to the full distribution of all function classes (for feature sets of size $D$) by evaluating its performance on function classes corresponding to feature sets $\mathcal{S}' \notin \{\mathcal{S}_1, \cdots, \mathcal{S}_K\}$.

**Experimental Setup.** We choose $D = d = 10, p = 124$. There is no curriculum learning – $d$ and $p$ remain fixed throughout training. Note that the total number of degree-2 monomials $= M_{tot} = \binom{d}{2} + \binom{d}{1} = 45 + 10 = 55$; and the total number of distinct feature sets $\mathcal{S}_k$'s (and hence function classes) $= \binom{M_{tot}}{D} = \binom{55}{10} \approx 3^{10}$. We train various models for different task diversities; $K \in \{10, 20, 40, 100, 500, 1000, 5000\}$. We evaluate on a batch of $B = 1280$ functions in two settings: **(a) In-Distribution (ID)** – test functions formed using randomly chosen function classes from the pretraining distribution; **(b) Out-of-Distribution (OOD)** – Test functions formed using randomly chosen function classes not in the pretraining distribution.

**Baselines.** We compare the performance of multi-task transformer models with the following baselines: **1. $\text{OLS}_{\mathcal{S}}$:** Here, we perform OLS on the basis formed by the gold feature set $\mathcal{S}$, which was used to define the function in the prompt that we wish to evaluate. This will correspond to an upper bound on the performance as at test time the transformer model has no information about the correct basis. **2. $\text{OLS}_{\Phi_M}$:** Here, OLS is performed on the basis formed by all degree-2 monomials $\Phi_M(\boldsymbol{x})$ for an input $\boldsymbol{x}$. Hence, this baseline can generalize to any of the feature set $\mathcal{S}$. However, since all degree-2 monomial features are considered by this baseline, it would require a higher number of input-output examples (equal to $M_{tot}$) for the problem to be fully determined. **3. $\text{Lasso}_{\Phi_M}$:** Similar to $\text{OLS}_{\Phi_M}$, we operate on all degree-2 monomial features, but instead of OLS we perform Lasso with $\alpha = 0.1$. It should also generalize to arbitrary feature sets $\mathcal{S}$, however, Lasso can take advantage of the fact that $|\mathcal{S}| = D \ll M_{tot}$; hence should be more efficient than $\text{OLS}_{\Phi_M}$.

**Results.** As a proxy for the Bayesian predictor ($\text{BP}_{\text{proxy}}$), we use the transformer trained on the full distribution of function families, since the computation of the exact predictor is expensive. From the plots in Figure 3, we observe that while for small values of $K$, the OOD generalization is poor but as we move to higher values of $K$, the models start to approach the performance of $\text{OLS}_{\Phi_M}$ and eventually $\text{Lasso}_{\Phi_M}$ on unseen $\mathcal{S}$'s. Further, they also start behaving like $\text{BP}_{\text{proxy}}$. However, this improvement in OOD performance comes at the cost of ID performance as task diversity ($K$) increases. Eventually, at larger $K$, both ID and OOD performances are identical. These observation are particularly interesting since the models learn to generalize to function classes out of the pre-training distribution and hence deviate from the Bayesian behavior which would lack such generalization

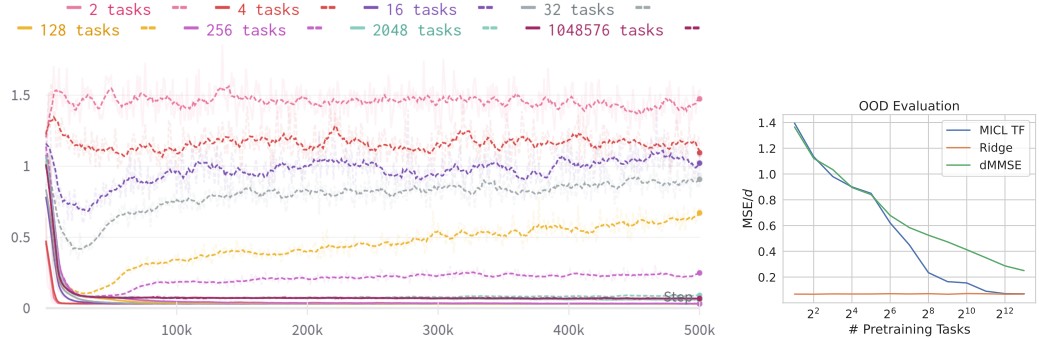

Figure 4: **Left:** Evolution of ID (solid lines) and OOD (dashed lines) losses during pretraining for representative task diversities. Task diversities $\{2^7 \cdots 2^{11}\}$ represent the **Gaussian forgetting region**. The moving average (over 10 training steps) of the losses is plotted for smoothing. **Right:** OOD loss given the full prompt length of 15 for the final checkpoint of models trained on various task diversities. Task diversities $\{2^7 \cdots 2^{11}\}$ represent the **transition region**.

and fit the pre-training distribution instead. We observe similar results for another family of function classes coming from Fourier series (details of these in Appendix §D.2).

In a concurrent work Raventós et al. (2023) also present a multi-task setting within MICL where a set of weight vectors define the pretraining distribution for the Noisy Linear Regression problem. Since we work with HMICL, our setting is more general; moreover, generalization to new function classes in our setting happens in a similar way as generalization to new tasks in Raventós et al. (2023). They emphasized deviation from the Bayesian predictor. What leads to these deviations? To understand this, in the next section we study pretraining inductive bias of transformers.

## 6 DEVIATIONS FROM BAYESIAN INFERENCE?

In the previous section we observed deviations from the Bayesian predictor in multitask generalization. To investigate this we study the pretraining dynamics of transformers in the first subsection. Another set of apparent deviations from Bayesian prediction is observed in literature when the problem is too hard or the transformer has limited capacity. We discuss these in the second subsection.

### 6.1 ICL TRANSFORMER FIRST GENERALIZES THEN MEMORIZES DURING PRETRAINING

We observe a very interesting phenomenon (which we term "forgetting") from multi-task experiments: *For certain task diversities, during pretraining, HMICL Transformer first generalizes (fits the full distribution) and later forgets it and memorizes (fits the pretraining distribution).*

The 'forgetting' phenomenon is general and occurs in our HMICL experiments in §5. However, here we focus on the the Noisy Linear Regression problem from Raventós et al. (2023) since forgetting is the cleanest in this setting. We briefly mention the problem setup and display the evidence for forgetting during pretraining, followed by its relation to the agreement of HMICL Transformer with the Bayesian predictors on the pretraining and full distributions.

**Problem Setup.** We follow the Noisy Linear Regression (NLR) setup from Raventós et al. (2023): $d = 8, p = 15$. (For details, see §D.3.) The pretraining distribution (PT$_{\text{dist.}}$) is induced by the uniform distribution on a fixed set of tasks (weight vectors). Several models are trained, one per task diversity $K \in \{2^1, 2^2, \cdots 2^{20}\}$. The full distribution of weight vectors is standard normal. (Hence we use the term "Gaussian distribution" to refer to the full distribution (FG$_{\text{dist.}}$).) To form a function $f$ for training, we randomly choose a weight vector $w$ from the pretraining distribution and define $f(x) = w^T x + \epsilon$, where $\epsilon \sim \mathcal{N}_d(0, \sigma^2 = 0.25)$.

**Evidence of forgetting and agreement with Bayesian predictors.** As we did in §5, we evaluate TF on tasks from both in- and out-of-pretraining distribution, where the tasks used to construct the test function come from pretraining distribution or the standard Gaussian distribution respectively; corresponding losses are called ID (Pretrain test) loss and OOD (Gaussian) loss. We also plot the Bayesian predictors for both pretraining (dMMSE) and full (Gaussian) distribution (Ridge regres-

sion) as defined in Raventós et al. (2023). In Figure 4 (left) we plot the evolution during pretraining of ID and OOD losses for representative task diversities (more details in §D.3) ID loss $\approx 0$ for all task diversities. We group them into the following 4 categories based on OOD loss and describe the most interesting one in detail (full classification in §D.3): **(1)** $2^1$ to $2^3$: *no generalization; no forgetting*; **(2)** $2^4$ to $2^6$: *some generalization; no forgetting*; **(3)** $2^7$ to $2^{11}$: *full generalization and forgetting* – OOD loss improves, reaches a minima $t_{min}$, at which it is same as ID loss, then it worsens. At $t_{min}$, OOD loss agrees with Ridge, then gradually deviates from it and at $t_{end}$ (end of pretraining), it is in between dMMSE and Ridge. We refer to this group of task diversities as the "Gaussian forgetting region" since the model generalizes to the full (Gaussian) distribution over tasks at $t_{min}$ but forgets it by $t_{end}$; **(4)** $2^{12}$ to $2^{20}$: *full generalization; no forgetting*.

The agreement of TF in OOD evaluation with Ridge or dMMSE as mentioned above is shown in §D.3. Figure 4 (right) plots the OOD loss given the full prompt length of 15 for the final checkpoint of models trained for various task diversities. As can be seen, smaller task diversities (up to $2^6$) agree with dMMSE (Bayesian predictor on $PT_{dist.}$), and larger task diversities (from $2^{12}$ onwards) agree with Ridge regression (Bayesian predictor on $FG_{dist.}$). (This observation was originally made by Raventós et al. (2023) and we present it for completeness.) Intermediate task diversities ($2^7$ to $2^{11}$) agree with neither of the two and we term them collectively as the **transition region**. We note that both **the Gaussian forgetting region and the transition region consist of the same set of task diversities** viz. $\{2^7, \cdots 2^{11}\}$. The phenomenon of forgetting provides an interesting contrast to grokking literature (e.g. Nanda et al. (2023)) and can possibly be explained via the perspective of simplicity bias. The extent of forgetting is directly proportional to the input dimension ($d$) and is robust to changes in hyperparameters (details, in section §D.3).

## 6.2 GRADIENT DESCENT AS A TRACTABLE APPROXIMATION OF BAYESIAN INFERENCE

Some recent results in the literature within MICL suggest that transformers compute their answer by gradient descent on in-context examples. Could this be related to Bayesian inference? We provide preliminary evidence for this in Appendix §E.

## 7 SUMMARY OF FURTHER RESULTS

In this section we summarize further results from the Appendix that verify the generality of Bayesian hypothesis. We test the hypothesis on a variety of linear and non-linear inverse problems in both MICL and HMICL setups and find that transformers are able to in-context learn and generalize to unseen functions from these function classes. In the cases where PME computation is tractable, we compare transformers with the exact Bayesian predictor (PME) and establish the agreement between the two. Where PME is intractable, we compare transformers with numerical solutions obtained using a Markov Chain Monte Carlo (MCMC) sampling algorithm Homan & Gelman (2014) (Figure 10 in Appendix). When even sampling based solutions do not converge, we compare with strong baselines that are known to be near optimal from prior work. For linear problems, we test on Dense, Sparse, Sign Vector, Low Rank and Skewed Covariance Regression. For these problems, we show that not only do transformers' errors agree with the Bayesian predictor (or the strong baselines), but also the weights of the function implied by the transformer. For the non-linear case, we explore regression problems for Fourier Series, Degree-2 Monomials, Random Fourier Features, and Haar Wavelets. For Bayesian inference the order of demonstrations does not matter for the class of problems used in our setup. Figure 11 experimentally verifies it for Dense Regression, where transformer's performance is independent of the permutation of in-context examples across different prompt lengths. Further, we note that in the HMICL setup, generalization to functions from the mixture might depend on different factors such as normalizing the outputs from each function class. We provide complete details for each of these function families and corresponding results in Appendix §B and §C.

## 8 CONCLUSION

In this paper we provided empirical evidence that the Bayesian perspective could serve as a unifying explanation for ICL. In particular, it can explain how the inductive bias of ICL comes from the pretraining distribution and how transformers solve mixtures of tasks. We also identified how transformers generalize to new tasks and this involves apparent deviation from Bayesian inference. There are many interesting directions for future work which we discuss in Appendix §F.

ACKNOWLEDGMENTS

K.A. was supported in part by the National Science Foundation under Grant No. IIS2125201. We would like to thank all the anonymous reviewers for their constructive feedback.

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

CONTENTS

## A  Technical Details

### A.1  PME Theoretical Details

We mentioned earlier that an ideal LM would learn the pretraining distribution. This happens when using the cross-entropy loss. Since we use the square loss in the ICL training objective, the predictions of the model can be computed using the posterior mean estimator (PME) from Bayesian statistics. For each prompt length $i$ we can compute PME by taking the corresponding summand in the ICL training objective

$$
\begin{aligned}
\min_\theta \mathbb{E}_{f,\boldsymbol{x}_{1:i}} \, \ell\left(M_\theta(P^i), f(\boldsymbol{x}_{i+1})\right) &= \min_\theta \mathbb{E}_{f,P^i} \, \ell\left(M_\theta(P^i), f(\boldsymbol{x}_{i+1})\right) \\
&= \min_\theta \mathbb{E}_{P^i} \, \mathbb{E}_f \left[\ell\left(M_\theta(P^i), f(\boldsymbol{x}_{i+1})\right) \mid P^i\right] \\
&= \mathbb{E}_{P^i} \, \min_\theta \mathbb{E}_f \left[\ell\left(M_\theta(P^i), f(\boldsymbol{x}_{i+1})\right) \mid P^i\right].
\end{aligned}
$$

The inner minimization is seen to be achieved by $M_\theta(P^i) = \mathbb{E}_f\left[f(\boldsymbol{x}_{i+1}) \mid P^i\right]$ as we use the squared-error loss. This is the optimal solution for prompt $P^i$ and what we refer to as PME.

**PME for a task mixture.** We describe the PME for a mixture of function classes. For simplicity we confine ourselves to mixtures of two function classes; extension to more function classes is analogous. Let $\mathcal{F}_1$ and $\mathcal{F}_2$ be two function classes specified by probability distributions $\mathcal{D}_{\mathcal{F}_1}$ and $\mathcal{D}_{\mathcal{F}_2}$, resp. As in the single function class case, the inputs $\boldsymbol{x}$ are chosen i.i.d. from a common distribution $\mathcal{D}_{\mathcal{X}}$. For $\alpha_1, \alpha_2 \in [0,1]$ with $\alpha_1 + \alpha_2 = 1$, an $(\alpha_1, \alpha_2)$-mixture $\mathcal{F}$ of $\mathcal{F}_1$ and $\mathcal{F}_2$ is the meta-task in which the prompt $P = \left(\boldsymbol{x}_1, f(\boldsymbol{x}_i), \cdots, \boldsymbol{x}_p, f(\boldsymbol{x}_p), \boldsymbol{x}_{p+1}\right)$ is constructed by first picking task $\mathcal{F}_i$ with probability $\alpha_i$ for $i \in \{1, 2\}$ and then picking $f \sim \mathcal{D}_{\mathcal{F}_i}$. Thus $p_{\mathcal{F}}(f) = \alpha_1 p_{\mathcal{F}_1}(f) + \alpha_2 p_{\mathcal{F}_2}(f)$, where $p_{\mathcal{F}}(\cdot)$ is the probability density under function class $\mathcal{F}$ which defines $\mathcal{D}_{\mathcal{F}}$. For conciseness in the following we use $p_1(\cdot)$ for $p_{\mathcal{F}_1}(\cdot)$ etc. Now recall that PME for function class $\mathcal{F}$ is given by

$$
M_{\theta,\mathcal{F}}(P) = \mathbb{E}_{f \sim \mathcal{D}_{\mathcal{F}}}\left[f(\boldsymbol{x}_{p+1}) \mid P\right] = \int p_{\mathcal{F}}(f|P)\, f(x)\, \mathrm{d}f. \tag{2}
$$

Here $\mathrm{d}f$ is a volume element in $\mathcal{F}$; this makes sense as all our function families $\mathcal{F}$ are continuously parametrized. We would like to compute $M_{\theta,\mathcal{F}}(P)$ in terms of PMEs for $\mathcal{F}_1$ and $\mathcal{F}_2$. To this end, we first compute

$$
\begin{aligned}
p_{\mathcal{F}}(f|P) &= \frac{p_{\mathcal{F}}(P|f) p_{\mathcal{F}}(f)}{p_{\mathcal{F}}(P)} = \frac{p(P|f) p_{\mathcal{F}}(f)}{p_{\mathcal{F}}(P)} = \frac{p(P|f)}{p_{\mathcal{F}}(P)}\left[\alpha_1 p_1(f) + \alpha_2 p_2(f)\right] \\
&= \frac{\alpha_1 p_1(P)}{p_{\mathcal{F}}(P)} \frac{p(P|f) p_1(f)}{p_1(P)} + \frac{\alpha_2 p_2(P)}{p_{\mathcal{F}}(P)} \frac{p(P|f) p_2(f)}{p_2(P)} \\
&= \frac{\alpha_1 p_1(P)}{p_{\mathcal{F}}(P)} p_1(f|P) + \frac{\alpha_2 p_2(P)}{p_{\mathcal{F}}(P)} p_2(f|P) \\
&= \beta_1\, p_1(f|P) + \beta_2\, p_2(f|P),
\end{aligned}
$$

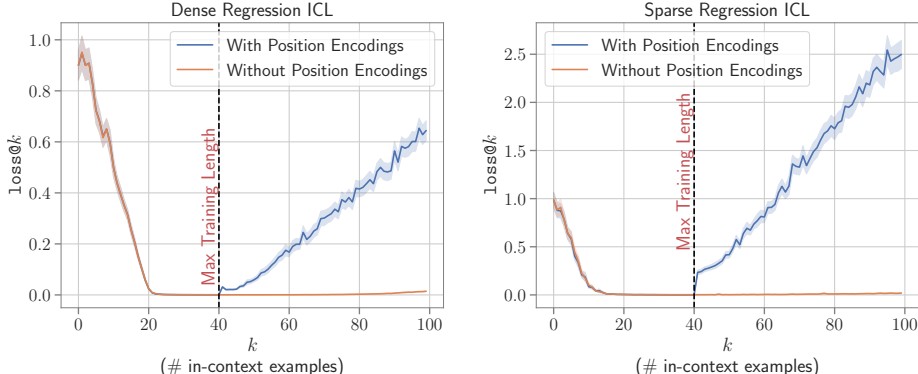

Figure 5: Impact of positional encodings on length generalization during in-context learning for dense and sparse linear regression tasks. For both tasks, the model was trained with $p = 40$ i.e. the maximum number of in-context examples provided.

where $\beta_1 = \frac{\alpha_1 p_1(P)}{p_{\mathcal{F}}(P)}$ and $\beta_2 = \frac{\alpha_2 p_2(P)}{p_{\mathcal{F}}(P)}$. Plugging this in equation 2 we get

$$M_{\theta,\mathcal{F}}(P) = \beta_1 \int p_1(f|P) \, f(x) \, \mathrm{d}f + \beta_2 \int p_2(f|P) \, f(x) \, \mathrm{d}f = \beta_1 M_{\theta,\mathcal{F}_1}(P) + \beta_2 M_{\theta,\mathcal{F}_2}(P).$$

(3)

## A.2 THE CURIOUS CASE OF POSITIONAL ENCODINGS.

Positional encodings both learnable or sinusoidal in transformer architectures have been shown to result in poor length generalization Bhattamishra et al. (2020); Press et al. (2022), i.e. when tested on sequences of lengths greater than those seen during training the performance tends to drop drastically. In our initial experiments, we observed this issue with length generalization in our in-context-learning setup as well (Figure 5). While there are now alternatives to the originally proposed position encodings like Rotary Embeddings Su et al. (2021) and ALiBi Press et al. (2022) which perform better on length generalization, we find that something much simpler works surprisingly well in our setup. We found that removing position encodings significantly improved the length generalization for both dense and sparse linear regression while maintaining virtually the same performance in the training regime as can be seen in Figure 5. These observations are in line with Bhattamishra et al. (2020) which shows that decoder-only transformers without positional encodings fare much better in recognizing formal languages as well as Haviv et al. (2022) that shows transformers language models without explicit position encodings can still learn positional information. Both works attribute this phenomenon to the presence of the causal mask in decoder-only models which implicitly provides positional information to these models. Hence by default in all our experiments, unless specified, we do not use any positional encodings while training our models.

## A.3 EXPERIMENTAL SETUP

We use Adam optimizer Kingma & Ba (2015) to train our models. We train all of our models with curriculum and observe that curriculum helps in faster convergence, i.e., the same optima can also be achieved by training the model for more training steps as also noted by Garg et al. (2022). Table 1 states the curriculum used for each experiment, where the syntax followed for each column specifying curriculum is [start, end, increment, interval]. The value of the said attribute goes from start to end, increasing by increment every interval train steps. Our experiments were conducted on a system comprising 32 NVIDIA V100 16GB GPUs. The cumulative training time of all models for this project was $\sim$ 30,000 GPU hours. While reporting the results, the error is averaged over 1280 prompts and shaded regions denote a 90% confidence interval over 1000 bootstrap trials.

We adapt Garg et al. (2022) code-base for our experiments. We use PytorchPaszke et al. (2019) and Huggingface TransformersWolf et al. (2020) libraries to implement the model architecture

Table 1: The values of curriculum attributes used for each experiment. $C_d, C_p$ and $C_{\text{freq}}$ denote the curriculum on number of input dimensions ($d$), number of points ($p$) and any other experiment-specific attribute respectively. For Fourier Series $C_{\text{freq}}$ refers to maximum frequency $N$)

| Experiment | Section | $C_d$ | $C_p$ | $C_{\text{freq}}$ |
|---|---|---|---|---|
| Dense, Sparse and Sign-Vector Regression | §B.1.1 | $[5, 20, 1, 2000]$ | $[10, 40, 2, 2000]$ | n/a |
| Low-Rank Regression | §B.1.1 | Fixed ($d = 100$) | Fixed ($p = 114$) | n/a |
| Fourier Series | §B.2.1 | Fixed ($d = 1$) | $[7, 43, 4, 2000]$ | $[1, 10, 1, 2000]$ |
| Fourier Series Mixture | §4 | Fixed ($d = 1$) | Fixed ($p = 40$) | Fixed ($N = 10$) |
| GMM Regression ($d = 10, p = 10$) | §3.1, §C.2 | $[5, 10, 1, 2000]$ | $[5, 10, 1, 2000]$ | n/a |
| GMM Regression ($d = 10, p = 20$) | §3.1, §C.2 | $[5, 10, 1, 2000]$ | $[10, 20, 2, 2000]$ | n/a |
| Degree-2 Monomial Basis Regression | §B.2.3 | Fixed ($d = 20$) | Fixed ($p = 290$) | n/a |
| Haar Wavelet Basis Regression | §B.2.4 | Fixed ($d = 1$) | Fixed ($p = 32$) | n/a |

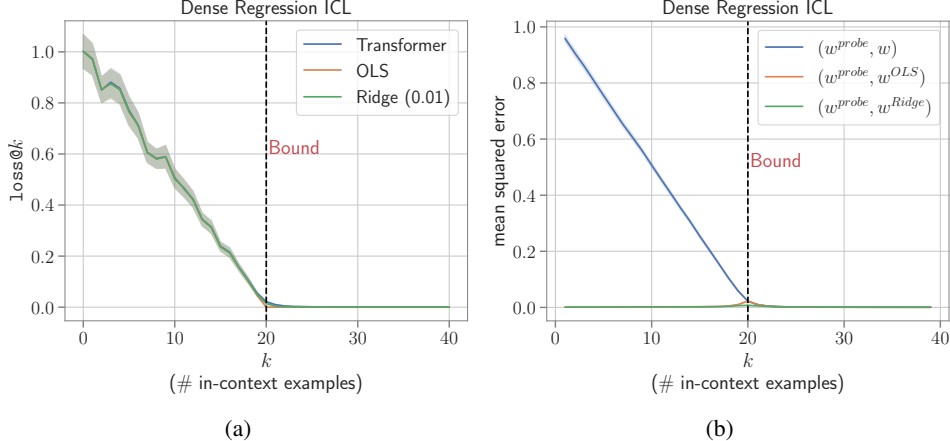

Figure 6: Results on the Dense Regression tasks mentioned in section §B.1.1.

and training procedure. For the baselines against which we compare transformers, we use `scikit-learn`'s[2] implementation of OLS, Ridge and Lasso, and for $L_\infty$ and $L_*$ norm minimization given the linear constraints we use CVXPY[3].

# B    LINEAR AND NON-LINEAR INVERSE PROBLEMS

Here, we discuss the results omitted from the §B.1.2 for conciseness. Figure 6 shows the results on the Dense Regression task and our experiments corroborate the findings of Akyürek et al. (2022), where transformers not only obtain errors close to OLS and Ridge regression for the dense regression task (Figure 6a) but the extracted weights also very closely align with weights obtained by the two algorithms (Figure 6b). This does indicate that the model is able to simulate the PME behavior for the dense regression class.

For sparse and sign-vector regression, we also visualize the weights recovered from the transformer for one of the functions for each family. As can be observed in Figure 8, for sparse regression at sufficiently high prompt lengths ($k > 10$), the model is able to recognize the sparse structure of the problem and detect the non-zero elements of the weight vector. Similarly, the recovered weights for sign-vector regression beyond $k > 10$, start exhibiting the sign-vector nature of the weights (i.e. each component either being +1 or -1).

We evaluate transformers on a family of linear and non-linear regression tasks. On the tasks where it is possible to compute the Bayesian predictor, we study how close the solutions obtained by the transformer and Bayesian predictor are. In this section, we focus only on single task ICL setting

---

[2] https://scikit-learn.org/stable/index.html
[3] https://www.cvxpy.org/

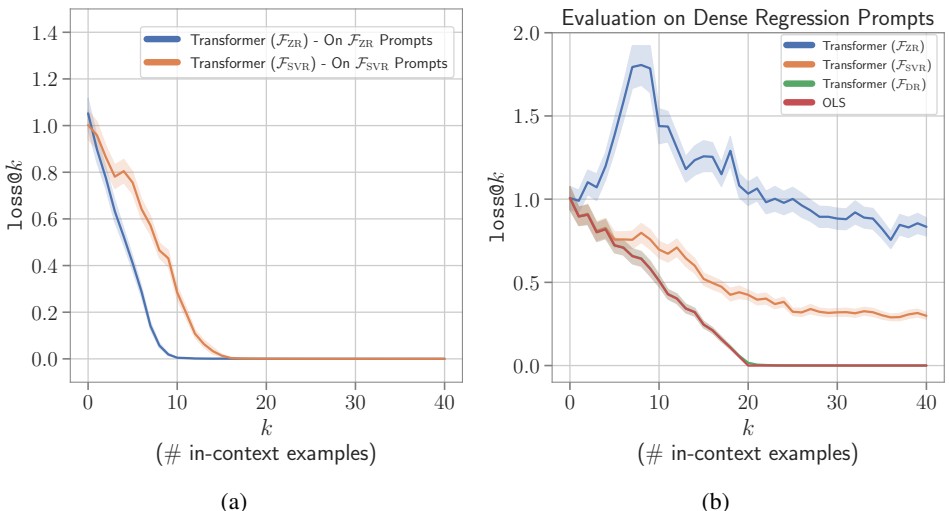

(a)                                                                        (b)

Figure 7: Evaluating transformer model trained for regression on task $\mathcal{F}_{\text{ZR}}$ with $w \in \{z; z \mid z \in \{-2, -1, 1, 2\}^{10}\}$, not satisfying the convex geometry conditions of Chandrasekaran et al. (2012). **Left:** Comparing the performance of this model, i.e. Transformer ($\mathcal{F}_{\text{ZR}}$), with Transformer ($\mathcal{F}_{\text{SVR}}$) when both are tested on their respective prompts. **Right:** Comparing the performance of Transformer ($\mathcal{F}_{\text{ZR}}$) with Transformer ($\mathcal{F}_{\text{SVR}}$) on Dense Regression ($\mathcal{F}_{\text{DR}}$) prompts. Transformer ($\mathcal{F}_{\text{ZR}}$) provides better performance than Transformer ($\mathcal{F}_{\text{SVR}}$) on in-distribution prompts but on OOD prompts (from $\mathcal{F}_{\text{DR}}$), Transformer ($\mathcal{F}_{\text{SVR}}$) performs better.

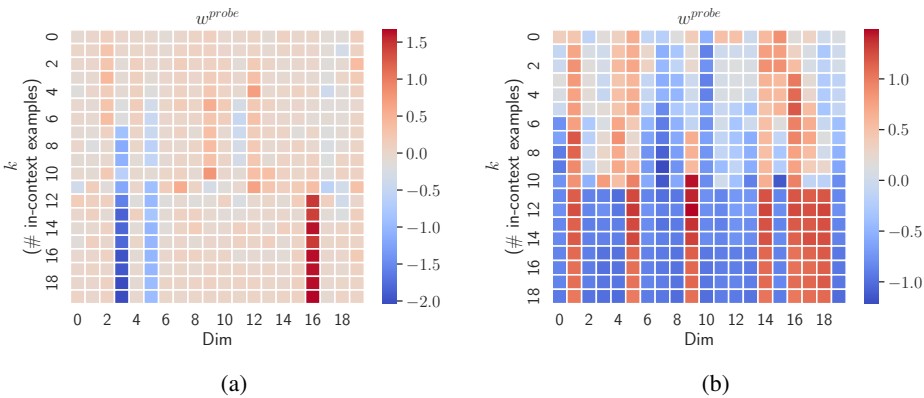

(a)                                                                        (b)

Figure 8: Visualizing recovered weights for sparse and sign vector regression for one of the examples in the test set.

(i.e. the model is trained to predict functions from a single family), while the mixture of tasks is discussed §3.

## B.1 LINEAR INVERSE PROBLEMS

In this section, the class of functions is fixed to the class of linear functions across all problems, i.e. $\mathcal{F} = \left\{ f : \boldsymbol{x} \mapsto \boldsymbol{w}^T \boldsymbol{x} \,|\, \boldsymbol{w} \in \mathbb{R}^d \right\}$; what varies across the problems is the distribution of $\boldsymbol{w}$. Problems in this section are instances of linear inverse problems. Linear inverse problems are classic problems arising in diverse applications in engineering, science, and medicine. In these problems, one wants to estimate model parameters from a few linear measurements. Often these measurements are expensive and can be fewer in number than the number of parameters ($p < d$). Such seemingly ill-posed problems can still be solved if there are structural constraints satisfied by the parameters. These constraints can take many forms from being sparse to having a low-rank structure. The sparse case was addressed by a famous convex programming approach Candes & Tao (2005); Donoho (2006) also known as compressed sensing. This was greatly generalized in later work to apply to many more types of inverse problems; see Chandrasekaran et al. (2012). In this section, we will show that transformers can solve many inverse problems in context—in fact all problems that we tried. The problem-specific structural constraints are encoded in the prior for $\boldsymbol{w}$.

### B.1.1 FUNCTION CLASSES AND BASELINES

**Dense Regression ($\mathcal{F}_{\textbf{DR}}$).** This represents the simplest case of linear regression as studied in Garg et al. (2022); Akyürek et al. (2022); von Oswald et al. (2022), where the prior on $\boldsymbol{w}$ is the standard Gaussian i.e. $\boldsymbol{w} \sim \mathcal{N}(\mathbf{0}_d, \boldsymbol{I})$. We are particularly interested in the underdetermined region i.e. $k < d$. Gaussian prior enables explicit PME computation: both PME and maximum a posteriori (MAP) solution agree and are equal to the minimum $L_2$-norm solution of the equations forming the training examples, i.e. $\min_{\boldsymbol{w}} \|\boldsymbol{w}\|_2$ s.t. $\boldsymbol{w}^T \boldsymbol{x}_i = f(\boldsymbol{x}_i), \forall i \leq k$. Standard Ordinary Least Squares (OLS) solvers return the minimum $L_2$-norm solution, and thus PME and MAP too, in the underdetermined region, i.e. $k < d$.

**Skewed-Covariance Regression ($\mathcal{F}_{\textbf{Skew-DR}}$).** This setup is similar to dense-regression, except that we assume the following prior on weight vector: $\boldsymbol{w} \sim \mathcal{N}(0, \boldsymbol{\Sigma})$, where $\boldsymbol{\Sigma} \in \mathbb{R}^{d \times d}$ is the covariance matrix with eigenvalues proportional to $1/i^2$, where $i \in [1, d]$. For this prior on $\boldsymbol{w}$, we can use the same (but more general) argument for dense regression above to obtain the PME and MAP which will be equal and can be obtained by minimizing $\boldsymbol{w}^T \boldsymbol{\Sigma}^{-1} \boldsymbol{w}$ w.r.t to the constraints $\boldsymbol{w}^T \boldsymbol{x}_i = f(\boldsymbol{x}_i)$. This setup was motivated by Garg et al. (2022), where it was used to sample $\boldsymbol{x}_i$ values for out-of-distribution (OOD) evaluation, but not as a prior on $\boldsymbol{w}$.

**Sparse Regression ($\mathcal{F}_{\textbf{SR}}$).** In sparse regression, we assume $\boldsymbol{w}$ to be an $s$-sparse vector in $\mathbb{R}^d$ i.e. out of its $d$ components only $s$ are non-zero. Following Garg et al. (2022), to sample $\boldsymbol{w}$ for constructing prompts $P$, we first sample $\boldsymbol{w} \sim \mathcal{N}(\mathbf{0}_d, \boldsymbol{I})$ and then randomly set its $d - s$ components as 0. We consider $s = 3$ throughout our experiments. While computing the PME appears to be intractable here, the MAP solution can be estimated using Lasso by assuming a Laplacian prior on $\boldsymbol{w}$ Tibshirani (1996). We tune the Lasso coefficient following Garg et al. (2022), i.e., by using a separate batch of data (1280 samples) and choose the single value that achieves the smallest loss.

**Sign-Vector Regression ($\mathcal{F}_{\textbf{SVR}}$).** Here, we assume $\boldsymbol{w}$ to be a sign vector in $\{-1, +1\}^d$. For constructing prompts $P$, we sample $d$ independent Bernoulli random variables $b_j$ with a mean of $0.5$ and obtain $\boldsymbol{w} = [2b_1 - 1, \cdots, 2b_d - 1]^T$. While computing the exact PME remains intractable in this case as well, the optimal solution for $k > d/2$ can be obtained by minimizing the $L_\infty$-norm $\|\boldsymbol{w}\|_\infty$ w.r.t. the constraints specified by the input-output examples ($\boldsymbol{w}^T \boldsymbol{x}_i = f(\boldsymbol{x}_i)$) Mangasarian & Recht (2011).

**Low-Rank Regression ($\mathcal{F}_{\textbf{LowRank-DR}}$).** In this case, $\boldsymbol{w}$ is assumed to be a flattened version of a matrix $\boldsymbol{W} \in \mathbb{R}^{q \times q}$ ($d = q^2$) with a rank $r$, where $r \ll q$. A strong baseline, in this case, is to minimize the nuclear norm $L_*$ of $\boldsymbol{W}$, i.e. $\|\boldsymbol{W}\|_*$ subject to constraints $\boldsymbol{w}^T \boldsymbol{x}_i = f(\boldsymbol{x}_i)$. To sample

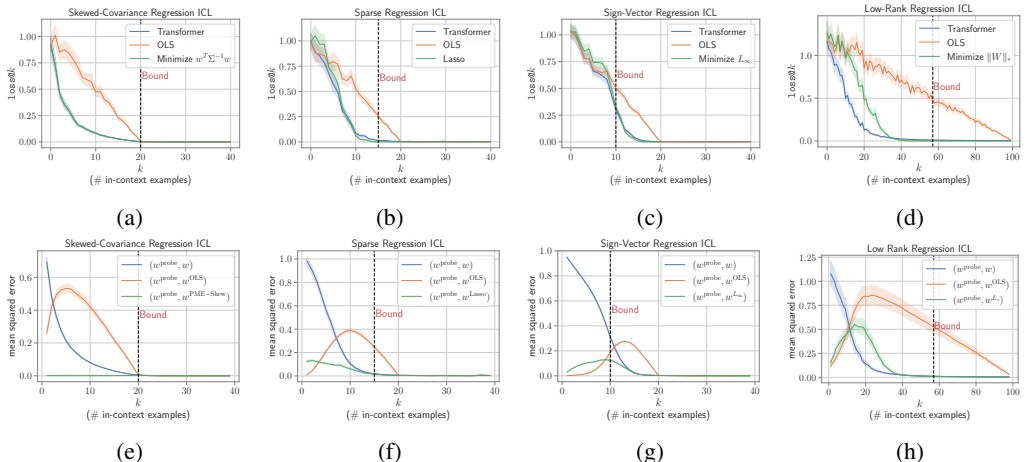

(a)        (b)        (c)        (d)

(e)        (f)        (g)        (h)

Figure 9: Comparing ICL in transformers for different linear functions with the relevant baselines. **Top**: `loss@k` values for transformers and baselines on skewed covariance, sparse, sign-vector, and low-rank regression tasks. **Bottom**: Comparing the errors between the implicit weights recovered from transformers $w^{\text{probe}}$ with the ground truth weights $w$ and weights computed by different baselines. $\boldsymbol{w}^{\text{PME-Skew}}$ denotes the weights obtained by minimizing $\boldsymbol{w}^T\boldsymbol{\Sigma}^{-1}\boldsymbol{w}$ for the skewed covariance regression task.

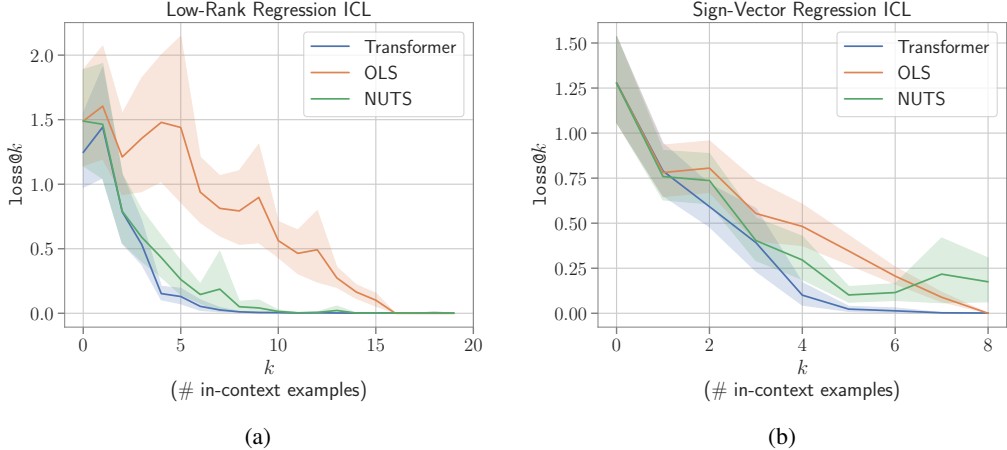

(a)                                (b)

Figure 10: Computing the PME by Markov Chain Monte Carlo sampling method (NUTS) for (a) Low-Rank Regression, and (b) Sign-Vector Regression. The problem dimension $d$ for Low-Rank Regression and Sign-Vector Regression is 16 ($4 \times 4$ matrix) and 8 respectively. As can be seen, the Transformer is close to the respective NUTS-approximated PME in both cases.

the rank-$r$ matrix $\boldsymbol{W}$, we sample $\boldsymbol{A} \sim \mathcal{N}(0, 1)$, s.t. $\boldsymbol{A} \in \mathbb{R}^{q \times r}$ and independently a matrix $\boldsymbol{B}$ of the same shape and distribution, and set $\boldsymbol{W} = \boldsymbol{A}\boldsymbol{B}^T$.

**An artificial task.** Techniques in prior work such as Chandrasekaran et al. (2012) require that for the exact recovery of a vector $\boldsymbol{w}$, the set of all these vectors must satisfy specific convexity conditions. However this requirement seems to be specific to these techniques, and in particular it's not clear if a Bayesian approach would need such a condition. To test this we define a task $\mathcal{F}_{\text{ZR}}$ where the convexity conditions are not met and train transformers for regression on this task. Here, $\boldsymbol{w} \in \{\boldsymbol{z}\boldsymbol{z} \mid \boldsymbol{z} \in \{-2, -1, 1, 2\}^{d/2}\}$, where $\boldsymbol{z}\boldsymbol{z}$ denotes $\boldsymbol{z}$ concatenated to itself. Note that the size of this set is $2^d$, the same as the size of $\{-1, 1\}^d$, and many elements, such as $zz$ with $z \in \{-1, 1\}^{d/2}$, lie strictly inside the convex hull.

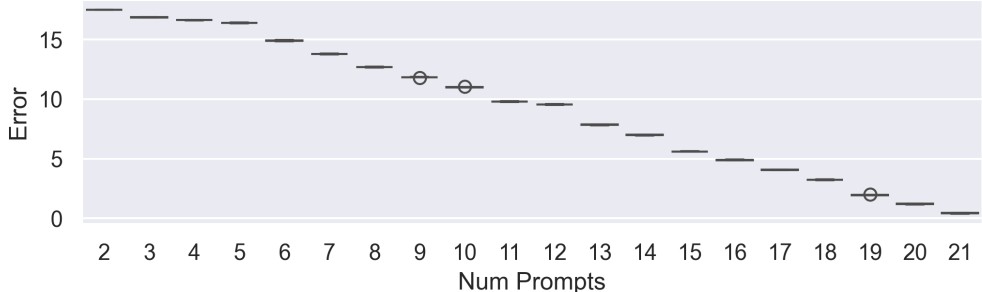

Figure 11: Box plot showing the effect of the order of prompts on the Transformer's ICL performance for Dense Regression. On x-axis we plot the number of in-context examples and on y-axis we have the quartiles for the errors obtained for different permutations of the examples. At each prompt length we consider 20 permutations. As can be seen, all the boxes are nearly flat, meaning that the errors have nearly zero variance with the permutations, and hence the model in this case is robust to the order of prompts.

**Recovery bounds.**    For each function class above, there is a bound on the minimum number of in-context examples needed for the exact recovery of the solution vector $\boldsymbol{w}$. The bounds for sparse, sign-vector and low-rank regression are $2s\log(d/s)+5s/4$, $d/2$, and $3r(2q-r)$ respectively Chandrasekaran et al. (2012).

### B.1.2   RESULTS

We train transformer-based models on the five tasks following §2.2. Each model is trained with $d = 20$ and $p = 40$, excluding Low-Rank Regression where we train with $d = 100$, $p = 114$, and $r = 1$. Figures 9b-9d compare the `loss@k` values on these tasks with different baselines. Additionally, we also extract the implied weights $\boldsymbol{w}^{\mathrm{probe}}$ from the trained models when given a prompt $P$ following Akyürek et al. (2022) by generating model's predictions $\{y_i'\}$ on the test inputs $\{\boldsymbol{x}_i'\}_{i=1}^{2d} \sim \mathcal{D}_{\mathcal{X}}$ and then solving the system of equations to recover $\boldsymbol{w}^{\mathrm{probe}}$. We then compare the implied weights $\boldsymbol{w}^{\mathrm{probe}}$ with the ground truth weights $\boldsymbol{w}$ as well as the weights extracted from different baselines to better understand the inductive biases exhibited by these models during in-context learning (Figures 9f-9h).

**Comparison with exact PME.**    Since results for dense regression have been already covered in Akyürek et al. (2022), we do not repeat them here, but for completeness provide them in Figure 6. For skewed-covariance regression, we observe that the transformer follows the PME solution very closely both in terms of the `loss@k` values (Figure 9a) as well as the recovered weights for which the error between $\boldsymbol{w}^{\mathrm{probe}}$ and $\boldsymbol{w}^{\mathrm{PME-Skew}}$ (weights obtained by minimizing $\boldsymbol{w}^T\boldsymbol{\Sigma}^{-1}\boldsymbol{w}$) is close to zero at all prompt lengths (Figure 9e).

**Comparison with numerical solutions.**    For Low-Rank Regression and Sign-Vector Regression, we provide comparisons with the numerical PME solutions obtained using No-U-Turn Sampling (NUTS) in Figure 10 and find that errors by transformers strongly agree with those of the numerical solution, and in some instances transformers actually perform slightly better (which we attribute to the numerical solutions also being an approximation).

**Comparison with strong baselines from Chandrasekaran et al. (2012)**    As can be seen in Figure 9, on all the tasks, transformers perform better than OLS and are able to solve the problem with $< d$ samples i.e. underdetermined region meaning that they are able to understand the structure of the problem. The error curves of transformers for the tasks align closely with the errors of Lasso (Figure 9b), $L_\infty$ minimization (Figure 9c), and $L_*$ minimization (Figure 9d) baselines for the respective tasks. Interestingly for low-rank regression transformer actually performs better. Though, due to the larger problem dimension, ($d = 100$) in this, it requires a bigger model: 24 layers, 16 heads, and 512 hidden size. In Figures 9f, 9g, and 9h, we observe that at small prompt lengths $w^{\mathrm{probe}}$ and $w^{\mathrm{OLS}}$ are close. We conjecture that this might be attributed to both $w^{\mathrm{probe}}$ and $w^{\mathrm{OLS}}$ being close to 0

for small prompt lengths (Figure 8). Prior distributions for all three tasks are centrally-symmetric, hence, at small prompt lengths when the posterior is likely to be close to the prior, the PME is close to the mean of the prior which is 0. At larger prompt lengths transformers start to agree with $\boldsymbol{w}^{\mathrm{Lasso}}$, $\boldsymbol{w}^{L_\infty}$, and $\boldsymbol{w}^{L_*}$. This is consistent with the transformer following PME, assuming $\boldsymbol{w}^{\mathrm{Lasso}}$, $\boldsymbol{w}^{L_\infty}$, and $\boldsymbol{w}^{L_*}$ are close to PME—we leave it to future work to determine whether this is true (note that for sparse regression Lasso approximates the MAP estimate which should approach the PME solution as more data is observed). The recovered weights $\boldsymbol{w}^{\mathrm{probe}}$ also agree with $\boldsymbol{w}^{\mathrm{Lasso}}$, $\boldsymbol{w}^{L_\infty}$, and $\boldsymbol{w}^{L_*}$ for their respective tasks after sufficient in-context examples are provided.

Finally, refer to Figure 7 for the results on task $\mathcal{F}_{\mathrm{ZR}}$. We observe that Transformers trained on this task ($\mathcal{F}_{\mathrm{ZR}}$) provide better performance than those trained on Sign-Vector Regression ($\mathcal{F}_{\mathrm{SVR}}$). Therefore, we can conclude that Transformers do not require any convexity conditions on weight vectors.

## B.2 Non-linear functions

Moving beyond linear functions, we now study how well transformers can in-context learn function classes with more complex relationships between the input and output, and if their behavior resembles the ideal learner i.e. the PME. Particularly, we consider the function classes of the form $\mathcal{F}_\Phi = \left\{ f(\cdot; \Phi) | f(\boldsymbol{x}; \Phi) = \boldsymbol{w}^T \Phi(\boldsymbol{x}), \boldsymbol{w} \in \mathbb{R}^\Delta \right\}$, where $\Phi : \mathbb{R}^d \to \mathbb{R}^\Delta$ maps the input vector $\boldsymbol{x}$ to an alternate feature representation. This corresponds to learning the mapping $\Phi(\boldsymbol{x})$ and then performing linear regression on top of it. Under the assumption of a standard Gaussian prior on $\boldsymbol{w}$, the PME for the dense regression can be easily extended for $\mathcal{F}_\Phi$: $\min_{\boldsymbol{w}} \|\boldsymbol{w}\|_2$, s.t. $\boldsymbol{w}^T \Phi(\boldsymbol{x}_i) = f(\boldsymbol{x}_i)$ for $i \in \{1, \cdots, p\}$.

### B.2.1 Fourier Series

A Fourier series is an expansion of a periodic function into a sum of trigonometric functions. One can represent the Fourier series using the sine-cosine form given by:

$$f(x) = a_0 + \sum_{n=1}^{N} a_n \cos\left(n\pi x/L\right) + \sum_{n=1}^{N} b_n \sin\left(n\pi x/L\right)$$

where, $x \in [-L, L]$, and $a_0$, $a_n$'s and $b_n$'s are known as Fourier coefficients and $\cos n\pi/L$ and $\sin n\pi/L$ define the frequency $n$ components. We can define the function class $\mathcal{F}_{\Phi_N}^{\mathrm{fourier}}$ by considering $\Phi$ as the Fourier feature map i.e. $\Phi_N(x) = [1, \cos(\pi x/L), \cdots, \cos(N\pi x/L), \sin(\pi x/L), \cdots, \sin(N\pi x/L)]^T$, and $\boldsymbol{w}$ as Fourier coefficients: $\boldsymbol{w} = [a_0, a_1, \cdots, a_N, b_1, \cdots, b_N]$. Hence, $\Phi_N(x) \in \mathbb{R}^d$ and $\boldsymbol{w} \in \mathbb{R}^d$, where $d = 2N + 1$.

For training transformers to in-context-learn $\mathcal{F}_{\Phi_N}^{\mathrm{fourier}}$, we fix a value of $N$ and sample functions $f \in \mathcal{F}_{\Phi_N}^{\mathrm{fourier}}$ by sampling the Fourier coefficients from the standard normal distribution i.e. $\boldsymbol{w} \sim \mathcal{N}(\boldsymbol{0}_d, \boldsymbol{I})$. We consider the inputs to be scalars, i.e. $x_i \in [-L, L]$ and we sample them i.i.d. from the uniform distribution on the domain: $x_i \sim \mathcal{U}(-L, L)$. In all of our experiments, we consider $N = 10$ and $L = 5$. At test time we evaluate on $\mathcal{F}_{\Phi_M}^{\mathrm{fourier}}$ for $M \in [1, 10]$, i.e. during evaluation we also prompt the model with functions with different maximum frequency as seen during training. As a baseline, we use OLS on the Fourier features (denoted as OLS Fourier Basis) which will be equivalent to the PME.

**Measuring inductive biases.** Once we train a transformer-based model to in-context learn $\mathcal{F}_{\Phi_N}^{\mathrm{fourier}}$, how can we investigate the inductive biases that the model learns to solve the problem? We would like to answer questions such as, when prompted with $k$ input-output examples what are the prominent frequencies in the function simulated by the model, or, how do these exhibited frequencies change as we change the value of $k$? We start by sampling in-context examples $(x_1, f(x_1), \cdots x_k, f(x_k))$, and given the context obtain the model's predictions on a set of $m$ test inputs $\{x_i'\}_{i=1}^m$, i.e. $y_i' = M_\theta\left((x_1, f(x_1), \cdots x_k, f(x_k), x_i')\right)$. We can then perform Discrete Fourier Transform (DFT) on $\{y_1', \cdots, y_m'\}$ to obtain the Fourier coefficients of the function output by $M$, which we can analyze to understand the dominant frequencies.

**Results.** The results of our experiments concerning the Fourier series are provided in Figure 12. Transformers obtain `loss@k` values close to the OLS Fourier Basis baseline (Figure 12a) indicating

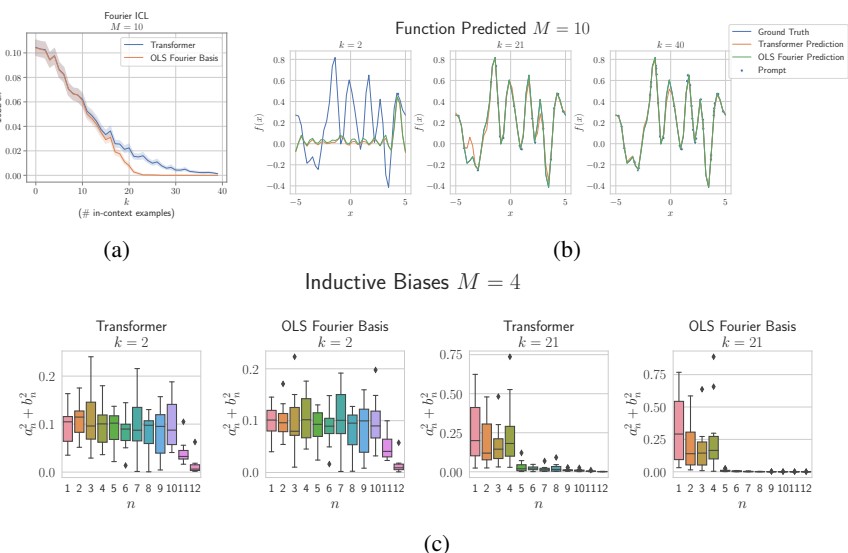

(a) (b)

(c)

Figure 12: Effectiveness of ICL in transformers for Fourier series family of functions. **Top left**: loss@$k$ values for transformer and OLS Fourier Basis baseline. **Top Right**: Visualizing the functions simulated by the transformer and the OLS Fourier Basis. **Bottom**: Measuring the frequencies of the simulated function by the transformer and the baseline.

at least for the smaller prompt lengths the model is able to simulate the behavior of the ideal predictor (PME). These plots use 12-layer transformers to obtain results, but we also investigate if bigger models help. Figure 13 plots bigger models with 18 and 21 layers where the agreement with PME is much better. Further, in Figures 12b and 12c, we could only discuss results for a subset of values of $M$ and $k$. The function visualizations for the transformer and Fourier OLS baseline for different combinations of $M$ and $k$ are provided in Figure 15. We have observations consistent with Figure 12b, where the function outputs of the transformer and the baseline align closely. Similarly, in Figure 14, we present the distribution of frequencies in the predicted functions for the two methods and again observe consistent findings.

Since the inputs $x_i$, in this case, are scalars, we can visualize the functions learned in context by transformers. We show one such example for a randomly selected function $f \sim \mathcal{F}_{\Phi_M}^{\text{fourier}}$ for prompting the model in Figure 12b. As can be observed, the functions predicted by both the transformer and baseline have a close alignment, and both approach the ground truth function $f$ as more examples are provided. Finally, we visualize the distribution of the frequencies for the predicted functions in Figure 12c. For a value of $M$, we sample 10 different functions and provide $k$ in-context examples to the model to extract the frequencies of the predicted functions using the DFT method. As can be observed, when provided with fewer in-context examples ($k = 2$) both Transformer and the baseline predict functions with all the 10 frequencies (indicated by the values of $a_n^2 + b_n^2$ in a similar range for $n \in [1, 10]$), but as more examples are provided they begin to recognize the gold maximum frequency (i.e. $M = 4$). The function visualizations for the transformer and Fourier OLS baseline for different combinations of $M$ and $k$ are provided in Figure 15. We have observations consistent with Figure 12b, where the function outputs of the transformer and the baseline align closely. Similarly, in Figure 14, we present the distribution of frequencies in the predicted functions for the two methods and again observe consistent findings. This suggests that the transformers are following the Bayesian predictor and are not biased towards smaller frequencies.

### B.2.2 RANDOM FOURIER FEATURES

Mapping input data to random low-dimensional features has been shown to be effective to approximate large-scale kernels Rahimi & Recht (2007). In this section, we are particularly interested in Random Fourier Features (RFF) which can be shown to approximate the Radial Basis Function kernel and are given as:

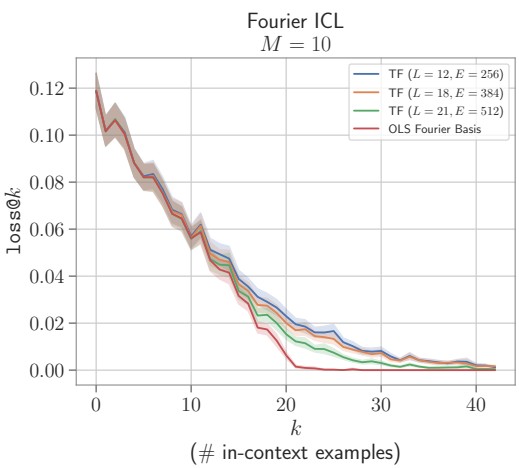

Figure 13: **Bigger models achieve better results on the Fourier Series task.** Plotting the squared error (averaged over 1280 prompts) for bigger transformer (TF) models trained for 500k steps on the Fourier Series task. Training setup is the same as used for the model plotted in Figure 12a (Section B.2.1), which is also plotted here (blue color) for comparison. $L$ and $E$ denote the number of layers and embedding size for TF models respectively.

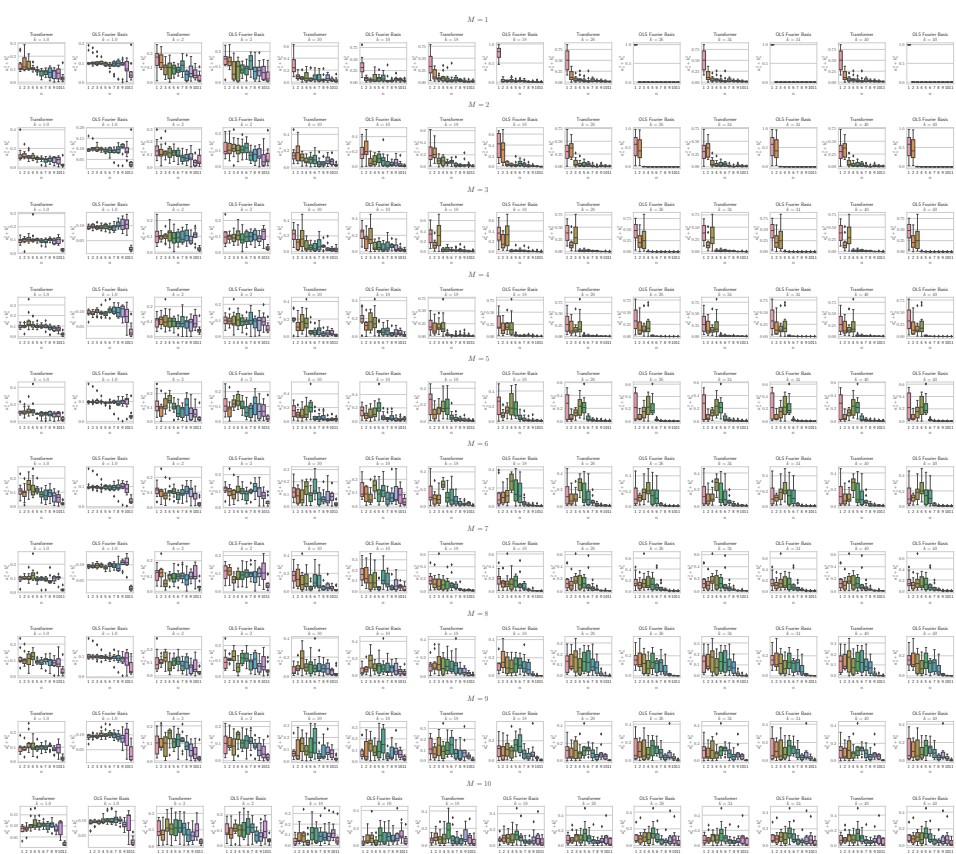

Figure 14: Measuring the frequencies of the simulated function by the transformer and the baseline for different values of $M$ (maximum frequency) and $k$ (number of in-context examples)

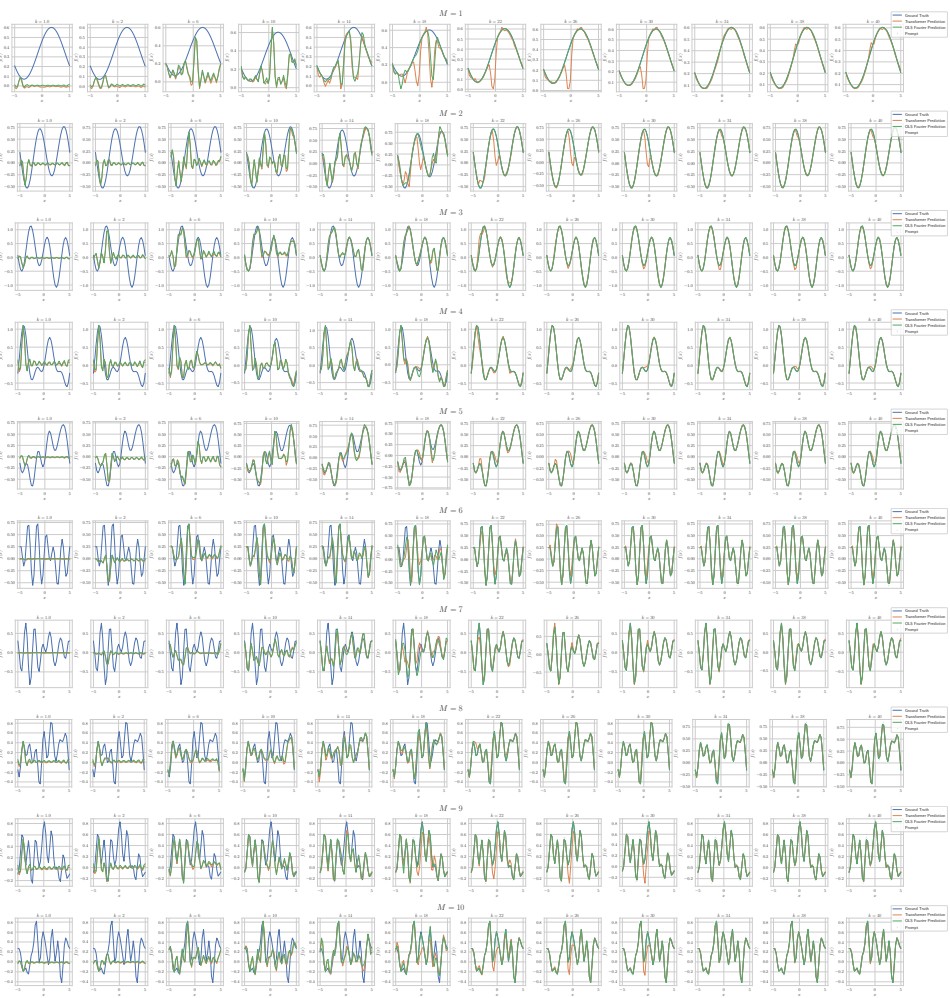

Figure 15: Visualizing the functions simulated by the transformer and the OLS Fourier Basis, for different values of $M$ (maximum frequency) and $k$ (number of in-context examples)

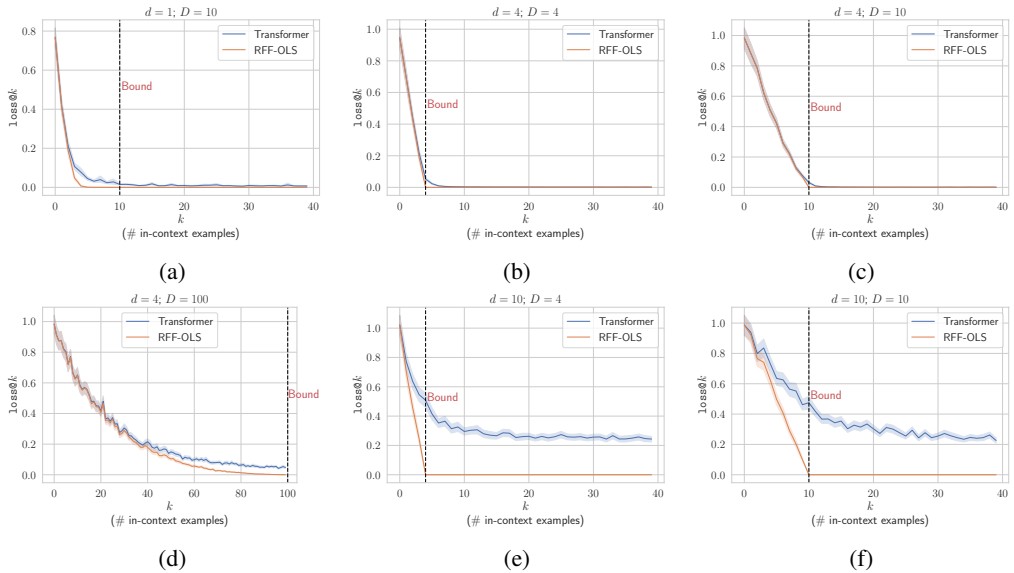

Figure 16: Comparing transformers performance on RFF function family ($\mathcal{F}_{\Phi_D}^{\text{RFF}}$) with the RFF-OLS baseline for different values of $d$ and $D$.

$$\Phi_D(\boldsymbol{x}) = \sqrt{\frac{2}{D}} [\cos{(\boldsymbol{\omega}_1^T \boldsymbol{x} + \delta_1)}, \cdots, \cos{(\boldsymbol{\omega}_D^T \boldsymbol{x} + \delta_D)}]^T$$

where $\boldsymbol{\omega}_i \in \mathbb{R}^d$ and $\delta_i \in \mathbb{R} \ \forall i \in [1, D]$, such that $\Phi_D : \mathbb{R}^d \to \mathbb{R}^D$. Both $\boldsymbol{\omega}_i$ and $\delta$ are sampled randomly, such that $\boldsymbol{\omega}_i \in \mathcal{N}(\boldsymbol{0}, \boldsymbol{I}_d)$ and $\delta_i \in (0, 2\pi)$. We can then define the function family $\mathcal{F}_{\Phi_D}^{\text{RFF}}$ as linear functions over the random fourier features i.e. $f = \boldsymbol{w}^T \Phi_D(\boldsymbol{x})$ such that $f \sim \mathcal{F}_{\Phi_D}^{\text{RFF}}$. While training the transformer on this function class, we sample $\boldsymbol{\omega}_i$'s and $\delta_i$'s once and keep them fixed throughout the training. As a baseline, we use OLS over $(\Phi_D(x), y)$ pairs which will give the PME for the problem (denote this as RFF-OLS).

**Results.** For this particular family, we observed mixed results for transformers, i.e. they fail to generalize to functions of the family when the complexity of the problem is high. The complexity of this function class is dictated by the length of the $\boldsymbol{\omega}_i$ vectors (and the inputs $\boldsymbol{x}$) i.e. $d$ and the number of random features $D$. We plot the loss@$k$ values for transformer models trained on $\mathcal{F}_{\Phi_D}^{\text{RFF}}$ for different values of $d$ and $D$ in Figure 16. As can be observed, the complexity of the problem for the transformers is primarily governed by $d$, where they are able to solve the tasks for even large values of $D$, however, while they perform well for smaller values of $d$ ($d = 1$ and $d = 4$), for $d = 10$, they perform much worse compared to the RFF-OLS baseline and the loss@$k$ doesn't improve much once $\sim 15$ in-context examples are provided.

### B.2.3 DEGREE-2 MONOMIAL BASIS REGRESSION

As stated in §B.2.1, the Fourier Series function class can be viewed as linear regression over the Fourier basis consisting of sinusoidal functions. Similarly, we define a function class $\mathcal{F}_{\Phi_M}^{\text{mon}(2)}$ with the basis formed by degree-2 monomials for any $d$-dimensional input vector $\boldsymbol{x}$.

Using the notation introduced in B.1.1 the basis for $\mathcal{F}_{\Phi_M}^{\text{mon}(2)}$ is defined as $\Phi_M(\boldsymbol{x}) = \{x_i x_j \mid 1 \leq i, j \leq d\}$. Each function $f \in \mathcal{F}_{\Phi_M}^{\text{mon}(2)}$ is a linear combination of basis and $\boldsymbol{w}$ i.e. $f(\boldsymbol{x}) = \boldsymbol{w}^T \Phi_M(\boldsymbol{x})$, where $\boldsymbol{w}$ is a $|\Phi_M|$-dimensional vector sampled from standard normal distribution.

For experimentation, we define a sub-family $\mathcal{F}_{\mathcal{S}}^{\text{mon}(2)}$ under $\mathcal{F}_{\Phi_M}^{\text{mon}(2)}$ by choosing a proper subset $\mathcal{S} \subset \Phi_M$ and linearly combining the terms in $\mathcal{S}$ to form $f$. This is equivalent to explicitly setting coefficients $w_i$ of terms in $\Phi_M - \mathcal{S}$ to 0. We experiment with $d = 20$, with the prompt length

$p = 290$ and $|\mathcal{S}| = 20$. We do not use curriculum ($d, p, |\mathcal{S}|$ are fixed for the entire duration of the training run).

**Baselines.** We use OLS fitted to the following bases as baselines: $\mathcal{S}$ basis (OLS$_\mathcal{S}$), all degree-2 monomials i.e., $\Phi_M$ basis (OLS$_{\Phi_M}$), and to a basis of all polynomial features up to degree-2 (OLS$_{\text{poly.}(2)}$). We also compare Lasso ($\alpha = 0.01$) fitted to all degree-2 monomials i.e., $\Phi_M$ basis (Lasso$_{\Phi_M}$) as a baseline.

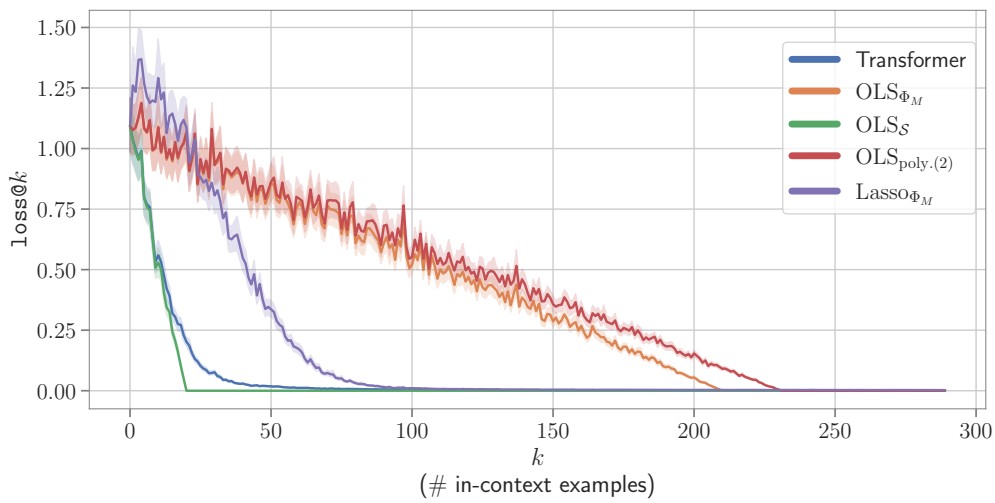

Figure 17: **In-Distribution evaluation results on $\mathcal{F}_\mathcal{S}^{\mathbf{mon(2)}}$ sub-family of degree-2 monomial basis regression.** Evaluation of transformer on prompts generated using the same $\mathcal{S}$ used during training.

**Results.** In Figure 17, we show the In-Distribution (ID) evaluation results for the $\mathcal{F}_\mathcal{S}^{\text{mon}(2)}$ experiments. Here, the test prompts contain functions formed by $\mathcal{S}$ (the same basis used during training). We observe that Transformers closely follow OLS$_\mathcal{S}$. The increasing order of performance (decreasing `loss@k` for $k \geq |\mathcal{S}|$) of different solvers is: OLS$_{\text{poly.}(2)} \leq$ OLS$_{\Phi_M} <$ Lasso$_{\Phi_M} <$ Transformers $<$ OLS$_\mathcal{S}$. Transformer's squared error takes a little longer than OLS$_\mathcal{S}$ to converge. Lasso$_{\Phi_M}$ is able to take the advantage of sparsity of the problem and is hence better than both OLS$_{\Phi_M}$ and OLS$_{\text{poly.}(2)}$, which respectively converge at $k = 210$ and $k = 231^4$. We also conduct an Out-of-Distribution (OOD) evaluation for $\mathcal{F}_\mathcal{S}^{\text{mon}(2)}$, whose results are shown in Figure 18. Here, we generate prompts from a basis $\mathcal{S}' \subset \Phi_M$ of the same size as $\mathcal{S}$ but differing from $\mathcal{S}$ in $n$ degree-2 terms, i.e. $|\mathcal{S}' - \mathcal{S}| = n$. We show the results for different values of $n$. Figure 18a shows the OLS$_\mathcal{S}$ undergoes a steep rise in errors momentarily at $k = |\mathcal{S}|$ (double descent). Figure 18b zooms into the lower error region of Figure 18a where we notice that Transformer mimics OLS$_\mathcal{S}$, while OLS$_{\mathcal{S}'}$ is the best-performing baseline (since it fits to the $\mathcal{S}'$ basis used to construct the prompts). Transformer does not undergo double descent (for $n = 1$) and is hence momentarily better than OLS$_\mathcal{S}$ at $k = |\mathcal{S}|$. Similar plots are shown for $n \in \{2, 3, 4, 5, 10, 15, 20\}$. As $n$ increases, the height of OLS$_\mathcal{S}$ peak increases and the Transformer also starts to have a rise in errors at $k = |\mathcal{S}|$. For $n = 20$, $\mathcal{S}'$ and $\mathcal{S}$ have nothing in common, and Transformer still follows OLS$_\mathcal{S}$ (OLS fitted to the training basis $\mathcal{S}$). As mentioned under §B.2, when the prior on weights $\boldsymbol{w}$ is Gaussian, the PME is the minimum $L_2$-norm solution. For $\mathcal{F}_\mathcal{S}^{\text{mon}(2)}$, that solution is given by OLS$_\mathcal{S}$. Therefore, the results suggest that the transformer is computing PME. In summary, transformers closely follow OLS$_\mathcal{S}$ in this set-up, and more so on the OOD data, where they even surpass OLS$_\mathcal{S}$'s performance when it experiences double descent.

---

[4]210 and 231 are the sizes of the bases to which OLS$_{\Phi_M}$ and OLS$_{\text{poly.}(2)}$ are fitted. Hence, they converge right when the problem becomes determined in their respective bases.

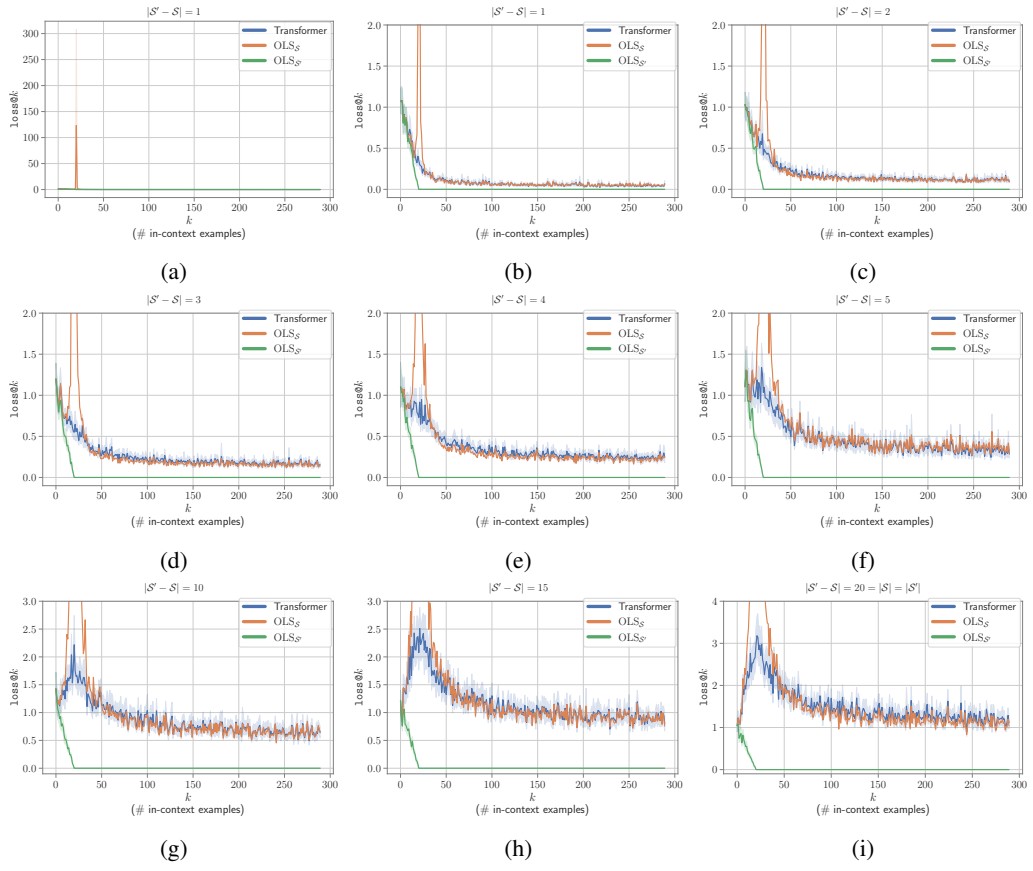

Figure 18: **Out-of-Distribution evaluation results on $\mathcal{F}_{\mathcal{S}}^{\mathbf{mon(2)}}$ sub-family of degree-2 monomial basis regression.** Evaluation of transformer trained on prompts generated using $\mathcal{S}'$, where $\mathcal{S}'$ contains $n$ degree-2 monomials not present in $\mathcal{S}$ that was used during training. We show results for different values of $n$.

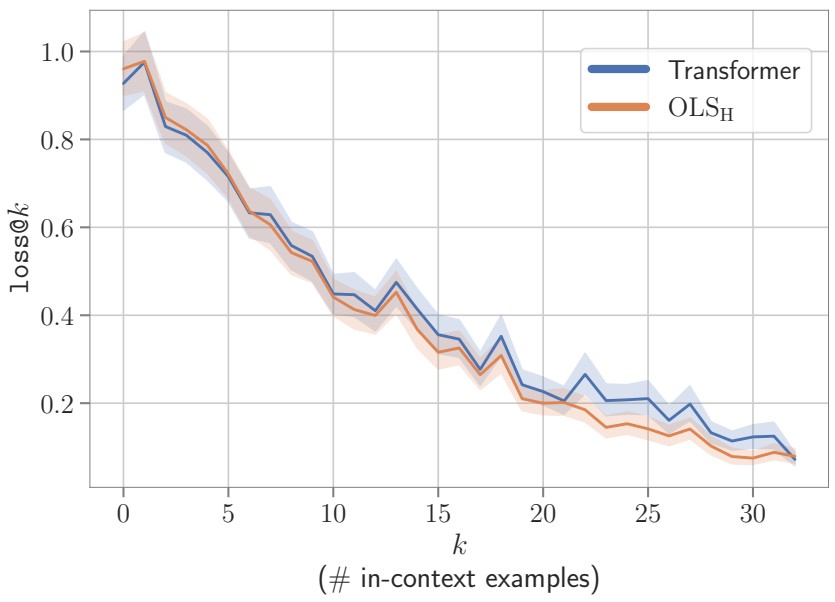

Figure 19: Evaluating Transformer trained on Haar Wavelet Basis Regression task ($\mathcal{F}_{\Phi_H}^{\mathrm{Haar}}$).

### B.2.4 Haar Wavelet Basis Regression

Similar to Fourier Series and Degree-2 Monomial Basis Regression, we also define another non-linear regression function family ($\mathcal{F}_{\Phi_H}^{\text{Haar}}$) using a different basis, $\Phi_H$, called the Haar wavelet basis. $\Phi_H$ is defined on the interval $[0, 1]$ and is given by:

$$\Phi_H(x) = \{x \in [0, 1] \mapsto \psi_{n,k}(x) : n \in \mathbb{N} \cup \{0\}, 0 \leq k < 2^n\} \cup \{\mathbf{1}\},$$
$$\psi_{n,k}(x) = 2^{n/2}\psi(2^n x - k), x \in [0, 1],$$
$$\psi(x) = \begin{cases} 1 & 0 \leq x < \frac{1}{2}, \\ -1 & \frac{1}{2} \leq x < 1, \\ 0 & \text{otherwise}, \end{cases}$$

where $\mathbf{1}$ is the constant function which is $1$ everywhere on $[0, 1]$. To define $f$, we sample $\boldsymbol{w}$ from $\mathcal{N}(0, 1)$ and compute its dot product with the basis, i.e. $\boldsymbol{w}^T \Phi_H(\cdot)$. We construct the prompt $P$ by evaluating $f$ at different values of $\boldsymbol{x} \sim \mathcal{U}(0, 1)$. The Transformer model is then trained on these prompts $P$.

We use $d = 1$ and $p = 32$, both of which are fixed throughout the training run, i.e. we do not use curriculum. We only consider the basis terms corresponding to $n \in \{0, 1, 2, 3\}$. The baseline used is OLS on Haar Wavelet Basis features (OLS$_H$). Note that for the model used throughout the paper (§2.2), at $k = 32$ the `loss@k` value is 0.18, while for a bigger model and OLS$_H$ it is 0.07. Therefore, for this task we report the results for the bigger model which has 24 layers, 16 heads and 512 hidden size.

**Results.** In Figure 19, we observe that Transformer very closely mimics the errors of OLS$_H$ (i.e. OLS fitted to the Haar Wavelet Basis) and converged to OLS$_H$ at $k = 32$. Since the prior on the weights $\boldsymbol{w}$ is Gaussian, OLS$_H$ is the PME. Hence, Transformer's performance on this task also suggests that it is simulating PME.

## C  Detailed Experiments for HMICL setup

### C.1  Why HMICL?

The distinction between MICL and HMICL is reminiscent of the distinction between the usual supervised learning and meta-learning. Consider linear regression as an example of the MICL setup, where all tasks are instances of linear regression and each weight vector defines a task. Further, consider a mixture of two "meta-tasks" or "function classes", say linear regression and decision trees. It is true that in an abstract sense, one could potentially consider each task to be defined by its parameters and thus ignore which type of "meta-task" it instantiates (linear regression or decision tree). Therefore, under this interpretation, MICL is equivalent to HMICL.

However, this view is too coarse-grained for our purposes. What is of interest, both from the application and theory perspectives, is a more fine-grained view about whether and how models learn to perform these different "meta-tasks". The hierarchical structure is central to this discussion. Previous work, e.g. Garg et al. (2022); Zhang et al. (2023), considers MICL with only a single "meta-task" and is thus not suitable for the type of analyses we perform. Compared to MICL, HMICL is arguably closer to the ICL for LLMs which can perform a vast variety of "meta-tasks." Is the training data of real-world LLMs hierarchical? Due to the complex nature of real-world training data distributions, it is hard to find concrete evidence of them being hierarchical, but we believe multi-task training in LMs like T5 Raffel et al. (2023) and FLAN-T5 Chung et al. (2022) (with a caveat for the latter being true for fine-tuning and not pre-training) or training of multilingual models Conneau et al. (2020) which involve pre-training corpora in different languages are some examples of the hierarchical nature of the training distribution in real-world LMs. To sum up, HMICL allows for a better terminology for investigating our models, with a potential for being related to real-world LLMs more closely than the vanilla MICL setting. Hence, we make the distinction between the two and treat them as separate settings in our work.

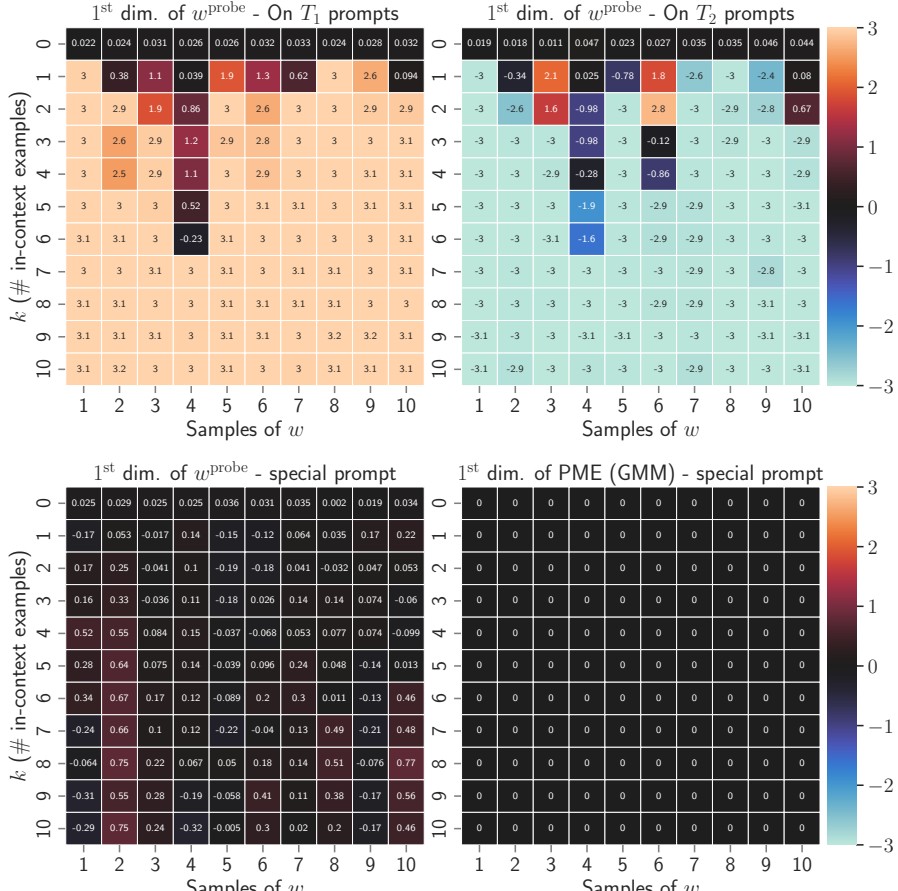

Figure 20: **Transformers simulate PME when trained on dense regression task-mixture ($d = 10, p = 10, \alpha_1 = \alpha_2 = \frac{1}{2}$) with weights having a mixture of Gaussian prior (GMM).** *(top)*: $1^{\text{st}}$ dimension of Transformer's probed weights across the prompt length. *(bottom)*: $1^{\text{st}}$ dimension of Transformer's probed weights and PME (GMM) across the prompt length for a specially constructed prompt.

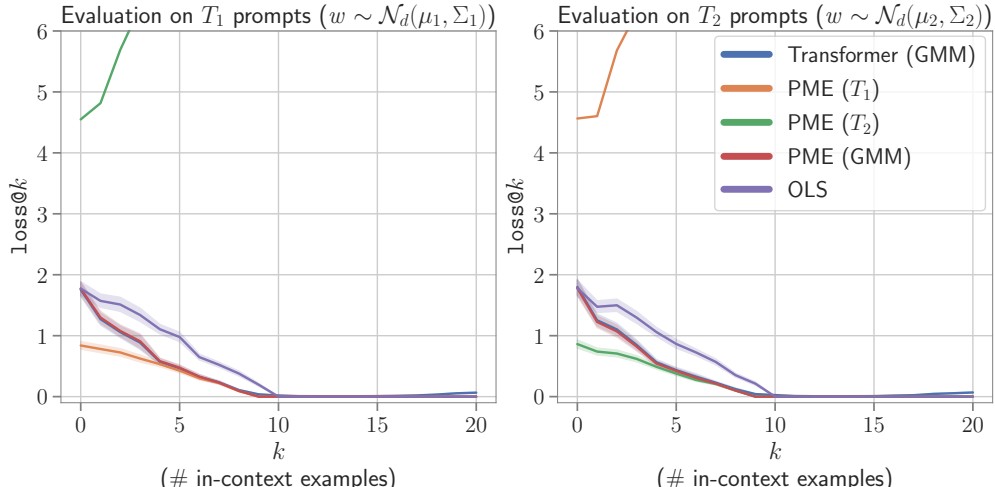

Figure 21: **Transformers simulate PME when trained on dense regression task-mixture ($d = 10, p = 10, \alpha_1 = \alpha_2 = \frac{1}{2}$) with weights having a mixture of Gaussian prior (GMM).** Comparing the performance of the Transformer, PMEs, and OLS in under- and over-determined regions. For all context lengths, the transformer follows PME(GMM) and is far from OLS in the under-determined region.

## C.2 GAUSSIAN MIXTURE MODELS (GMMS)

Here we discuss some details regarding §3.1 and more results on GMMs. We start with a description of how we calculate PMEs for this setup.

**Computation of PMEs.** As mentioned in §A.1 and §B.2, we can compute the individual PMEs for components $T_1$ and $T_2$ by minimizing the $L_2$ distance between the hyperplane induced by the prompt constraints and the mean of the Gaussian distribution. In particular, to compute PME for each Gaussian component of the prior, we solve a system of linear equations defined by the prompt constraints ($\boldsymbol{w}_i^T x_i = y_i, \forall i \in \{1, 2, .., p\}$) in conjunction with an additional constraint for the first coordinate, i.e. $(\boldsymbol{w})_1 = +3$ (for $\mathcal{N}_d(\boldsymbol{\mu}_1, \boldsymbol{\Sigma}_1)$ or $\boldsymbol{w}_1 = -3$ (for $\mathcal{N}_d(\boldsymbol{\mu}_2, \boldsymbol{\Sigma}_2)$). Given these individual PMEs, we calculate the PME of the mixture using Eq. 3.

Now we discuss more results for GMMs. First, we see the evolution of $\beta$'s (from Eq. 3), PME (GMM), and Transformer's probed weights across the prompt length (Figures 22 and 23). Next, we see the results for the Transformer models trained on the mixture with unequal weights, i.e. $\alpha_1 \neq \alpha_2$ (Figure 24) and for the $p = 20$ model (Figure 25).

**Agreement of weights between Transformer and PME(GMM). Figure 20 (top)** shows the evolution of the first dimension of the Transformer weights, i.e. $(\boldsymbol{w}^{\text{probe}})_1$, with prompt length $k$. We see that Transformer is simulating PME (GMM), which approaches PME ($T_{\text{prompt}}$) with increasing prompt length ($k$). Note that regardless of $k$, the first dimension of PME ($T_i$) is $(\boldsymbol{\mu}_i)_1$, the first dimension of the mean of the prior distribution $T_i$ since the Gaussian has a fixed value in the first dimension. Note that PME (GMM) approaches PME ($T_{\text{prompt}}$) with increasing $k$ (Eq. 3). Also note that in our setting, regardless of $k$ the first dimension of PME ($T_i$) is $(\boldsymbol{\mu}_i)_1$, the first dimension of the mean of the prior distribution $T_i$, since $T_i$ has a fixed value (i.e. zero variance) in the first dimension. Hence, if Transformer is simulating PME (GMM), the first dimension of Transformer's weights $(\boldsymbol{w}^{\text{probe}})_1$ must approach $(\boldsymbol{\mu}_1)_1$ (when $T_{\text{prompt}} = T_1$) and $(\boldsymbol{\mu}_2)_1$ (when $T_{\text{prompt}} = T_2$). This is exactly what we observe as $(\boldsymbol{w}^{\text{probe}})_1$ approaches $+3$ and $-3$ on $T_1$ and $T_2$ prompts respectively. At prompt length 0, in the absence of any information about the prompt, $(\boldsymbol{w}^{\text{probe}})_1 \approx 0$. This agrees with Eq. 3 since $0 = (\boldsymbol{\mu}_1)_1.\beta_1 + (\boldsymbol{\mu}_2)_1.\beta_2$, where $(\boldsymbol{\mu}_1)_1 = +3, (\boldsymbol{\mu}_2)_1 = -3, \beta_1 = \alpha_1 = 0.5$ and $\beta_2 = \alpha_2 = 0.5$ when prompt $P$ is empty. The figure shows that with the increasing evidence from the prompt, the transformer shifts its weights to $T_{\text{prompt}}$'s weights as evidenced by the first coordinate changing from 0 to $+3$ or $-3$ based on the prompt. In **Figure 20 (bottom)**, we check the behavior of Transformer and PME (GMM) on specially constructed prompts $P$ where $(\boldsymbol{x}_i)_1 = 0$

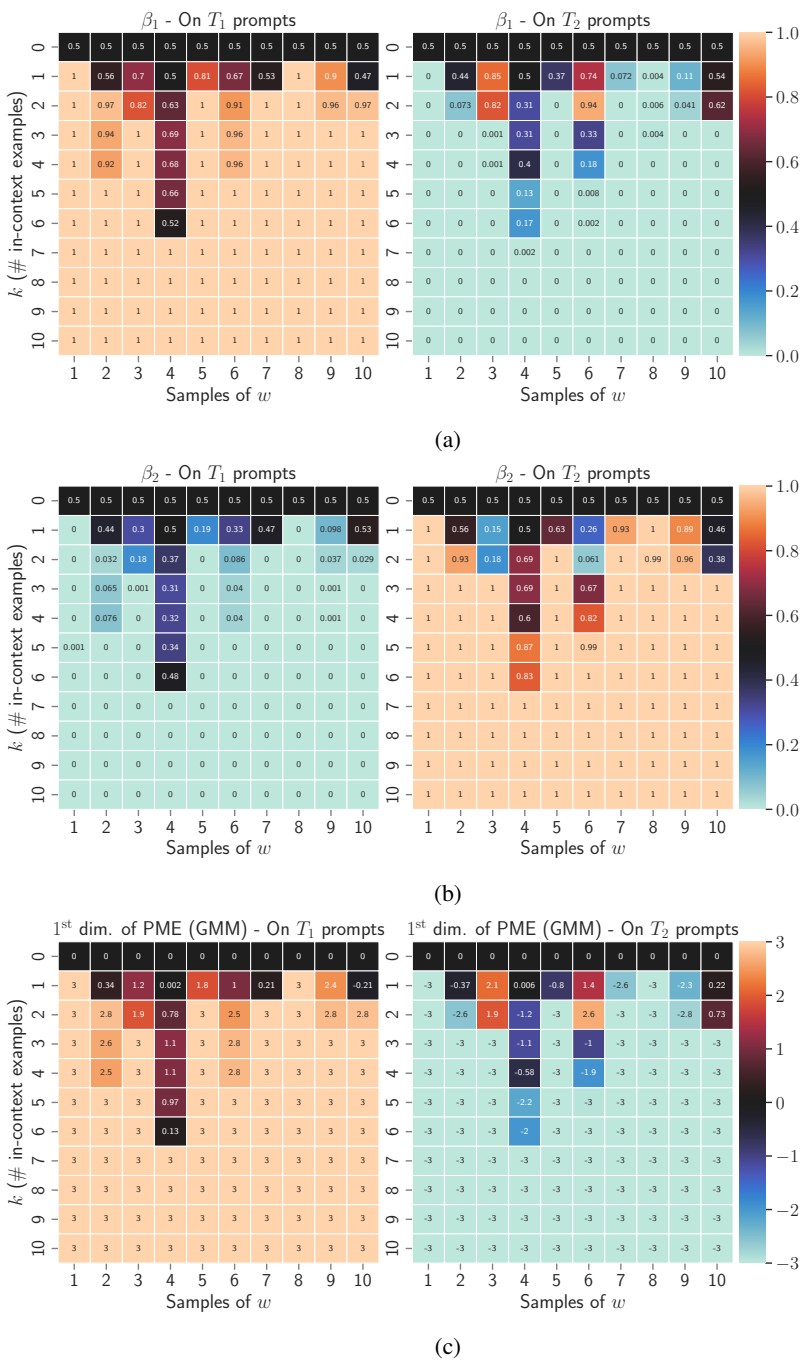

Figure 22: Evolution (as heatmaps) with prompt length ($k$) of $\beta$'s and PME (GMM) appearing in Eq. 3 for the model trained with $d = 10, p = 10, \alpha_1 = \alpha_2 = \frac{1}{2}$. We show 10 different samples of $\boldsymbol{w}$ for each plot.

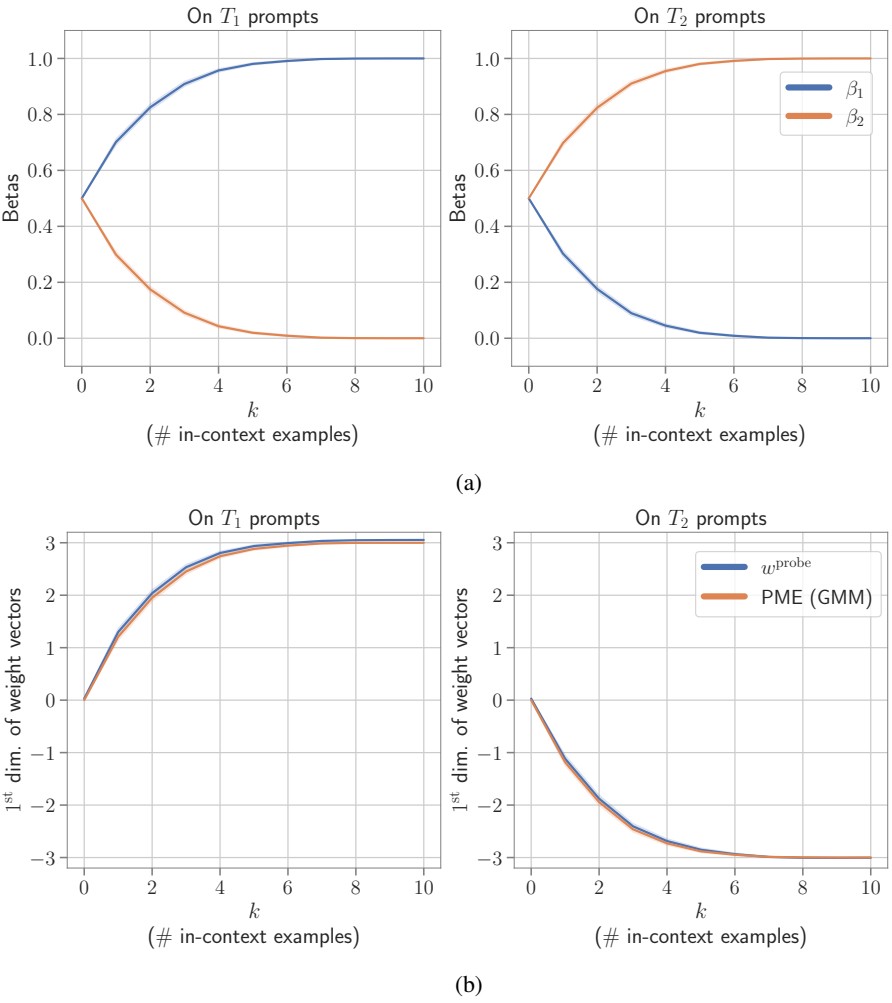

Figure 23: Evolution (as line plots) with prompt length ($k$) of $\beta$'s, PME (GMM), and $\boldsymbol{w}^{\mathrm{probe}}$ for the model trained with $d = 10, p = 10, \alpha_1 = \alpha_2 = \frac{1}{2}$. We show the values averaged over 1280 samples.

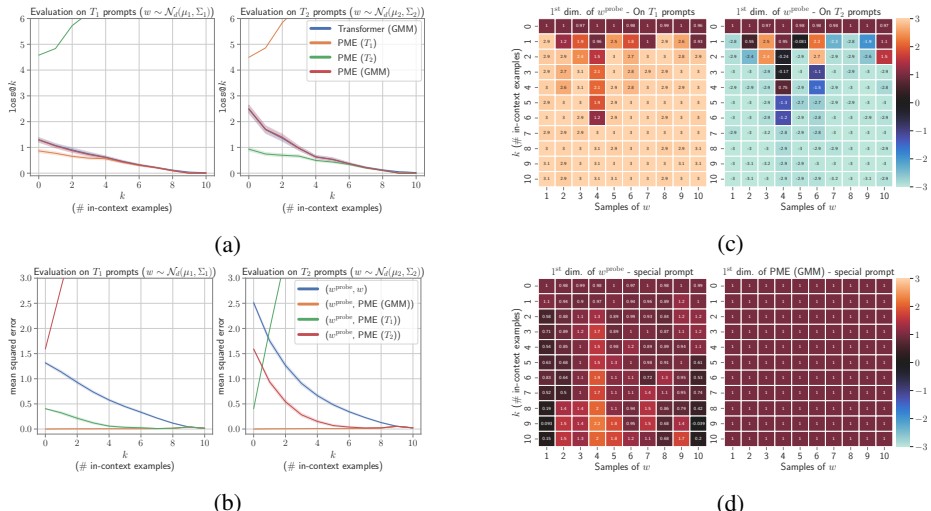

Figure 24: **Transformers simulate PME when trained on dense regression task-mixture ($d = 10, p = 10, \alpha_1 = \frac{2}{3}, \alpha_2 = \frac{1}{3}$) with weights having a mixture of Gaussian prior (GMM). (a)**: Comparing the performance of the Transformer with Posterior Mean Estimator (PME) of individual Gaussian components (PME ($T_1$) and PME ($T_2$)) and of the mixture PME (GMM). **(b)**: MSE between the probed weights of the Transformer and PMEs. **(c)**: 1st dimension of Transformer's probed weights across the prompt length. **(d)**: 1st dimension of Transformer's probed weights and PME (GMM) across the prompt length for a specially constructed prompt.

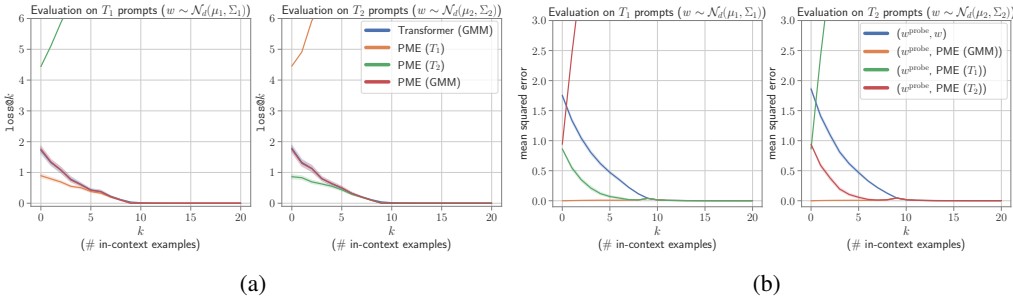

Figure 25: **Transformers simulate PME when trained on dense regression task-mixture ($d = 10, p = 20, \alpha_1 = \alpha_2 = \frac{1}{2}$) with weights having a mixture of Gaussian prior (GMM). Left:** Comparing the performance of the Transformer with Posterior Mean Estimator (PME) of individual Gaussian components (PME ($T_1$) and PME ($T_2$)) and of the mixture PME (GMM). **Right:** MSE between the probed weights of the Transformer and PMEs.

and $(\boldsymbol{x}_i)_{2:d} \sim \mathcal{N}(0, 1), \forall i \in \{1, \cdots, p\}$. For our setup, choosing such $\boldsymbol{x}_i$'s guarantees that no information about the distribution of $\boldsymbol{w}$ becomes known by observing $P$ (since the only distinguishing dimension between $T_1$ and $T_2$ is the 1st dimension and that does not influence the prompt in this case as $(\boldsymbol{x}_i)_1 = 0$). We note that Transformer's weights are all $\approx 0$ regardless of the prompt length, agreeing with the PME (GMM). Observing more examples from the prompt does not reveal any information about the underlying distribution of $\boldsymbol{w}$ in this case. Moreover, in **Figure 21** we plot the errors for the Transformer model, PMEs, and OLS for the over-determined region. In the over-determined case $(d > 10)$, the solution is unique, hence all the predictors including Transformer give the same solution and have errors $\approx 0$. Also, as shown, Transformer's errors are smaller than OLS errors and agree with the PME (GMM) errors for all context lengths. This shows that Transformer is indeed simulating the mixture PME and not OLS. All of this evidence strongly supports our hypothesis that Transformer behaves like the ideal learner and computes the Posterior Mean Estimate (PME).

**Evolution of $\beta$'s, PME (GMM), and $\boldsymbol{w}^{\mathrm{probe}}$.** Figure 22 plots the evolution of $\beta$'s and 1st dimension of PME (GMM) for 10 different $\boldsymbol{w}$'s. The $\beta$'s (Figures 22a and 22b) are $0.5$ (equal to $\alpha$'s) at $k = 0$ (when no information is observed from the prompt). Gradually, as more examples are observed from the prompt, $\beta_{T_{\mathrm{prompt}}}$ approaches $1$, while $\beta_{T_{\mathrm{other}}}$ approaches $0$. This is responsible for PME (GMM) converging to PME ($T_{\mathrm{prompt}}$) as seen in §3.1. The 1st dimension of PME (GMM) (Figure 22c) starts at $0$ and converges to $+3$ or $-3$ depending on whether $T_{\mathrm{prompt}}$ is $T_1$ or $T_2$. Figure 23 shows the same evolution in the form of line plots where we see the average across 1280 samples of $\boldsymbol{w}$. In Figure 23a, $\beta_{T_{\mathrm{prompt}}}$ approaches $1$, while $\beta_{T_{\mathrm{other}}}$ approaches $0$ as noted earlier. Consequently, in Figure 23b, 1st dimension of PME (GMM) approaches $+3$ or $-3$ based on the prompt. The 1st dimension of Transformer's probed weights, i.e. $(\boldsymbol{w}^{\mathrm{probe}})_1$ almost exactly mimics PME (GMM).

**Unequal weight mixture with $\alpha_1 = \frac{2}{3}$ & $\alpha_2 = \frac{1}{3}$.** Figure 24 shows the results for another model where $\alpha's$ are unequal ($d = 10, p = 10, \alpha_1 = \frac{2}{3}, \alpha_2 = \frac{1}{3}$). The observations made for Figure 1 in §3.1 still hold true, with some notable aspects: **(1)** The difference between prediction errors, i.e. $\texttt{loss@}k$ (24a), of PME (GMM) and PME ($T_1$) is smaller than that of the uniform mixture ($\alpha_1 = \alpha_2 = \frac{1}{2}$) case, while the difference between prediction errors and weights of PME (GMM) and PME ($T_2$) is larger. This is because, at prompt length $= 0$, PME (GMM) is a weighted combination of component PMEs with $\alpha$'s as coefficients (Eq. 3). Since $\alpha_1 > \alpha_2$, PME (GMM) starts out as being closer to $T_1$ than $T_2$. Also, since the Transformer follows PME (GMM) throughout, its prediction errors also have similar differences (as PME (GMM)'s) with PMEs of both components $T_1$ and $T_2$. **(2)** Transformer's probed weights ($\boldsymbol{w}^{\mathrm{probe}}$), which used to have the same MSE with PME ($T_1$) and PME ($T_2$) at $k = 0$, now give smaller MSE with PME ($T_1$) than PME ($T_2$) on prompts from both $T_1$ and $T_2$ (Figure 24b). This is a consequence of PME (GMM) starting out as being closer to $T_1$ than $T_2$ due to unequal mixture weights as discussed above. Since Transformer is simulating PME (GMM), $\boldsymbol{w}^{\mathrm{probe}}$ is also closer to PME ($T_1$) than PME ($T_2$) at $k = 0$ regardless of which component ($T_1$ or $T_2$) the prompts come from. Due to $\boldsymbol{w}^{\mathrm{probe}}$ mimicking $T_1$ more than $T_2$ we also observe in Figure 24b that $\boldsymbol{w}^{\mathrm{probe}}$ gives smaller MSE with $\boldsymbol{w}$ (ground truth) when $T_{\mathrm{prompt}} = T_1$ compared to when $T_{\mathrm{prompt}} = T_2$. **(3)** The 1st dimension of Transformer's weights $((\boldsymbol{w}^{\mathrm{probe}})_1)$ and PME (GMM) is $1$ instead of $0$ when the prompt is either empty (24c) or lacks information regarding the distribution of $\boldsymbol{w}$ (24d). It happens because $(\boldsymbol{w}^{\mathrm{probe}})_1 \approx$ 1st dimension of PME (GMM) $= (\boldsymbol{\mu}_1)_1 . \beta_1 + (\boldsymbol{\mu}_2)_1 . \beta_2 = (+3)(\frac{2}{3}) + (-3)(\frac{1}{3}) = 1$. Note that $\beta_1 = \alpha_1 = \frac{2}{3}$ and $\beta_2 = \alpha_2 = \frac{1}{3}$ when prompt $P$ is empty at $k = 0$ (Eq. 3). When $P$ is inconclusive of $\boldsymbol{w}$, $\beta_1 = \alpha_1$ and $\beta_2 = \alpha_2 \forall k \in \{1, 2, \cdots, p\}$.

**Transformer model trained with longer prompt length ($p = 20$).** Figure 25 depicts similar evidence as Figure 1 of Transformer simulating PME (GMM) for a model trained with $d = 10, p = 20, \alpha_1 = \alpha_2 = \frac{1}{2}$. We see that all the observations discussed in §3.1 also hold true for this model. Transformer converges to PME (GMM) and PME ($T_{\mathrm{prompt}}$) w.r.t. both $\texttt{loss@}k$ (Figure 25a) and weights (Figure 25b) at $k = 10$ and keeps following them for larger $k$ as well.

In summary, all the evidence strongly suggests that Transformer performs Bayesian Inference and computes PME corresponding to the task at hand. If the task is a mixture, Transformer simulates the PME of the task mixture as given by 3.

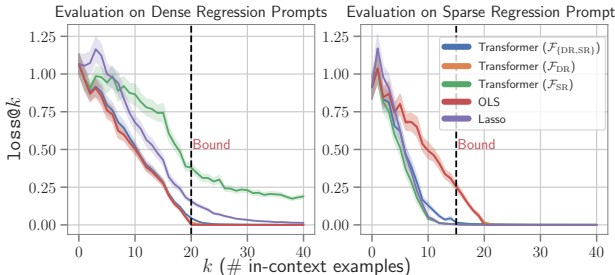

Figure 26: Comparing the performance of a Transformer model trained on dense and sparse regression mixture $\mathcal{F}_{\{DR, SR\}}$ with baselines, as well as single task models, trained on $\mathcal{F}_{DR}$ and $\mathcal{F}_{SR}$ individually.

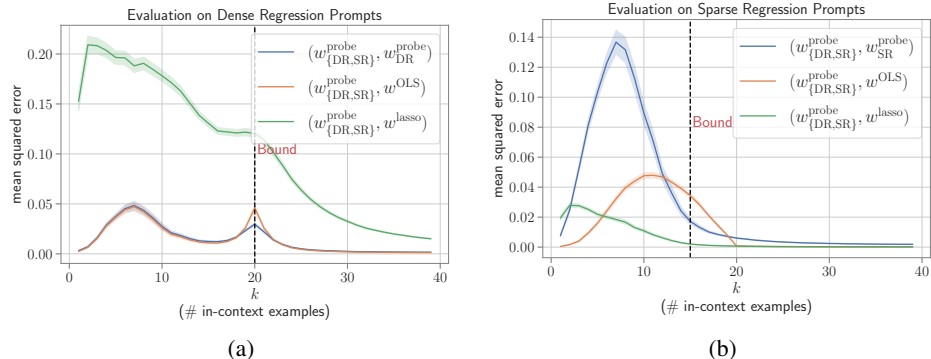

Figure 27: Comparing the errors between the weights recovered from the mixture model trained on $\mathcal{F}_{\{DR, SR\}}$ mixture and different single task models and baselines while evaluating on $\mathcal{F}_{DR}$ and $\mathcal{F}_{SR}$ prompts

### C.3 MORE COMPLEX MIXTURES

We start by training transformer models on the mixture of dense linear regression ($\mathcal{F}_{DR}$) and sparse linear regression ($\mathcal{F}_{SR}$) function classes. The function definition remains the same for both these classes i.e. $f : \boldsymbol{x} \mapsto \boldsymbol{w}_i^T \boldsymbol{x}$, but for $\mathcal{F}_{DR}$ we consider a standard gaussian prior on $w$ and a sparse prior for $\mathcal{F}_{SR}$. We use the sparse prior from Garg et al. (2022), where we first sample $\boldsymbol{w} \sim \mathcal{N}(\boldsymbol{0}_d, \boldsymbol{I})$ and then randomly set its $d - s$ components as 0. We consider $s = 3$ throughout our experiments. Unless specified we consider the mixtures to be uniform i.e. $\alpha_i = 0.5$ and use these values to sample batches during training.

During the evaluation, we test the mixture model (denoted as Transformer $\mathcal{F}_{\{DR, SR\}}$) on the prompts sampled from each of the function classes in the mixture. We consider the model to have in-context learned the mixture of tasks if it obtains similar performance as the single-task models specific to these function classes. For example, a transformer model trained on the dense and sparse regression mixture (Transformer $\mathcal{F}_{\{DR, SR\}}$) should obtain performance similar to the single-task model trained on dense regression function class (Transformer $\mathcal{F}_{DR}$), when prompted with a function $f \sim \mathcal{F}_{DR}$ and vice-versa.

**Results.** The results for the binary mixtures of linear functions are given in Figure 26. As can be observed, the transformer model trained on $\mathcal{F}_{\{DR, SR\}}$ obtains performance close to the OLS baseline as well as the transformer model specifically trained on the dense regression function class $\mathcal{F}_{DR}$ when evaluated with dense regression prompts. On the other hand, when evaluated with sparse regression prompts the same model follows Lasso and single-task sparse regression model (Transformer ($\mathcal{F}_{SR}$)) closely. As a check, note that the single-task models when prompted with functions from a family different from what they were trained on, observe much higher errors, confirming that the transformers learn to solve individual tasks based on the in-context examples provided. Similar to GMMs in §3.1, here also we compare the implied weights from multi-task models under prompts for both $\mathcal{F}_{DR}$

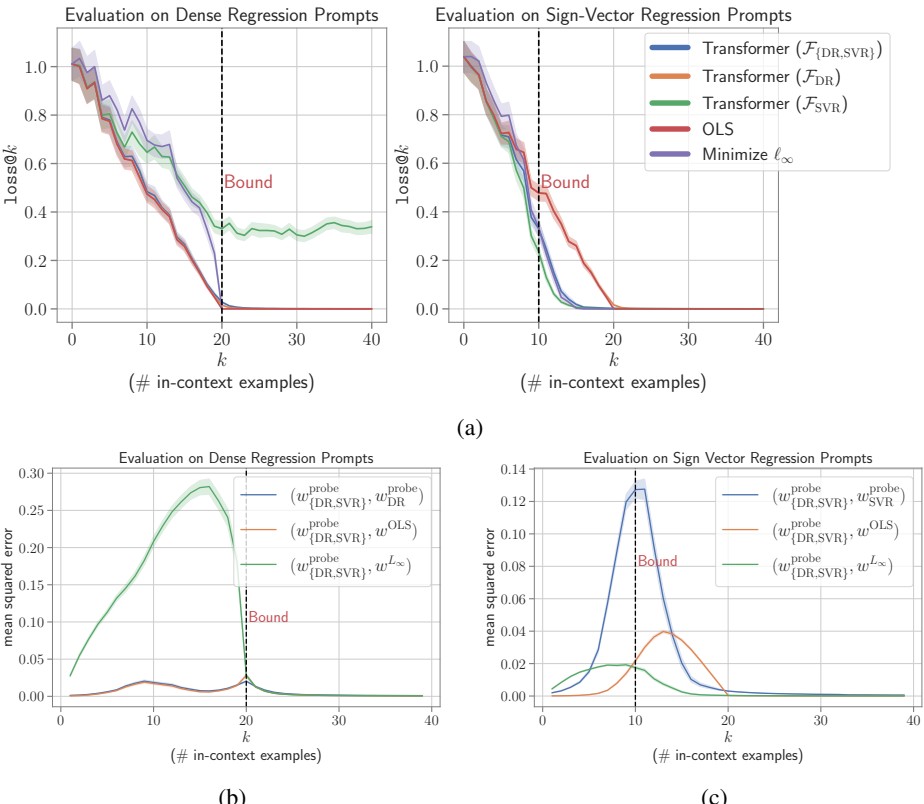

Figure 28: Comparing the performance of a Transformer model trained on dense and sign-vector regression mixture $\mathcal{F}_{\{\text{DR, SVR}\}}$ with baselines, as well as single task models, trained on $\mathcal{F}_{\text{DR}}$ and $\mathcal{F}_{\text{SVR}}$ individually. **Top**: Comparing loss@$k$ values of the mixture model with single-task models with different prompt distributions. **Bottom**: Comparing the errors between the weights recovered from the mixture model and different single task models and baselines while evaluating on $\mathcal{F}_{\text{DR}}$ and $\mathcal{F}_{\text{SVR}}$ prompts.

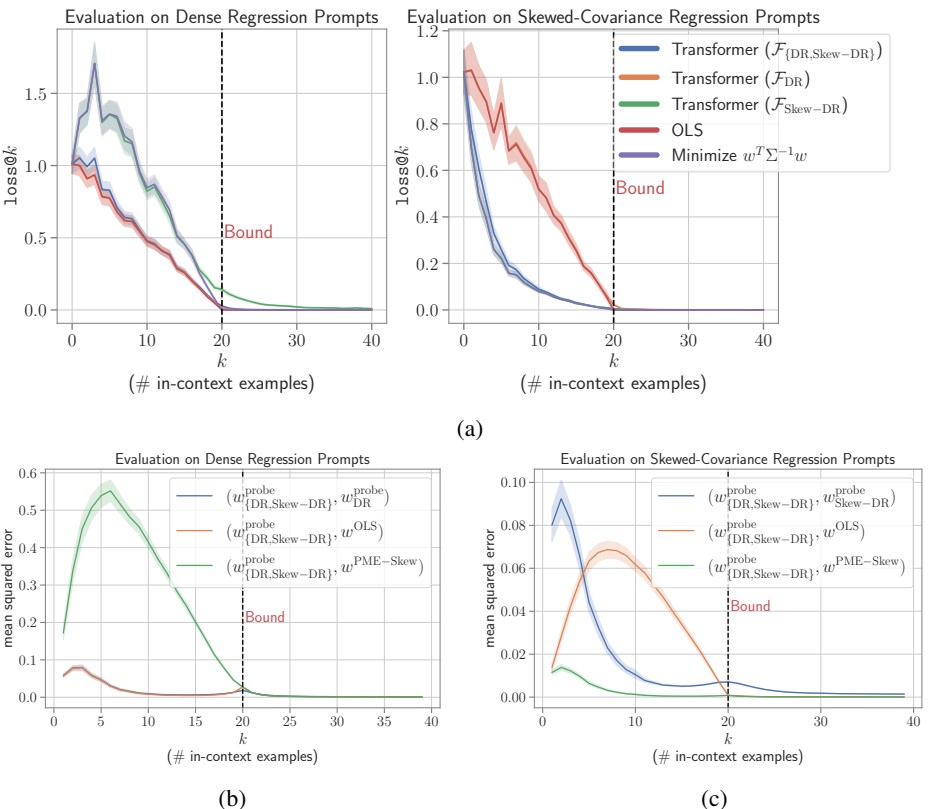

(a)

(b)                                                          (c)

Figure 29: Comparing the performance of a Transformer model trained on dense and skewed-covariance regression mixture $\mathcal{F}_{\{DR, Skew\text{-}DR\}}$ with baselines, as well as single task models, trained on $\mathcal{F}_{DR}$ and $\mathcal{F}_{Skew\text{-}DR}$ individually. **Top**: Comparing loss@$k$ values of the mixture model with single-task models with different prompt distributions. Red (OLS) and orange (Transformer ($\mathcal{F}_{DR}$)) curves overlap very closely, so are a bit hard to distinguish in the plots. Similarly in the top right plot, purple (Minimize $\boldsymbol{w}^T \boldsymbol{\Sigma}^{-1} \boldsymbol{w}$) and green (Transformer $\mathcal{F}_{Skew\text{-}DR}$) curves overlap. **Bottom**: Comparing the errors between the weights recovered from the mixture model and different single task models and baselines while evaluating on $\mathcal{F}_{DR}$ and $\mathcal{F}_{Skew\text{-}DR}$ prompts.

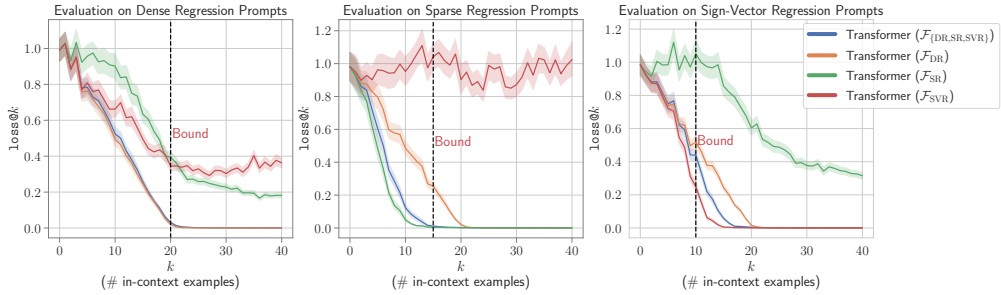

Figure 30: Comparing the performance of transformer model trained to in-context learn $\mathcal{F}_{\{DR, SR, SVR\}}$ mixture family with the corresponding single task models.

and $\mathcal{F}_{\text{SR}}$ and show that here again they agree with the weights recovered from single-task models as well as the strong baselines in this case (OLS and Lasso). We provide the plots for the weight agreement in this case in Figure 27.

Next, we describe the results for other homogeneous mixtures $\mathcal{F}_{\{\text{DR, SVR}\}}$, $\mathcal{F}_{\{\text{DR, Skew-DR}\}}$ and $\mathcal{F}_{\{\text{DR, SR, SVR}\}}$, as well as heterogeneous mixtures $\mathcal{F}_{\{\text{DR, DT}\}}$ and $\mathcal{F}_{\{\text{DT, NN}\}}$. As can be seen in Figure 28, the transformer model trained on $\mathcal{F}_{\{\text{DR, SVR}\}}$ mixture, behaves close to OLS when prompted with $f \in \mathcal{F}_{\text{DR}}$ and close to the $L_\infty$ minimization baseline when provided sign-vector regression prompts ($f \in \mathcal{F}_{\text{SVR}}$). We also have similar observations for the $\mathcal{F}_{\{\text{DR, Skew-DR}\}}$ mixture case in Figure 29, where the multi-task ICL model follows the PME of both tasks when sufficient examples are provided from the respective task. Similarly, for the model trained on the tertiary mixture $\mathcal{F}_{\{\text{DR, SR, SVR}\}}$ (as can be seen in Figure 30), the multi-task model can simulate the behavior of the three single-task models depending on the distribution of in-context examples. On $\mathcal{F}_{\text{SR}}$ and $\mathcal{F}_{\text{SVR}}$ prompts the multi-task model performs slightly worse compared to the single-task models trained on $\mathcal{F}_{\text{SR}}$ and $\mathcal{F}_{\text{SVR}}$ respectively, however once sufficient examples are provided (still $< 20$), they do obtain close errors. This observation is consistent with the PME hypothesis i.e. once more evidence is observed the $\beta$ values PME of the mixture should converge to the PME of the task from which prompt $P$ is sampled. The results on heterogeneous mixtures we discuss in detail below:

**Heterogeneous Mixtures**: Up until now, our experiments for the multi-task case have been focused on task mixtures where all function families have the same parameterized form i.e $w^T x$ for linear mixtures and $w^T \Phi(x)$ for Fourier mixtures. We now move to more complex mixtures where this no longer holds true. In particular, we consider dense regression and decision tree mixture $\mathcal{F}_{\{\text{DR, DT}\}}$ and decision tree and neural network mixture $\mathcal{F}_{\{\text{DT, NN}\}}$.

We follow Garg et al. (2022)'s setup for decision trees and neural networks. We consider decision trees of depth 4 and 20-dimensional input vectors $\boldsymbol{x}$. A decision tree is sampled by choosing the split node randomly from the features at each depth, and the output of the function is given by the values stored in the leaf nodes which are sampled from $\mathcal{N}(0, 1)$. For neural networks, we consider 2-layer (1 hidden + 1 output) multi-layer perceptrons (MLP) with ReLU non-linearity i.e. $f(x) = \sum_{i=1}^{r} \alpha_i \text{ReLU}(w_i^T \boldsymbol{x})$, where $\alpha \in \mathbb{R}$ and $\boldsymbol{w}_i \in \mathbb{R}^d$. The network parameters $a_i$s and $\boldsymbol{w}_i$s are sampled from $\mathcal{N}(0, 2/r)$ and $\mathcal{N}(0, 1)$ respectively. The input vectors $\boldsymbol{x}_i$s are sampled from $\mathcal{N}(0, 1)$ for both tasks. We consider greedy tree learning and stochastic gradient descent [5] over a 2-layer MLP as our baselines for decision trees and neural networks respectively. The values of hyperparameters for baselines such as the number of gradient descent steps, initial learning rate for Adam, etc. are the same as Garg et al. (2022).

The results for the two mixtures are provided in Figure 31. The mixture model Transformer ($\mathcal{F}_{\{\text{DR, DT}\}}$) follows the single task model Transformer ($\mathcal{F}_{\text{DR}}$) when provided in-context examples from $f \sim \mathcal{F}_{\text{DR}}$ and agrees with Transformer ($\mathcal{F}_{\text{DT}}$) when prompted with $f \sim \mathcal{F}_{\text{DT}}$ (Figure 31a. Similarly, we have consistent findings for $\mathcal{F}_{\{\text{DT, NN}\}}$ mixture as well, where the model learns to solve both tasks depending upon the input prompt (Figure 31b).

### C.4 FOURIER SERIES MIXTURE DETAILED RESULTS

We consider a mixture of Fourier series function classes with different maximum frequencies, i.e. $\mathcal{F}_{\Phi_{1:N}}^{\text{fourier}} = \{\mathcal{F}_{\Phi_1}^{\text{fourier}}, \cdots, \mathcal{F}_{\Phi_N}^{\text{fourier}}\}$. We consider $N = 10$ in our experiments and train the models using a uniform mixture with normalization. During evaluation, we test individually on each $\mathcal{F}_{\Phi_M}^{\text{fourier}}$, where $M \in [1, N]$. We compare against consider two baselines: i) OLS Fourier Basis $\mathcal{F}_{\Phi_M}^{\text{fourier}}$ i.e. performing OLS on the basis corresponding to the number of frequencies $M$ in the ground truth function, and ii) $\mathcal{F}_{\Phi_N}^{\text{fourier}}$ which performs OLS on the basis corresponding to the maximum number of frequencies in the mixture i.e. $N$.

Figure 32a plots the `loss@`$k$ metric aggregated over all the $M \in [1, N]$ for the model and the baselines. The performance of the transformer lies somewhere in between the gold-frequency baseline (OLS Fourier Basis $\mathcal{F}_{\Phi_M}^{\text{fourier}}$) and the maximum frequency baseline ($\mathcal{F}_{\Phi_N}^{\text{fourier}}$), with the model performing much better compared to the latter for short prompt lengths ($k < 20$) while the former baseline performs better. We also measure the frequencies exhibited by the functions predicted by the transformer in Figure 32b. We observe that transformers have a bias towards lower frequen-

---

[5]In practice, we use Adam just like Garg et al. (2022)

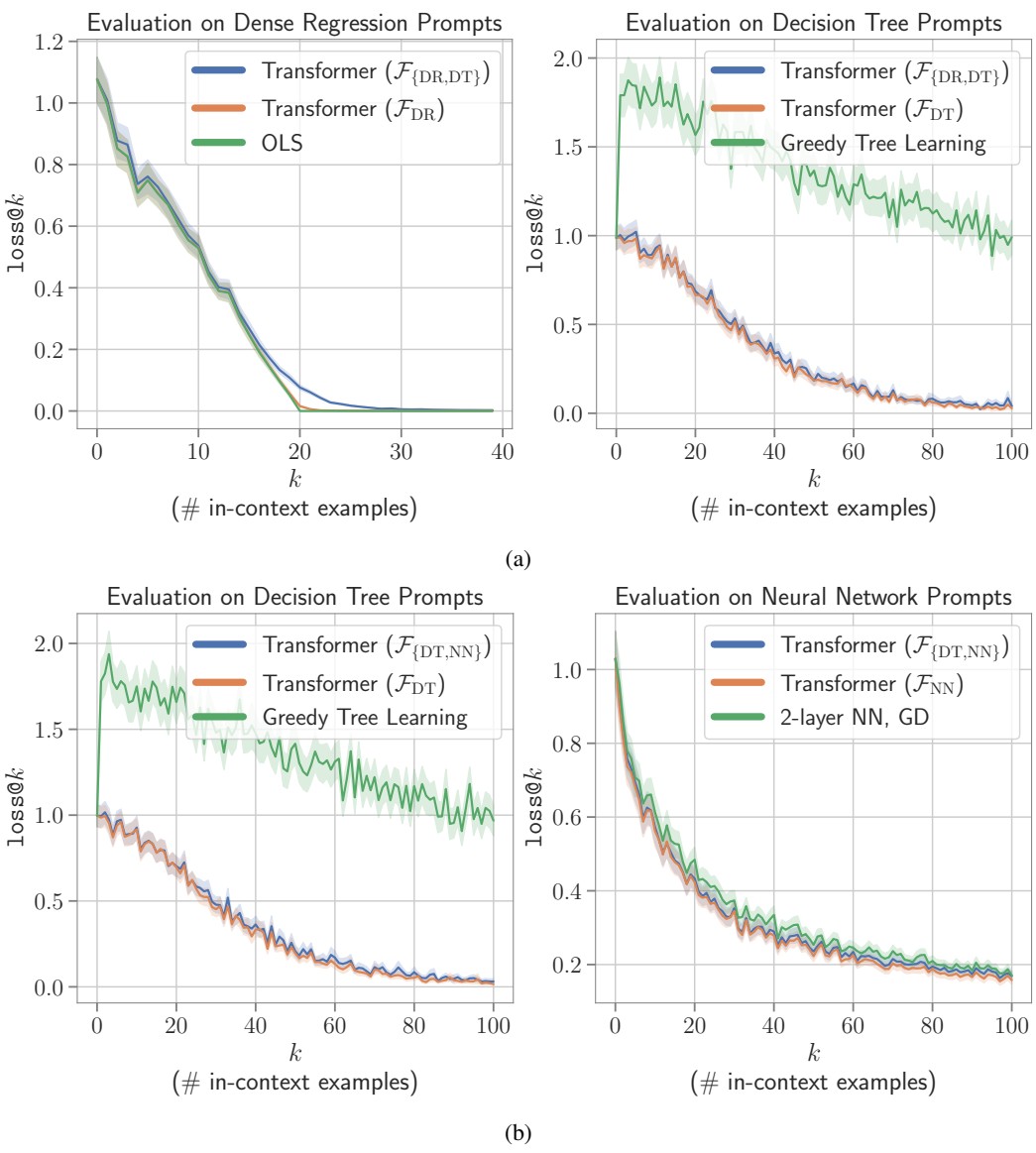

Figure 31: Multi-task in-context learning for heterogeneous mixtures.

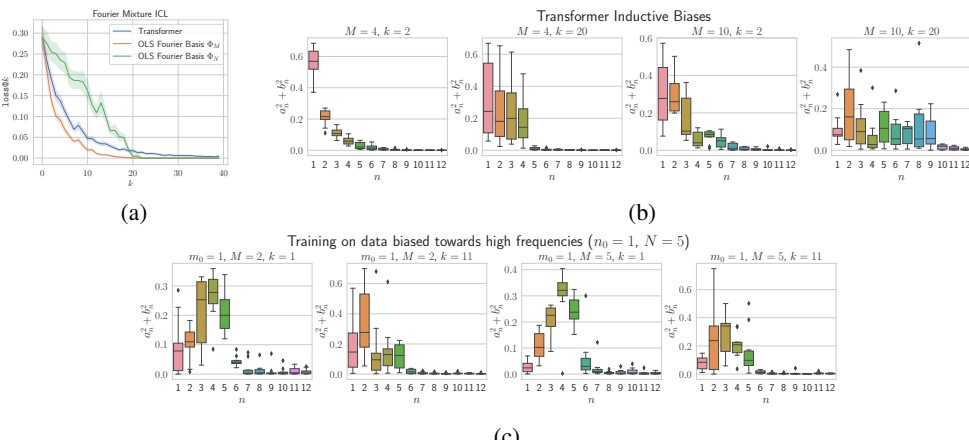

(a)                                                                    (b)

(c)

Figure 32: In-context learning on the Fourier series mixture class. **Top Left**: Comparing transformers with the baselines. Errors are computed on batches of 128 for $M \in [1, 10]$ and aggregated in the plot. **Top Right**: Visualizing the frequencies of the simulated function by transformers. **Bottom**: Training transformer on high-frequency biased Fourier mixture $\mathcal{F}_{\Phi_{1:N,N}^{fourier}}$ and visualizing the simulated frequencies of the trained model.

cies when prompted with a few examples; however, when given sufficiently many examples they are able to recover the gold frequencies. This simplicity bias can be traced to the training dataset for the mixture since lower frequencies are present in most of the functions of the mixture while higher frequencies will be more rare: Frequency 1 will be present in all the function classes whereas frequency $N$ will be present only in $\mathcal{F}_{\Phi_N}^{\text{fourier}}$. Our results indicate that the simplicity bias in these models during in-context learning arises from the training data distribution. We confirm the above observations by detailing results for different combinations of $M$ and $k$ in Figure 33.

### C.4.1   COMPLEXITY BIASED PRE-TRAINING

To further verify this observation, we also consider the case where the training data is biased towards high frequencies and check if transformers trained with such data exhibit bias towards high frequencies (complexity bias). To motivate such a mixture, we first define an alternate fourier basis: $\Phi_{n_0,N}(x) = [\cos(n_0\pi/L), \sin(n_0\pi/L), \cos((n_0+1)\pi/L), \sin((n_0+1)\pi/L), \cdots, \cos(N\pi/L), \sin(N\pi/L)]$, where $n_0 \geq 0$ is the minimum frequency in the basis. $\Phi_{n_0,N}$ defines the function family $\mathcal{F}_{\Phi_{n_0,N}}^{\text{fourier}}$ and equivalently we can define the mixture of such function classes as $\mathcal{F}_{\Phi_{1:N,N}^{\text{fourier}}} = \{\mathcal{F}_{\Phi_{1,N}}^{\text{fourier}}, \cdots, \mathcal{F}_{\Phi_{N,N}}^{\text{fourier}}\}$. One can see such a mixture will be biased towards high frequency; frequency $N$ is present in each function class of the mixture, while frequency 1 is only present in $\mathcal{F}_{\Phi_{1,N}}^{\text{fourier}}$. We train a transformer model on such a mixture for $N = 5$ and at test time, we evaluate the model on functions $f \sim \mathcal{F}_{\Phi_{m_0,M}}^{\text{fourier}}$ Figure 32c shows the inductive biases measure from this trained model and we can clearly observe a case of complexity bias, where at small prompt lengths, the model exhibited a strong bias towards the higher end of the frequencies that it was trained on i.e. close to 5.

We also trained models for higher values of the maximum frequency i.e. $N = 10$ for the high-frequency bias case, but interestingly observed the model failed to learn this task mixture. Even for $N = 5$, we noticed that the convergence was much slower compared to training on the simplicity bias mixture $\mathcal{F}_{\Phi_{1:N}}^{\text{fourier}}$. This indicates, while in this case, the origin of simplicity bias comes from the training data, it is harder for the model to learn to capture more complex training distributions, and simplicity bias in the pre-training data distribution might lead to more efficient training Mueller & Linzen (2023).

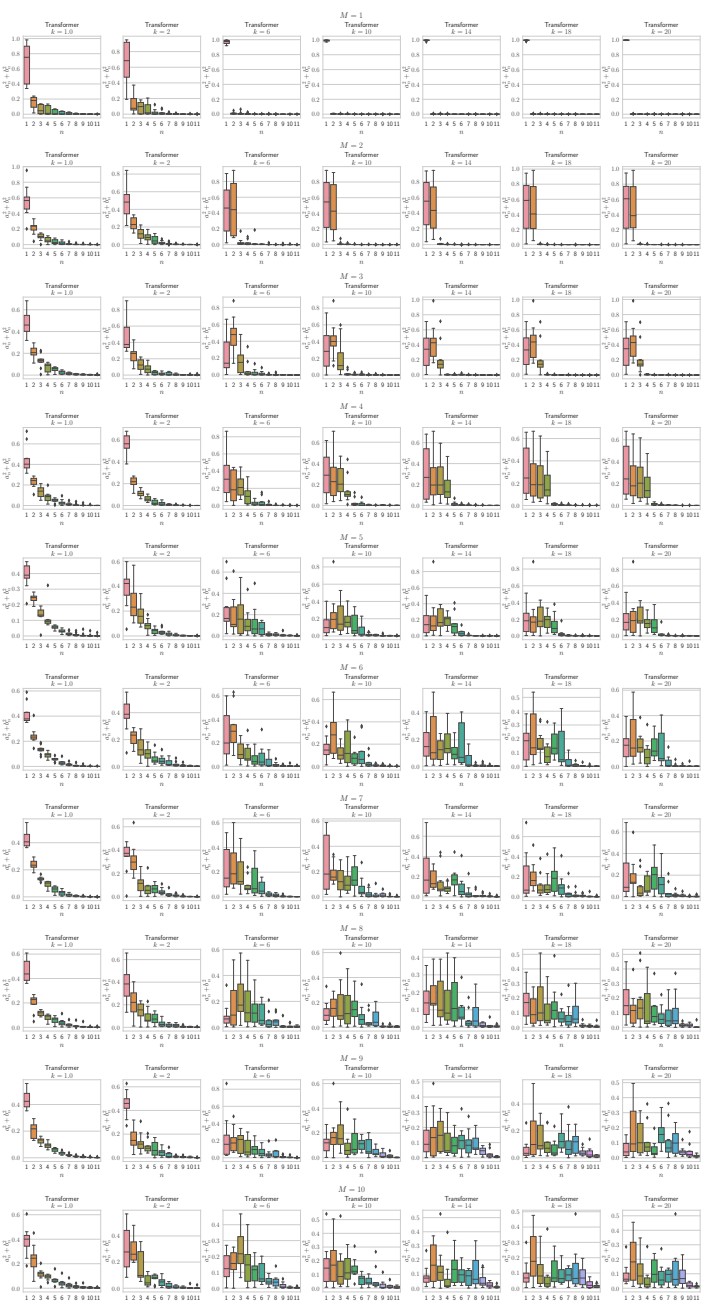

Figure 33: In-context learning of Fourier series mixture class. Measuring the frequencies of the simulated function by the transformer for different values of $M$ (maximum frequency) and $k$ (number of in-context examples). Showcases the simplicity bias behavior exhibited by the model at low frequencies.

## C.5   CONDITIONS NECESSARY FOR MULTI-TASK ICL

We observed that the training setup can also influence the ability of transformers to simulate the Bayesian predictor during ICL. Particularly, in our initial experiments with $\mathcal{F}_{\{DR, SR\}}$ mixture (§C.3), transformers failed to learn to solve the individual tasks of the mixture and were following OLS for both $\mathcal{F}_{DR}$ and $\mathcal{F}_{SR}$ prompts. To probe this, we first noted that the variance of the function outputs varied greatly for the two tasks, where for dense regression it equals $d$ and equals the sparsity parameter $s$ for sparse regression. We hypothesized that the model learning to solve just dense regression might be attributed to the disproportionately high signal from dense regression compared to sparse. To resolve this, we experimented with increasing the sampling rate for the $\mathcal{F}_{SR}$ task family during training. Particularly on training the model with $\alpha_{SR} = 0.87$, we observed that the resulting model did learn to solve both tasks. Alternatively, normalizing the outputs of the two tasks such that they have the same variance and using a uniform mixture ($\alpha_{SR} = 0.5$) also resulted in multi-task in-context learning capabilities (also the setting of our experiments in Figure 26). Hence, the training distribution can have a significant role to play in the model acquiring abilities to solve different tasks as has been also observed in other works on in-context learning in LLMs Razeghi et al. (2022); Chan et al. (2022a).

We also studied if the curriculum had any role to play in the models acquiring multi-task in-context learning capabilities. In our initial experiments without normalization and non-uniform mixtures, we observed that the model only learned to solve both tasks when the curriculum was enabled. However, training the model without curriculum for a longer duration ($\approx$ more training data), we did observe it to eventually learn to solve both of the tasks indicated by a sharp dip in the evaluation loss for the sparse regression task during training. This is also in line with recent works Hoffmann et al. (2022); Touvron et al. (2023), which show that the capabilities of LLMs can be drastically improved by scaling up the number of tokens the models are trained on. Detailed results concerning these findings are in Figure 34 of the Appendix.

Figure 34 compares transformer models trained on $\mathcal{F}_{\{DR, SR\}}$ mixture with different setups i.e. training without task-normalization and uniform mixture weights $\alpha_i$'s (Figure 34a), training without task-normalization and non-uniform mixture weights (Figure 34b), and training with task normalization and uniform mixture weights (Figure 34c). As described above, we perform task normalization by ensuring that the outputs $f(x)$ for all the tasks have the same variance, which results in all the tasks providing a similar training signal to the model. To perform normalization, we simply divide the weights $w$ sampled for the tasks by a normalization constant, which is decided according to the nature of the task. With this, we make sure that the output $y = w^T x$ has a unit variance. The normalization constants for different tasks are provided in Table 2.

Table 2: Normalization constants used for different tasks to define normalized mixtures for multi-task ICL experiments. Here $d$ denotes the size of the weight vectors used in linear-inverse problems as well as the last layer of the neural network. $s$ refers to the sparsity of sparse regression problems, $r$ is the hidden size of the neural network and $N$ refers to the maximum frequency for Fourier series.

| Function Family | Normalization Constant |
|---|---|
| Dense Regression | $\sqrt{d}$ |
| Sparse Regression | $\sqrt{s}$ |
| Sign-Vector Regression | $\sqrt{d}$ |
| Fourier-Series | $N$ |
| Degree-2 Monomial Basis Regression | $\sqrt{|\mathcal{S}|}$ |
| Decision Trees | 1 |
| Neural Networks | $\sqrt{\frac{dr}{2}}$ |

All the experiments discussed above (like most others in the main paper) were performed using curriculum learning. As discussed above, we investigated if the curriculum has any effect on multi-task ICL capabilities. The results for the same are provided in Figure 35.

We also explore the effect of normalization on multi-task ICL in Figure 36 for $\mathcal{F}_{\{DR, SVR\}}$ task. As can be seen in Figure 36a, for this particular mixture even while training the model without normalization, the model exhibited multi-task ICL, which can be explained by both tasks having

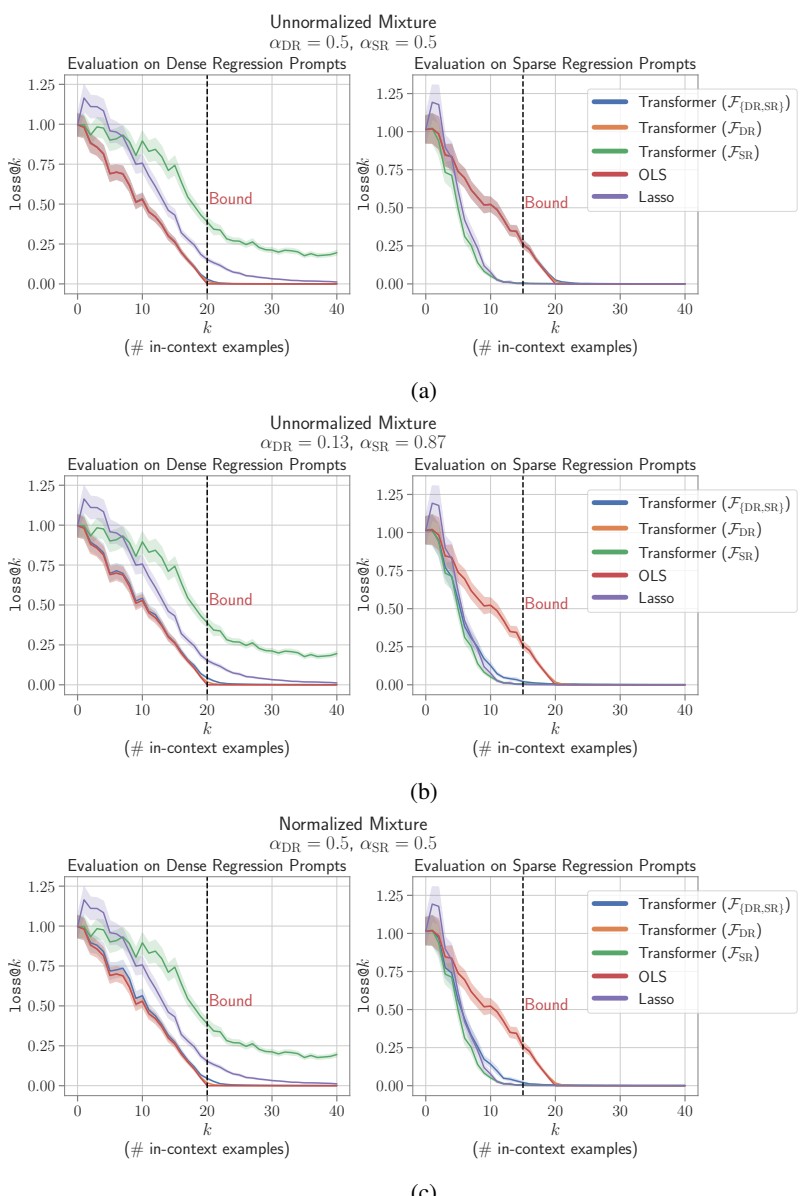

Figure 34: Conditions affecting multi-task ICL in transformers. **Top**: Evaluating `loss@`$k$ for transformer model trained on $\mathcal{F}_{\{DR, SR\}}$ task family without normalization and considering uniform mixtures (i.e. $\alpha_{DR} = \alpha_{SR} = 0.5$), and comparing with single-task models and baselines. While the blue curve (Transformer $\mathcal{F}_{\{DR, SR\}}$) is hard to see here, it is because it overlaps almost perfectly with the red curve corresponding to OLS in both cases.**Center**: Similar plots as above but for the model trained on the mixture $\mathcal{F}_{\{DR, SR\}}$ with non-uniform weights i.e. $\alpha_{DR} = 0.13, \alpha_{SR} = 0.87$. **Bottom**: Training the model with the normalized (and uniform) mixture such that outputs for the two tasks have the same variance. All the models are **trained with the curriculum**. The discussion continues in Figure 35 for the models trained without curriculum.

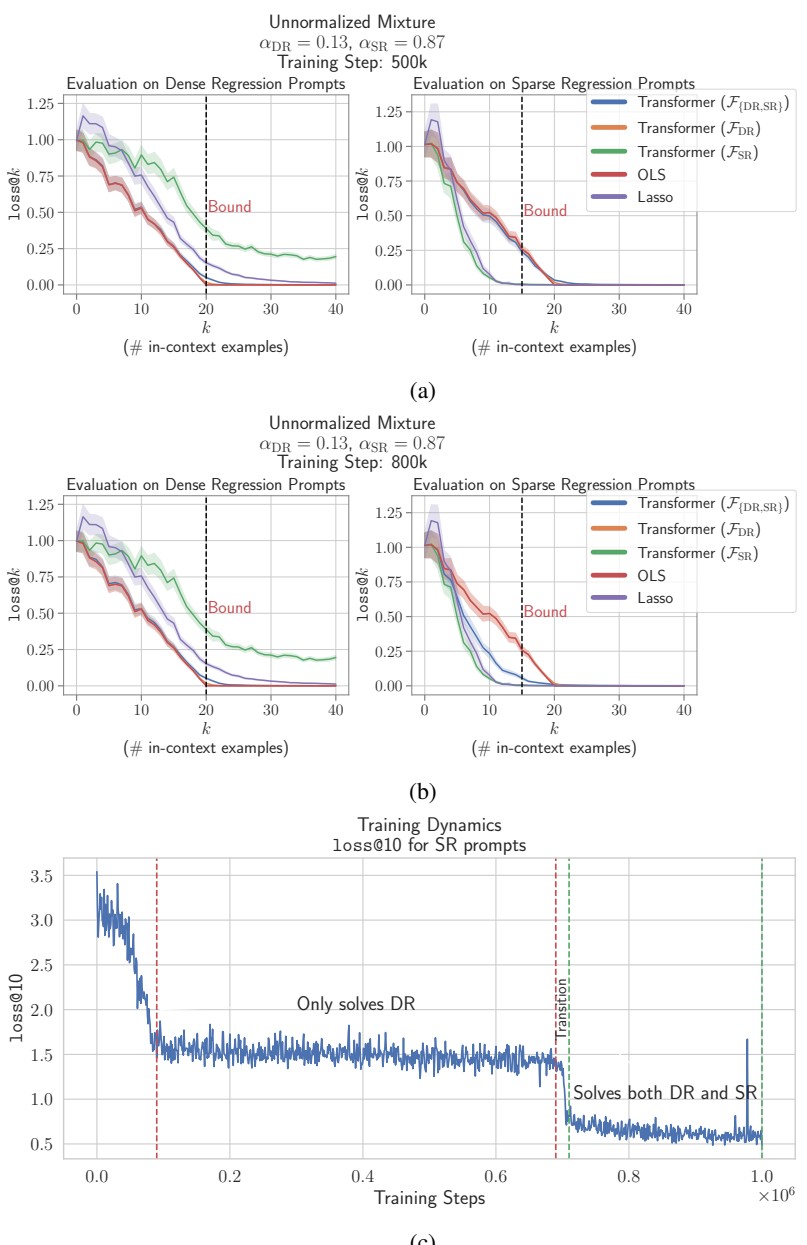

Figure 35: Evaluating transformer model trained **without curriculum** on $\mathcal{F}_{\{DR, SR\}}$ task family without normalization and non-uniform weights i.e. $\alpha_{DR} = 0.13, \alpha_{SR} = 0.87$ (similar to Figure 34b). **Top**: Evaluating the checkpoint corresponding to the 500k training step of the aforementioned model. Again, the blue curve (Transformer $\mathcal{F}_{\{DR, SR\}}$) is hard to see here, but it is because it overlaps almost perfectly with the red curve corresponding to OLS in both cases. **Center**: Evaluating the same model but a much later checkpoint i.e. at 800k training step. **Bottom**: Evolution of `loss@10` on $\mathcal{F}_{SR}$ prompts while training the above model.

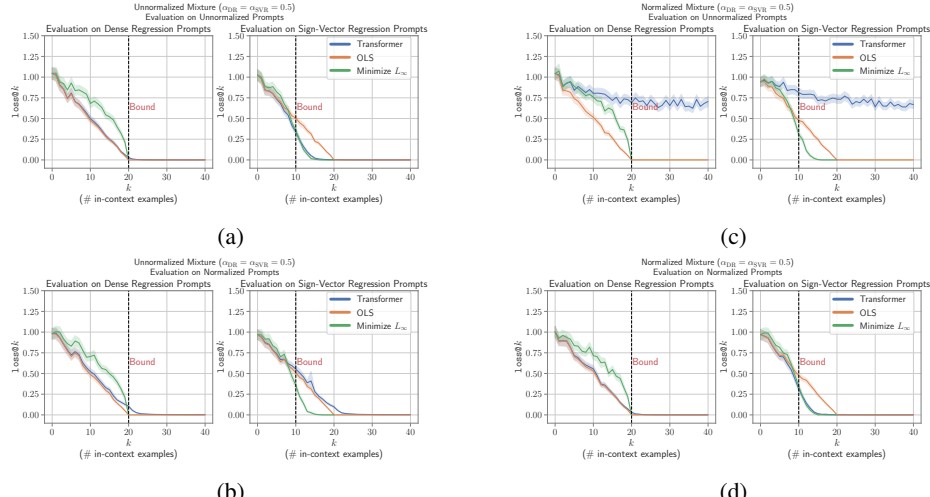

(a)            (c)

(b)            (d)

Figure 36: **Effect of output normalization on multi-task ICL in transformers. Top Left (a)**: A transformer model is trained on a uniform mixture of $\mathcal{F}_{\{DR, SVR\}}$ task family (i.e. $\alpha_{DR} = \alpha_{SVR} = 0.5$) without normalization. Evaluating $\texttt{loss@}k$ for this model on unnormalized prompts (where outputs $f(x)$ are not normalized to have unit variance i.e. same as training). Note that for the $\mathcal{F}_{\{DR, SVR\}}$ task family even without normalization the outputs $f(x)$ have the same mean and variance ($\mu = 0, \sigma^2 = 20$) for both the tasks. **Bottom Left (b)**: Evaluating $\texttt{loss@}k$ for the model in (a) on normalized prompts (where outputs $f(x)$ for both tasks are normalized to have unit variance). **Top Right (c)**: A transformer model is trained on a uniform mixture of $\mathcal{F}_{\{DR, SVR\}}$ task family (i.e. $\alpha_{DR} = \alpha_{SVR} = 0.5$) with normalization. Evaluating $\texttt{loss@}k$ for this model on unnormalized prompts. **Bottom Right (d)** Evaluating $\texttt{loss@}k$ for the model in (c) on normalized prompts. All the models are trained with the curriculum.

the same output variance (i.e. $d$). Interestingly, when we evaluate this model (i.e. the one trained on unnormalized mixture) on in-context examples which have the outputs $f(x_i)$'s normalized, the model fails to solve $\mathcal{F}_{SVR}$ and follows OLS baseline for both the tasks. We hypothesize that since this situation represents Out of Distribution (OOD) evaluation and the model might not be robust towards performing multi-task ICL on prompts that come from a different distribution than those seen during training. Exploring OOD generalization in the multi-task case is a compelling direction that we leave for future work.

Table 3: **Multi-task generalization results for Monomials problem**. The first row is ID evaluation, second row is OOD evaluation. As task diversity ($K$) increases, the model starts behaving like $\text{Lasso}_{\Phi_M}$ and $\text{BP}_{\text{proxy}}$, and its ID and OOD losses become almost identical, i.e. it generalizes to OOD.

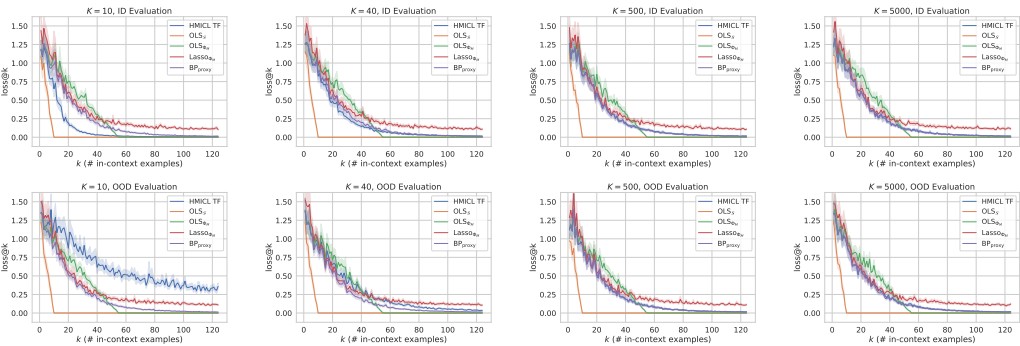

Table 4: **Multi-task generalization results for Fourier Series problem**. The first row is ID evaluation, second row is OOD evaluation. As task diversity $(K)$ increases, the model starts behaving like $\text{Lasso}_{\Phi_N}$ and $\text{BP}_{\text{proxy}}$, and its ID and OOD losses become almost identical, i.e. it generalizes to OOD.

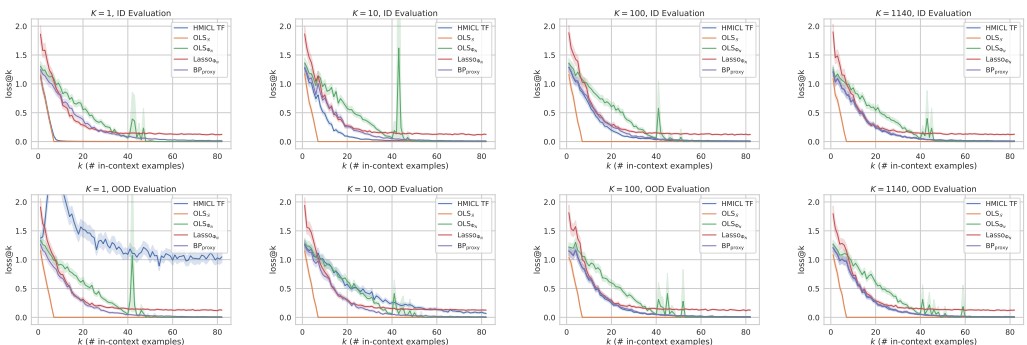

# D   DETAILS REGARDING MULTI-TASK GENERALIZATION EXPERIMENTS

Like section §B.1.1, we tune the Lasso coefficient on a separate batch of data (1280 samples) and choose the single value that achieves the smallest loss.

## D.1   MONOMIALS MULTI-TASK

Plots for various task diversities that we experiment with are shown in Table 3.

## D.2   FOURIER SERIES MULTI-TASK

**Fourier Series problem – MICL setting.**   Please refer to the setup defined in §B.2.1, which comes under the MICL setting as it corresponds to a single function class, $\mathcal{F}_{\Phi_N}^{\text{fourier}}$.

**Extending Fourier Series problem to HMICL setting.**   For extension to HMICL, we use multiple subsets of frequencies $\mathcal{S}_k$'s to define the mixture. Each $\mathcal{S}_k$ defines a function class $\mathcal{F}_{\mathcal{S}_k}^{\text{fourier}}$. The pretraining distribution is induced by the uniform distribution $\mathcal{U}(\mathcal{F})$ over a collection of such function classes, $\mathcal{F} = \{\mathcal{F}_{\mathcal{S}_1}^{\text{fourier}}, \cdots, \mathcal{F}_{\mathcal{S}_K}^{\text{fourier}}\}$, where $\mathcal{S}_k \subset \Phi_N(x)$, the full basis. For example, $\mathcal{S}_k$ could be $[1, \cos(2\pi x/L), \cos(6\pi x/L), \cos(9\pi x/L), \sin(2\pi x/L), \sin(6\pi x/L), \sin(9\pi x/L)]^T$, consisting of sine and cosine frequencies corresponding to integers $2, 6$ and $9$. (Note that $1$, the intercept term, is a part of every $\mathcal{S}_k$). $K$ feature sets $\mathcal{S}_k$'s, each of size $D$, are chosen at the start of the training and remain fixed. $K$ is the task diversity of the pretraining distribution. To sample a training function for the TF, we first sample a function class $\mathcal{F}_{\mathcal{S}_k}^{\text{fourier}}$ with replacement from $\mathcal{U}(\mathcal{F})$ and then sample a function from the chosen class; $f(x) = w^T \mathcal{S}_k(x)$, where $w \sim \mathcal{N}_D(\mathbf{0}, \mathbf{I})$. Similar to the Monomials problem, our aim is to check if TF trained on $\mathcal{U}(\mathcal{F})$ can generalize to the full distribution of all function classes (for feature sets of size $D$) by evaluating its performance on function classes corresponding to feature sets $\mathcal{S}' \notin \{\mathcal{S}_1, \cdots, \mathcal{S}_K\}$.

**Training Setup.**   $d = 1, p = 82, N = 20$. So, the full basis, $\Phi_N(x)$, had 20 frequencies. $D = 3$. We experiment with $K \in \{1, 10, 100, 200, 400, 800, 1140\}$.

**Evaluation Setup.**   The baselines we consider are $\text{OLS}_\mathcal{S}$, $\text{OLS}_{\Phi_N}$, and $\text{Lasso}_{\Phi_N}$ For Lasso, we use $\alpha = 0.1$. We again evaluate in two settings: **(a) ID**: On functions from the pretraining distribution. **(b) OOD**: On functions not in the pretraining distribution by sampling from a function family not corresponding to any of the $\mathcal{S}'_k s$ used to define the pretraining distribution.

**Results.**   Plots for various task diversities that we experiment with are in Table 4. The trend is the same as it was for Monomials problem, i.e. ID performance degrades and OOD performance

improves as $K$ increases. As $K$ increases, TF's performance on ID and OOD becomes identical (from $K = 100$ onwards) and similar to the $\text{Lasso}_{\Phi_N}$ and $\text{BP}_{\text{proxy}}$ baselines.

### D.3 DETAILS ON THE PHENOMENON OF FORGETTING

**Problem Setup.** We follow the Noisy Linear Regression (NLR) setup from Raventós et al. (2023): $d = 8, p = 15$ (without curriculum learning). The noise variance $\sigma^2 = 0.25$. For this problem, the transformer has 8 layers, 128-dimensional embeddings, and 2 attention heads, and is trained with a batch size of 256 for 500k steps. One-cycle triangle learning rate schedule Smith & Topin (2018) is used with 50% warmup. Detailed plots for the four groups of task diversities mentioned in §6.1 are in Figure 37. OOD loss curves with Bayesian predictors for various checkpoints for task div $2^8$ are in Figure 38. For other representative task diversities, the OOD loss curves of TF and Bayesian predictors are in Table 5. Plots showing mean squared errors of implied weights of TF with Bayesian predictors are in Table 6.

**Classification of task diversities.** ID loss $\approx 0$ for all task diversities during pretraining. We group them into the following 4 categories based on OOD loss:

**1. $2^1$ to $2^3$** (no generalization; no forgetting) – OOD loss never decreases, converges to a value worse than or same as at the start of the training ($t_0$), agrees with dMMSE at the end of the training ($t_{end}$). [Figure 37a]

**2. $2^4$ to $2^6$** (some generalization and forgetting) – OOD loss improves, reaches a minima $t_{min}$, then worsens. OOD loss is worse than ID loss throughout pretraining and agrees with dMMSE at $t_{end}$ (i.e., any generalization to Gaussian distribution is forgotten by $t_{end}$). [Figure 37b]

**3. $2^7$ to $2^{11}$** (full generalization and forgetting) – OOD loss improves, reaches a minima $t_{min}$, at which it is same as ID loss, then it worsens. At $t_{min}$, OOD loss agrees with Ridge (Figure 38a), then gradually deviates from it and at $t_{end}$, it is in between dMMSE and Ridge (e.g., Figure 38c). We refer to this group of task diversities as the "Gaussian forgetting region" since **the model generalizes to the full (Gaussian) distribution over tasks at $\mathbf{t_{min}}$ but forgets it by $\mathbf{t_{end}}$.** [Figures 37c, 38]

**4. $2^{12}$ to $2^{20}$** (full generalization; no forgetting) – Throughout pretraining, OOD and ID losses are identical and OOD loss agrees with Ridge. [Figure 37d]

**Relation to Simplicity bias?** The phenomenon of forgetting (displayed by task diversity groups 2 and 3 above) is an interesting contrast to the grokking literature and in particular to Nanda et al. (2023), where they find that the model first memorizes and then generalizes (which, on the surface, is the opposite of what we observe). We can explain forgetting from the perspective of simplicity bias. Since $\text{PT}_{\text{dist.}}$ is discrete and perhaps contains lots of unnecessary details, the model instead finds it easier to generalize to the 'simpler' Gaussian distribution which is continuous and much more nicely behaved. Hence, **we speculate that the simplicity of the $\text{PT}_{\text{dist.}}$ is inversely proportional to the number of tasks it contains**. Very small task diversities (group 1) are exceptions to this rule since their $\text{PT}_{\text{dist.}}$ is arguably much simpler than $\text{FG}_{\text{dist.}}$. So, we do not see forgetting in those cases as the model prefers to only learn $\text{PT}_{\text{dist.}}$. Thus, we hypothesize that the simplicities of the distributions have the following order ($G_i$ denotes group $i$): $\text{PT}_{\text{dist.}}(G_2) \approx \text{PT}_{\text{dist.}}(G_3) < \text{PT}_{\text{dist.}}(G_4) < \text{FG}_{\text{dist.}} \ll \text{PT}_{\text{dist.}}(G_1)$.

**Robustness and effect of the number of dimensions.** The phenomenon of forgetting is robust to model sizes (Figure 39), to changes in learning rate and its schedule (Figure 40), and position encodings (Monomials and Fourier Series multi-task setups use a 12-layer transformer that does not have position encodings). We also experimented with NLR problems having dimensions $d = 3$ and $d = 16$ (Figure 41) and found that the extent of forgetting (denoted by the disagreement of TF's and Ridge's loss on OOD evaluation) is directly proportional to the input dimension ($d$). Note that following Raventós et al. (2023) we keep the signal-to-noise ratio ($d/\sigma^2$) constant across these experiments by adjusting the noise scale to ensure that noise has a proportional effect and the observations are due to change in dimension alone.

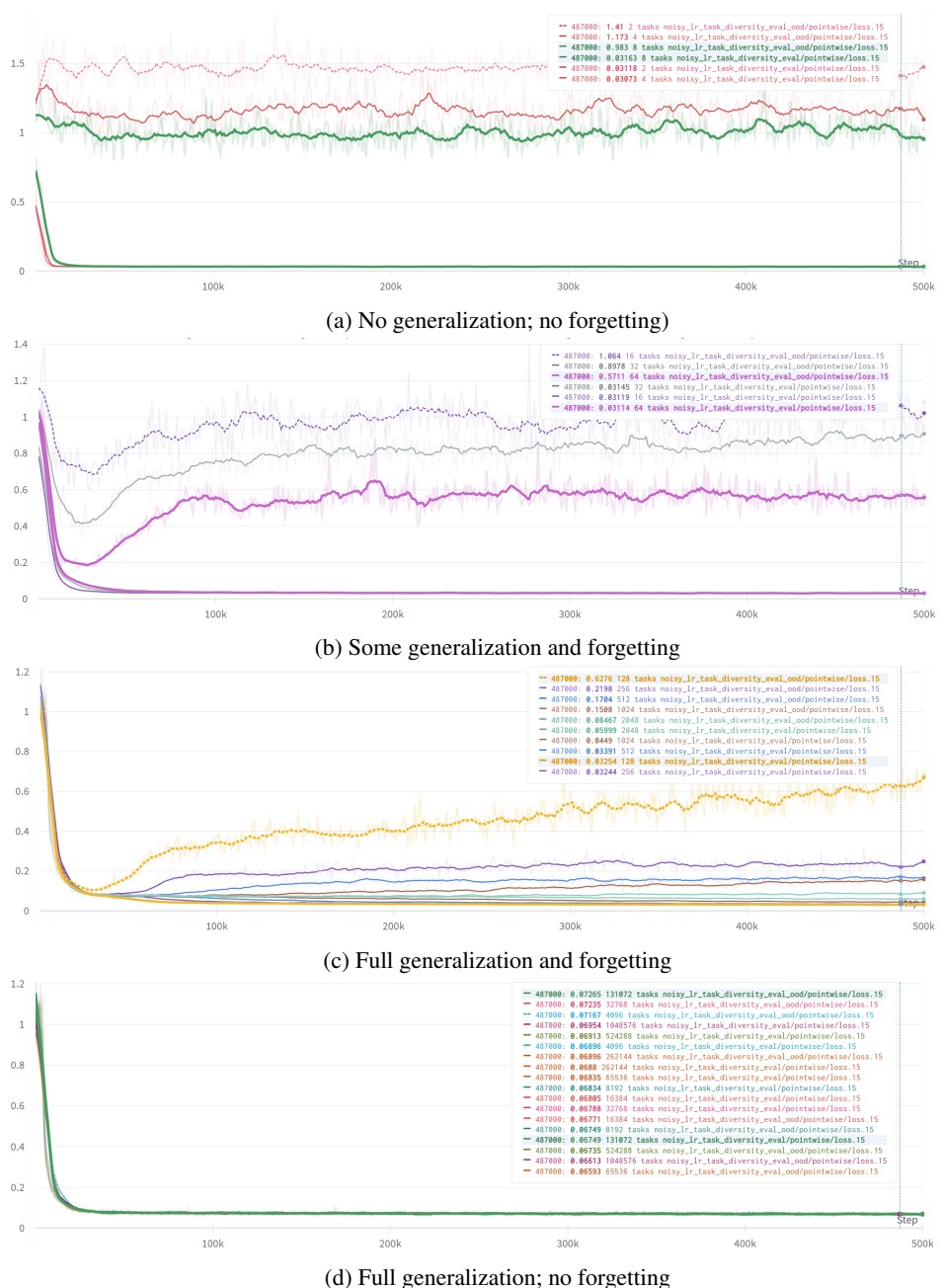

(a) No generalization; no forgetting)

(b) Some generalization and forgetting

(c) Full generalization and forgetting

(d) Full generalization; no forgetting

Figure 37: **Evolution of ID and OOD losses during pretraining for different task diversity groups for the Noisy Linear Regression problem. The forgetting phenomenon is depicted by groups in Figures (b) and (c)**. The moving average (over 10 train steps) of **ID (*eval)** and **OOD (*eval_ood)** losses are plotted, with the original (non-averaged) curves shown in a lighter shade. A checkpoint towards the end of the training is highlighted. We see that as we increase task diversity (i.e. go from group (a) towards (d)), the difference between ID and OOD losses decreases. Groups (b) and (c) are noteworthy as they display the phenomenon of forgetting, where the models' OOD loss at an earlier checkpoint is the same as ID loss, but it increases later.

Table 5: OOD loss curves of TF and Bayesian predictors for various checkpoints of models corresponding to task diversities ($K$) $2^3$, $2^5$, $2^8$, $2^{16}$ respectively in rows. Each plot presents the loss across different prompt lengths. For task diversities $2^5$ and $2^8$, plots in the first column represent the point of minima ($t_{min}$). For task diversities $2^3$ and $2^{16}$, plots in the first column represent an earlier checkpoint.

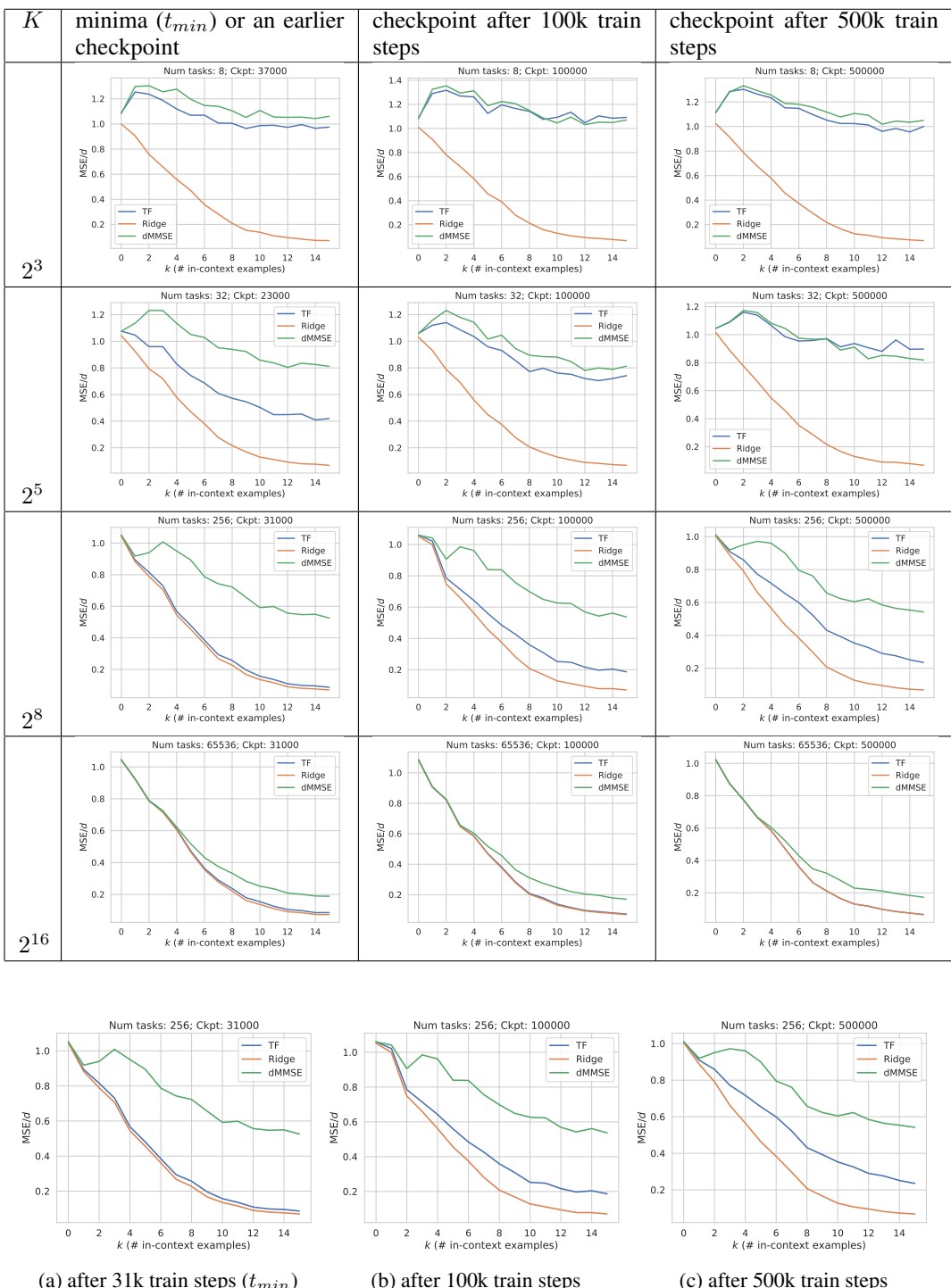

Figure 38: Plotting the OOD loss for various checkpoints during training for task diversity $2^8$, along with the Bayesian predictors. At $t_{min}$, the model agrees with Ridge regression for all prompt lengths but later deviates and converges to somewhere in the middle of two Bayesian predictors.

Table 6: Mean squared errors of implied weights of TF with Bayesian predictors during OOD evaluation for various checkpoints of models corresponding to task diversities $(K)$ $2^3, 2^5, 2^8, 2^{16}$ respectively in rows. Each plot presents the implied difference of weights across different prompt lengths. For task diversities $2^5$ and $2^8$, plots in the first column represent the point of minima $(t_{min})$. For task diversities $2^3$ and $2^{16}$, plots in the first column represent an earlier checkpoint.

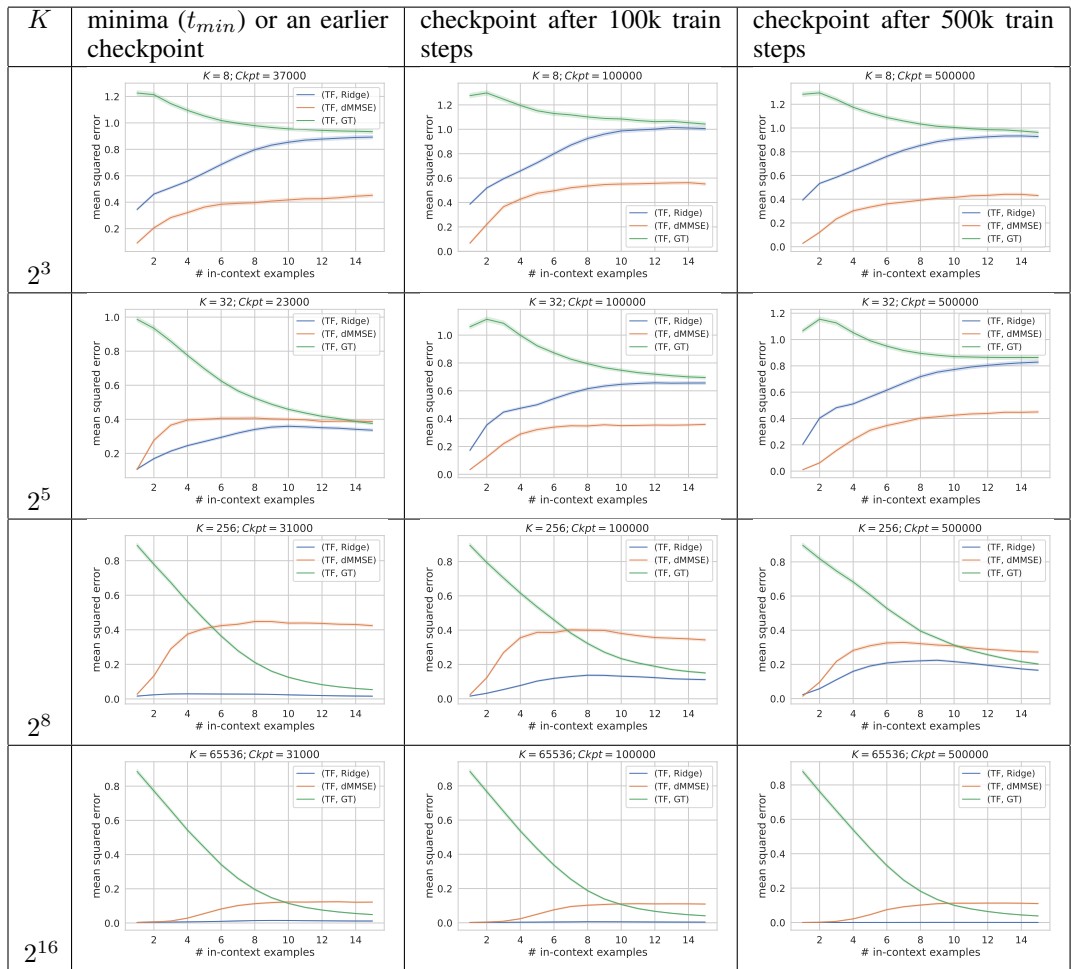

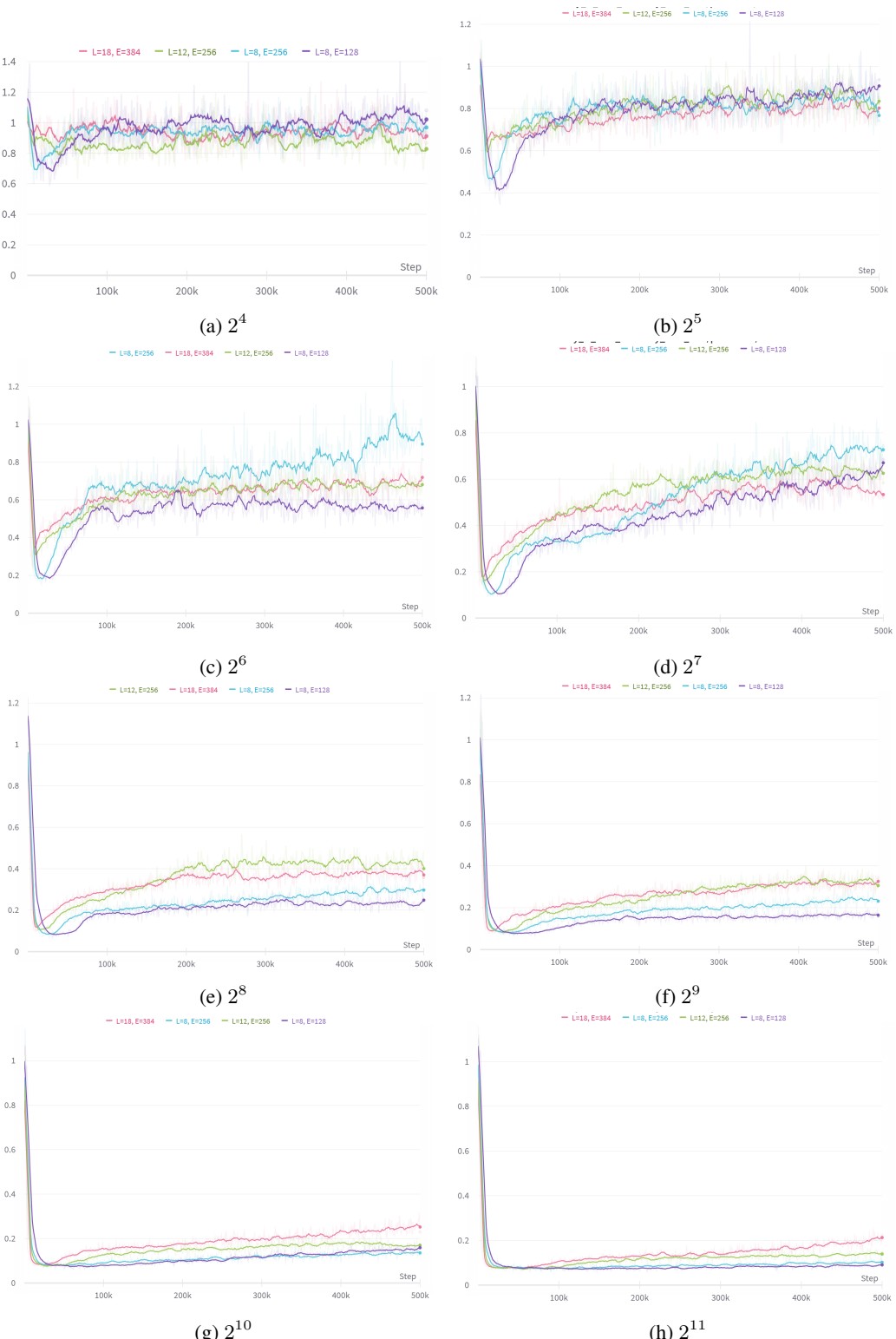

Figure 39: **Effect of model size on forgetting.** The moving average (over 10 train steps) of OOD losses for various models of different sizes trained on pretraining data with task diversities $2^4$ to $2^{11}$ are plotted. In the legend, $L$ and $E$ denote the number of layers and embedding size of the trained models respectively. As can be observed, all the models show forgetting across task diversities. Additionally, for larger task diversities (rows 3, 4), viz. $2^8$ to $2^{11}$, bigger models exhibit higher OOD loss over the course of training, i.e. show more forgetting! For smaller task diversities (rows 1, 2), viz. $2^4$ to $2^7$, there is no clear trend in the extent of forgetting across model sizes.

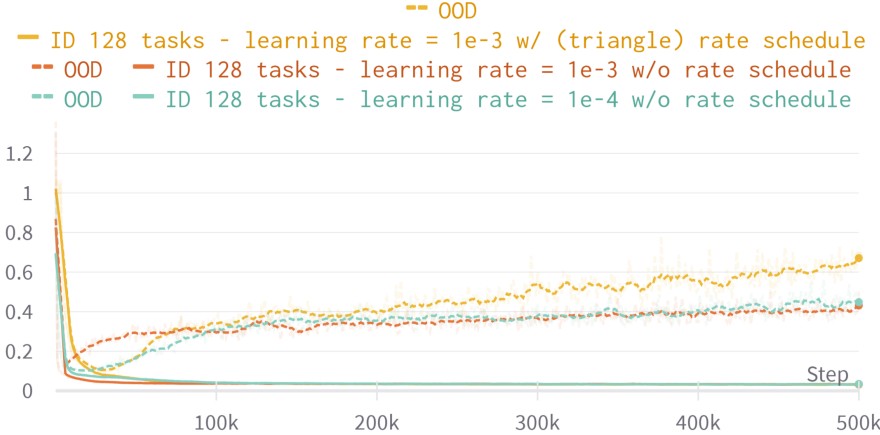

Figure 40: The moving average (over 10 train steps) of ID (solid lines) and OOD (dashed lines) losses for task diversity $2^7$ for different learning rates, with & without learning rate schedule are plotted. While the nature and extent of forgetting changes, the phenomenon itself is robust and is observed across all settings.

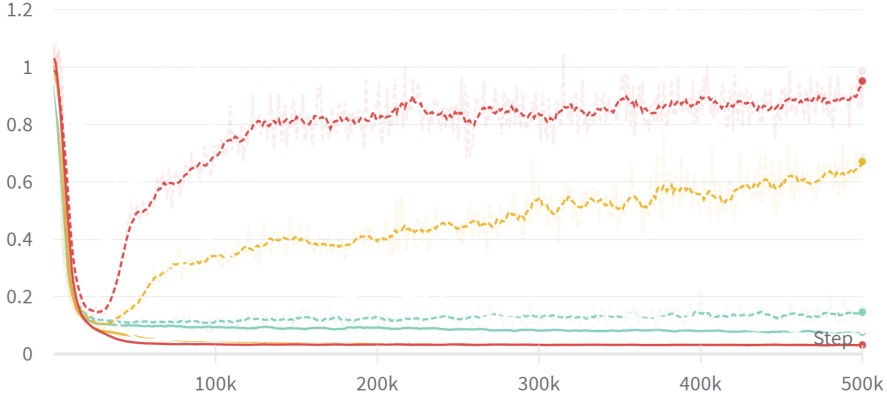

Figure 41: The moving average (over 10 train steps) of ID (solid lines) and OOD (dashed lines) losses for task diversity $2^7$ for different input dimensions (3 (green), 8 (yellow), 16 (red)) are plotted. The extent of forgetting is directly proportional to the input dimension ($d$).

## E   GRADIENT DESCENT AS A TRACTABLE APPROXIMATION OF BAYESIAN INFERENCE

We describe two recent results showing apparent deviation from Bayesian prediction: Garg et al. (2022) showed that training transformers on functions sampled from the class of 2-layer neural networks, resulted in transformers performing very close to GD during ICL. For linear regression, Akyürek et al. (2022) and von Oswald et al. (2022) showed that low-capacity transformers (i.e., those with one or few layers) agree with GD on the squared error objective. Why does GD arise in ICL when Bayesian prediction is the optimal thing to do? Towards answering this, we propose the following hypothesis: *The reason transformers appear to do GD is because GD may be providing the best approximation to Bayesian inference within the capacity constraints.* One line of evidence for this comes from Mingard et al. (2021) (and the references therein) who showed that for a range of deep neural network architectures GD solutions correlate remarkably well with the Bayesian predictor. Further, it's well known that for convex problems, GD provably reaches the global optima, hence approaching the Bayes-optimal predictor with increasing number of steps. von Oswald et al. (2022) shows that multiple layers of transformers can simulate multiple steps of gradient descent for linear regression and Akyürek et al. (2022) highlights that as the capacity of transformers is increased they converge towards Bayesian inference.

## F   FURTHER CONCLUDING REMARKS

Much more remains to be done to determine how extensively transformers mimic the Bayesian predictor. Relation between the pretraining distribution and ICL inductive bias and its relation to real-world LLMs needs to be further fleshed out. The intriguing forgetting phenomenon needs to be better understood. How is it related to pretraining simplicity bias? Further progress on the relation between gradient-based optimization and Bayesian inference would be insightful. The case of decision trees studied in Garg et al. (2022) is an interesting specific problem where the relationship between Bayesian inference and gradient descent remains unclear. While we studied out-of-distribution performance on new function families, out-of-distribution performance on new input distributions is also of interest.

The present work focused on continuous functions setting which is easier to study from the Bayesian perspective. What happens for the real-world LLMs? How strongly does the Bayesian view hold there and what kind of deviations exist? The order of demonstrations is known to have significant influence on the output of the LLMs. Thus more nuance would be necessary for LLMs. Assuming that the Bayesian view does hold for LLMs in some form a potential practical implication is that it can help us choose demonstrations in an informed manner to make the conditional distribution converge faster to the intended output. Moreover, it can also potentially help us understand real-world LLM phenomena like hallucination and jailbreaking assuming the nature of the implied posterior distribution characterizes them. Finally, we treated transformers as black boxes: opening the box and uncovering the underlying mechanisms transformers use to do Bayesian prediction would be very interesting.

