# OpenReview forum: "In-Context Learning through the Bayesian Prism"
_ICLR.cc/2024/Conference — ICLR 2024 poster_

### Official Review · Reviewer_Rt1y · 2023-10-18

**Soundness:** 3 good
**Presentation:** 2 fair
**Contribution:** 3 good
**Rating:** 8
**Confidence:** 4

**Summary:**

This paper empirically showed the relationship between trained transformers and Bayesian optimal estimators. They showed that in Gaussian Mixture Models(GMM) with equal rates and two components, the prediction of trained transformers follows the PME over the two components, which is kind of different to the PME over prior distribution being the distribution of either. They also showed that the prediction follows the PME of a strong baseline (when the PME is intractible) for linear regression problems like dense linear regression, sparse linear regression, signed vector linear regression or low-rank linear regression.

Then they investigated the simplicity bias of ICL of transformers. They showed with a single task distribution, the trained TF (transformers) do not give any bias on the frequency of training data, but with multiple sources of tasks  distribution, they will tend to give more weight to the low-frequency --- which is natural, since the low-frequency data appears in more tasks than high-frequency ones. They then show the in-distribution and OOD generalization capacity of IC-trained TFs on linear regression on some polynomial features, and showed that they mimics the OLS on the subset of features that appears in the test prompts.

They also showed some interesting phenomenon which are novel and  worth investigating in the future. For example, they showed the deviation from the IC-trained TFs and the Bayesian optimal estimators, which well supplements the experiments of Garg;s paper and Akyureck's paper who both showed that TFs mimic the PME. Additionally, they showed a very interesting 'forgetting' phenomenon.

 I will vote for acceptance for this paper. Based on the authors' reply, I raised my score to eight.

**Strengths:**

1. They have sufficient experiments and the results show some very interesting phenomenon. In some linear and non-linear tasks, the TFs are competitive with the PME or some strong baselines, which showed that TFs are very efficient to learn some algorithms on different linear/non-linear regression tasks.

2. I totally agree with the implication the authors proposed in page 5 that TFs do not need to first recognize the underlying tasks and then solve it  in order to approximate the Bayesian optimal solutions. The authors frankly show that in some cases, the prediction of TFs mimics the PME while in some case the prediction can deviate from that.

3. For  the simplicity bias and the generalization parts, the authors gave many baseline for comparasion and showed that TFs could potentially do something very non-trivial. The experimental evidence is sufficient to convince me that the generalization capacity for OOD tasks increases when the task diversity improves.

**Weaknesses:**

See 'Questions'.

**Questions:**

Here, I have some questions, and as well as some personal suggestions. I will appreciate it if you can address my confusion.

1. You have proposed a setting called HMICL, which you claim is different from the MICL setting proposed in [1]. I am wondering what is the difference between two settings (more formally, from the definitions in [2])? In HMICL, the task is sampled from a hierarchical distribution, which can still be viewed as 'a specific distribution', right? So to my understanding, you did not propose a 'new setting' but instead, you are considering the exact MICL setting where the task distribution is hierarchical, right? I am also wondering why you think this hierarchical structure of task distributions matters in reality? Is there any evidence showing that the linguistic data or some other real-world data follows this hierarchical distribution?

2. In PME paragraph in page 3, you claim that 'the predictions of the model can be computed using the posterior mean estimator (PME) from Bayesian statistics.' I think this is not that rigorous, since this happens only when the function class you are considering (here, it's the TF class) are expressive enough so that the PME is included in this function class, right? (Namely, this only happens in the realizable case). When the model class is not large enough, the TFs can not express the PME and hence, can only 'approximate' it.

3. In GMM experiments, it's good to see that TFs follows the PME over the prior of two mixed components. Two questions I have: (1): In the figure, you only show k <= 10, which is a under-determined case for a linear model, since the dimension you use is d=10. In the over-determined case, what is the PME (is it OLS) and how does the TFs behave?  (2). Can you also plot the curve for OLS for all context length? I am wondering what the difference between a TFs and OLS here.

4. For the experiments on multiple linear models (DR, SR, SVR, Skew-DR,etc), even though the PME is not analytically tractible, it is still possible to numerically compute the PME. Have you tried numerical computation for these PME and compare the TF to them? I think this will be a strong evidence to say whether the TFs are really mimicking the Bayesian optimal estimators. Although this could be hard in high-dimension case (I am not an expert in Bayesian but I guess so), it should be doable in a d=5 or d=10 case?

Another question is, how do you determine the regularization coefficient of LASSO? Does this require more data (if so, that mean you are actually using a longer context to do LASSO than to do ICL using TFs.)

5. For the experiments on Frourier series and second-degree polynomials, an important question is that, how did you input the data? Did you simply encode original x_i s into token matrices, or you use \phi(x_i) (where \phi is a Frourier basis or a second-degree polynomial) to serve as the input?

6. I kind of feel that there are more recent works about the relationship between GD, ICL, OLS and distribution shift [2,3,4]. I am wondering whether their results can somehow explain some of your experiments or to some extent are related to your results?

[1]. What Can Transformers Learn In-Context? A Case Study of Simple Function Classes?
[2]. Trained Transformers Learn Linear Models In-Context.
[3]. A Closer Look at In-Context Learning under Distribution Shifts
[4]. Transformers learn to implement preconditioned gradient descent for in-context learning

---

> ### Author Response · Authors · 2023-11-17
>
> We thank the reviewer for carefully going through our work and providing helpful feedback. Below we try to answer their questions and concerns:
>
> **Q1. HMICL and MICL**: While the reviewer is correct in a formal sense, we believe it’s clarifying to use our naming. Roughly speaking, the distinction between MICL and HMICL is reminiscent of the distinction between the usual supervised learning and meta-learning. As an example, in the MICL setup, consider linear regression. Here all tasks are instances of linear regression and each weight vector $w$ defines a task. Now if we consider a mixture of two “meta-tasks” (what we call function classes in the paper), say linear regression and decision trees, then it is true that in an abstract sense, one could potentially consider each task to be defined by its parameters and thus ignore which type of “meta-task” it instantiates (linear regression or decision tree). The formal definition in [2] which allows general hypothesis classes can similarly model HMICL.
>
> Thus, we agree with the reviewer that on this interpretation, MICL is equivalent to HMICL. However, this view is too coarse-grained for our purposes. What is of interest, both from the application and theory perspectives, is a more fine-grained view about whether and how models learn to perform these different “meta-tasks”. And the hierarchical structure is central for even talking about this. Previous work, e.g.  [1, 2], considers MICL with only a single “meta-task” and is thus not suitable for the type of analyses we perform. Compared to MICL, HMICL is arguably closer to the ICL for LLMs which can perform a vast variety of “meta-tasks.” Due to the complex nature of real world training data distributions, it is hard to find concrete evidence of them being hierarchical, but we believe multi-task training in LMs like T5 [5] and FLAN-T5 [6] (with a caveat for the latter being true for fine-tuning and not pre-training) or training of multilingual models [7] which involve pre-training corpora in different languages are some examples of the hierarchical nature of the training distribution in real-world LMs.
>
> We welcome further feedback and any suggestions for better naming.
>
> **Q2. PME**:  We apologize for confusion here. In the line ‘the predictions of the model can be computed using the posterior mean estimator (PME)’, we mean the ideal model (unlimited data and infinite compute) that we mention in the previous sentence and not transformers specifically. We fully agree that transformers can only approximate PME and how well they approximate it will depend on the scale of the model. We will correct this line in the final version to avoid this confusion.
>
> **Q3. GMM plots with OLS for over-determined region**: We thank the reviewer for suggesting these experiments. We provide the answers to the two questions below:
>
> (1) Figure 21 in Appendix plots the errors for the transformer model from Figure 1, PMEs, and OLS for the over-determined region. In the over-determined case $(d>10)$, the solution is unique, hence all the predictors including transformer give the same solution and have errors $\approx 0$.
>
> (2) As shown in Figure 21, transformer’s errors are smaller than OLS errors and agree with the PME(GMM) errors for all context lengths. This shows that Transformer is indeed simulating the mixture PME and not OLS.
>
> **We address the remaining concerns in the next comment.**
>
>
> [1] Garg et al. 2022.  What Can Transformers Learn In-Context? A Case Study of Simple Function Classes?
>
> [2] Zhang et al. 2023. Trained Transformers Learn Linear Models In-Context.
>
> [5] Raffel et al. 2019. Exploring the Limits of Transfer Learning with a Unified Text-to-Text Transformer
>
> [6] Chung et al. 2022. Scaling Instruction-Finetuned Language Models
>
> [7] Conneau et al., ACL 2020. Unsupervised Cross-lingual Representation Learning at Scale

---

> ### Author Response · Authors · 2023-11-17
>
> Continuing from the last comment, we address the remaining concerns of the reviewer here.
>
> **Q4.1. Sampling based approximations of PME**: Thank you for asking this question. We should first clarify that for Dense Regression and Skewed-Covariance Regression, we can compute Bayesian predictors exactly and we are able to show transformers for these two problems perform very close to the Bayesian predictor (Figures 6 and 9 in Appendix). However, for other linear inverse problems where PME seems intractable, based on the reviewer’s suggestion, we have tried Markov Chain Monte Carlo sampling method (No U-Turn Sampler (NUTS)) to approximate the PME. Particularly for Low-Rank and Sign-Vector Regression problems, for small problem dimensions i.e. 16 and 8 respectively, we were able to approximate the PME. We provide the plots in Figure 11 in Appendix and TF can be seen to follow NUTS-based PME in both cases. In fact, transformers seem to perform slightly better than the NUTS solution, which we believe could be due to the errors in the approximations due to sampling. We had less success in approximating the PME for sparse regression, which we believe might be due to the complex nature of the prior which involves a mixture of discrete (to select which features to keep) and continuous distributions (for the values of selected features). However, the alignment between transformers and exact PMEs for Dense and Skewed-covariance regression as well as numerically approximated PMEs for Low-Rank and Sign-Vector regression do provide strong evidence of transformers behaving like Bayesian predictor during ICL for linear inverse problems.
>
> **Q4.2. Lasso Coefficient**: We tune the LASSO coefficient following Garg et al. [1] i.e. by using a separate batch of data (1280 sequences) and choose the single value that achieves the smallest loss. This chosen parameter is then used during inference and no separate tuning is done for the in-context examples. Hence, all predictors (including LASSO) get the same number of examples as the context length in ICL using TFs.
>
> **Q5. Feature Encoding for non-linear regression tasks**: Thanks for this question. We just provide $x_i$ as the inputs, so the model needs to figure out $\Phi$ on its own.
>
> **Q6. Relation with previous work**: We thank the reviewer for asking this question. In short, these papers study a related but distinct topic and hence do not have any bearing on our results as detailed next. Both [2] and [3] study ICL on linear regression and consider different distribution shifts to measure how robust or brittle transformers are towards different types of shifts. While we also consider OOD generalization for degree-2 monomial regression and noisy linear regression, we are analyzing something distinct: we study generalization to unseen function classes or tasks and relate it to deviations of transformers during ICL from the Bayesian predictor. Our findings show that transformers are actually capable of better generalization than the Bayesian predictor (considering the pre-training distribution). Similarly, [2] studies the training dynamic of single-layer attention-only transformers on the linear regression problem to prove that they can learn to solve the problem in-context under careful initializations. Models here do not solve the problem exactly but rather provide the best approximation possible by a single-layer model characterized by a single step of gradient descent. [4] has closely related results. The emphasis of [2, 4] is on mathematical proof and this necessitates an extremely simple setup distinct from ours. On the other hand, we show empirically that for a wide variety of linear and nonlinear function classes as well as their mixtures, the behavior of sufficiently large transformers during ICL can be explained using the Bayesian perspective. Further, we use this to study the inductive biases of transformers during ICL (the "Simplicity Bias" section) as well as the cases where the behavior of transformers deviates from that of the Bayesian predictor. We hope this helps the reviewer put our work in perspective compared to the mentioned papers.
>
> [1]. Garg et al. 2022.  What Can Transformers Learn In-Context? A Case Study of Simple Function Classes?
>
> [2]. Zhang et al. 2023. Trained Transformers Learn Linear Models In-Context.
>
> [3]. Ahuja et al. 2023. A Closer Look at In-Context Learning under Distribution Shifts
>
> [4]. Ahn et al. 2023. Transformers learn to implement preconditioned gradient descent for in-context learning

---

> > ### Comment · Reviewer_Rt1y · 2023-11-17
> > **Great additional experiments**
> >
> > Thanks the authors for their endeavor to do these experiments in a short rebuttal time.
> >
> > I think the authors' clarification on the difference between MICL and HMICL is convincing. Most importantly, the additional experiment on approximating PME using MCMC method is great. This provides evidenced that TFs are actually approximating the PME under different context length, which makes the claim stronger.
> >
> > Based on the authors' rebuttal, I raised the score to eight.

---

> > > ### Author Response · Authors · 2023-11-19
> > >
> > > Thank you for your appreciation and for suggesting additional experiments that helped strengthen our claims.

---

### Official Review · Reviewer_C7Ly · 2023-10-30

**Soundness:** 2 fair
**Presentation:** 3 good
**Contribution:** 3 good
**Rating:** 6
**Confidence:** 4

**Summary:**

The paper focuses on the phenomenon of in-context learning (ICL) in large language models (LLMs) and aims to understand the underlying inductive biases that enable successful generalization to new functions. The authors propose a hierarchical meta-ICL setup that encompasses multiple task families and investigate the empirical performance of LLMs in this expanded setting. They find evidence that high-capacity transformers mimic the behavior of Bayesian predictors in cases where Bayesian inference is tractable. This Bayesian perspective sheds light on the inductive biases of ICL and how transformers perform tasks when trained on multiple tasks. Additionally, the paper explores situations where transformers deviate from the Bayesian predictor, leading to new insights and hypotheses.

**Strengths:**

1. The paper addresses a significant problem by conducting extensive experiments that aim to provide an explanation for the in-context learning capabilities of large language models (LLMs) from a Bayesian inference perspective. This research direction holds promise for advancing our understanding of LLMs.
2. The paper is well-organized and effectively communicates its ideas. The authors present a natural and logically motivated experimental setting, which not only facilitates comprehension but also encourages further exploration and experimentation in the field.

**Weaknesses:**

1. The settings examined in this study predominantly rely on synthetic data, which creates a notable disparity between the experiments and real-world data. This limits the generalizability and applicability of the findings to realistic scenarios.
2. Although the experiments are comprehensive and thought-provoking, there are certain definitions that would benefit from further clarification. Specifically, many explanations in the current version are contingent upon the capacity constraints of transformers. To enhance the illustration of the impact of transformer capacity, it is suggested that the authors include plots depicting the size of transformers in relation to the observed deviation phenomenon. This would provide a clearer understanding of the relationship between capacity and performance.
3. As an empirical investigation paper, I would suggest that the authors consider providing the code to facilitate further investigation and replication of their findings.

**Questions:**

It is well-known that the order of demonstrations has a significant impact on the final results of in-context learning. However, it appears that the current experimental settings do not account for this influential factor. Please correct me if I have misunderstood any aspects. Moreover, I am curious if increasing the model size of transformers can alleviate the forgetting phenomenon.

---

> ### Author Response · Authors · 2023-11-17
>
> We thank the reviewer for going through our work and providing insightful comments. Below we try to address their concerns:
>
> **W1. About synthetic setting**: We agree with the reviewer that our work is mostly concerned with synthetic datasets to study how in-context learning works in transformers. The reason for choosing the stylised setup for our investigation stems from the fact that the real-world datasets used to train LLMs are difficult to model theoretically. Our work builds upon a long line of prior works, e.g., [1, 2, 3], that make simplifying assumptions on the pre-training data distribution to theoretically understand in-context learning in transformers. Our hierarchical meta ICL setup is also arguably closer to in-context learning in practical LLMs which can realize different classes of tasks (sentiment analysis, QA, summarization etc.) depending upon the inputs provided. Our work provides evidence on a diverse set of problems that the behavior of transformers during ICL agree with the Bayesian predictor. How well this holds for LLMs in practice is an open question, but we believe our work takes a step towards answering this question.
>
> **W2. Effect of Model Size on Deviations**: As per reviewer’s suggestion, we conducted experiments with transformers of varying sizes. For the forgetting phenomenon, bigger models for larger task diversities have larger OOD losses as the training progresses ⇒ more forgetting. For smaller task diversities, there is no clear trend, however the forgetting phenomenon continues to hold. Hence, the deviations are not alleviated by increasing model size. (Figure 40, Appendix).
>
> **W3. Code Availability**: We will be releasing our code publicly upon the publication of our work. For now, we provide the anonymous link to the code repository for reviewer's reference: https://anonymous.4open.science/r/icl-bayesian-prism/README.md.
>
> **Q1. Impact of order of examples**: In our initial experiments, we did evaluate the influence of order of examples on the ICL performance. In particular, for dense linear regression, we found that the ICL performance is fairly robust to the order of examples. We have added Figure 10 in Appendix which shows a box plot for loss@k values at different prompt lengths, and for each prompt length we consider 20 permutations. As can be seen the standard deviation of the errors is close to zero across prompt lengths. Hence, the order of the in-context examples do not impact the ICL performance here. We did expect such behavior to hold in our setup, since we deal with continuous functions and things like majority label bias and recency bias [4], which are a source of variance in performance due to the order of examples for NLP tasks, do not apply here.
>
> **Q2. Effect of Model Size on Forgetting**: As mentioned in response to W2, increasing the model size does not alleviate forgetting. Instead, it leads to more forgetting for larger task diversities, as the model fits the pretraining distribution better over the course of training and hence has higher OOD losses. (Figure 40, Appendix).
>
> [1] Xie et al. 2021. An Explanation of In-context Learning as Implicit Bayesian Inference
>
> [2] Garg et al. NeurIPS 2022. What Can Transformers Learn In-Context? A Case Study of Simple Function Classes
>
> [3]  Akyürek et al. ICLR 2023. What learning algorithm is in-context learning? Investigations with linear models
>
> [4] Zhao et al. ICML 2021. Calibrate Before Use: Improving Few-Shot Performance of Language Models

---

> ### Comment · Reviewer_C7Ly · 2023-11-18
>
> Thanks for the response. The paper presents an intriguing setup and experimental phenomenon to explore In-Context Learning, despite the synthetic nature of the setup. I maintain my current positive evaluation and attitude towards this paper, while increasing my confidence.

---

> > ### Author Response · Authors · 2023-11-19
> >
> > Thank you for your response. We would appreciate further detailed feedback and would be happy to answer any further questions.

---

### Official Review · Reviewer_P1i1 · 2023-10-31

**Soundness:** 2 fair
**Presentation:** 2 fair
**Contribution:** 2 fair
**Rating:** 6
**Confidence:** 2

**Summary:**

This paper considers in-context learning for large language models. This paper extends the previous work "What can transformers learn in-context? A case study of simple function classes." by considering multiple function families, out-of-distribution detection, and Bayesian prediction tasks. This paper conducts extensive experiments and large-scale analysis in the experiment section.

**Strengths:**

The author conducts extensive experiments on prompting engineering for large language models over multiple tasks.

**Weaknesses:**

- The paper is not well organized which makes it very hard to read through.
    - The Figures are very hard to understand, making it challenging to summarize the conclusion from the figures.
    - It's not a good idea to keep stating "refer to Appendix" for the main methodology part of the paper.
    - It is also not a good idea to keep stating that some steps in the proposed method are similar to a specific paper and cite that paper. At least, the authors need to give a clear reason for doing so.
- The paper mentioned a lot of technique terms without giving any rational and proper definitions. For example,
    - the "Bayesian predictor" is not properly explained in the context of "in-context learning".
    - I'm not familiar with The "deviations from Bayesian prediction" . The author should explain that clearly.
    - The "Simplicity bias" should be better replaced with "Occam’s razor" or "no free lunch theorem". Because the suggested terms are used very often in the literature.
    - If you mean "generalization is out of distribution (OOD) detection". Then just using OOD detection would reduce a lot of confusion.
- The correctness of Equation 2 is questionable. What is the exact definition of "$df$", the differentiation with respect to a function $f$?
- As the main component of this work, the paper lacks the reason for extending ICL to hierarchical ICL. Why do you want to sample from a mixture of the function family?

**Questions:**

- Considering the paper's main topic is about prompting engineering and large language models, the author might consider trying for a compatible conference in NLP.

---

> ### Author Response · Authors · 2023-11-17
>
> We thank the reviewer for going through our work and providing useful feedback. Below we try to answer their concerns and questions:
>
> 1. **Readability**: We will take the reviewer’s comments into account and will add more descriptive captions so that the figures are easier to understand. About references to the appendix, we believe that given the contents and the scope of our work, it would have been difficult to fit all the technical details in the main paper given the page limit. However, the details in the appendix are not primary for understanding the core theme and contributions of our work. However, we will take this feedback into account and move some of the details to the main paper. About references to “some specific” paper, we are not sure about which paper the reviewer is referring to here. If it is Akyurek et al. 2022 [1], we cite their work while describing the weight recovery method in page 3, since we use their method for our experiments, but we do provide all the details about how the method works after citing their work. In case they are referring to Raventos et al. (2023) [2], while we do provide all the important details from their work relevant to our experimental setup and findings, we will add some more details from their work in Section 6 to improve readability.
>
> 2. **Technical definitions**:
> - **Bayesian Predictor in context of ICL** : Informally, we have a probability distribution on tuples $(x, f(x))$ induced by the priors on $x$ and $f$. This in turn, induces a distribution on the prompts, namely $(x_1, f(x_1)), (x_2, f(x_2)), …, (x_i, f(x_i)), x_{i+1}$. Conditioned on the prompt we would like to sample $f(x_{i+1})$---this is the posterior predictive distribution (PPD). We use the mean of the PPD as the Bayesian predictor since the nature of our setup requires a point estimate. The formula for PME in the paper formalizes this. We provide the formal definition of Bayesian predictor in context of ICL in Section 2 (under the PME paragraph) and discuss the relationship between ICL and Bayesian inference in the Bayesian predictor paragraph in the introduction Section.
>
> - **Deviations from Bayesian prediction**: “Deviations” here refer to predictions of transformers during ICL that do not match with the Bayesian predictor given the pre-training distribution and the observed data.
>
> - **Simplicity Bias**: Simplicity bias is by now a well-established term ML literature; for example, [3, 4, 5, 6].
>
> - **Generalization**: We agree that a more descriptive term should be used here. We will change it to "OOD Generalization" in the final version for clarity.
>
> 3. **Differential of a function $df$**:  This is standard notation in Bayesian inference. We can view a function as a point in a higher-dimensional Euclidean space defined by the function parameters. E.g, for linear regression, this reduces to the weight vector $w$ and hence $df$ is simply $dw$.
>
> 4. **Why HMICL?**: Prior work on theoretically understanding ICL, considers the simpler case where functions are always sampled from a single function class (which we term as MICL). We present a more advanced setup where the functions can come from different classes. Further, our setup can serve as a testbed for investigating the origins of multi-task learning in large language models (LLMs) because HMICL is arguably closer to ICL in real-world LLMs, which can realize different classes of tasks (sentiment analysis, QA, summarization etc) depending upon the inputs provided. We show that the ability to solve a mixture of tasks arises naturally as a consequence of Bayesian prediction, which can be viewed as computing the posterior when the prior comes from a mixture of function classes. This challenges the existing intuition of multi-task learning where the model first identifies the task and then solves it.
>
>
>
> [1] Akyürek et al. ICLR 2023. What learning algorithm is in-context learning? Investigations with linear models
>
> [2] Raventós et al. 2023. Pretraining task diversity and the emergence of non-Bayesian in-context learning for regression
>
> [3] Valle-Pérez et al. ICLR 2019. Deep learning generalizes because the parameter-function map is biased towards simple functions.
>
> [4] Shah et al. NeurIPS 2020. The Pitfalls of Simplicity Bias in Neural Networks.
>
> [5] Mingard et al. 2019. Neural networks are a priori biased towards Boolean functions with low entropy
>
> [6] Nakkiran et al., NeurIPS 2019. SGD on Neural Networks Learns Functions of Increasing Complexity.

---

> > ### Author Response · Authors · 2023-11-23
> >
> > Please let us know if there are any unanswered questions. We are happy to further clarify and appreciate any feedback. Thanks.

---

### Meta-Review · Area_Chair_pYQZ · 2023-12-14

**Metareview:**

The paper studies an extension of the in-context learning setting with linear functions - by considering a mixture of tasks. The paper relates the performance of transformers to that of the Bayes optimal predictor across various settings. The reviewers are all generally positive but bring out good points of improvement in the reviews. Overall, I recommend the authors to revise the pitch of the paper to make more clear what the crisp contributions/surprising observations and insights are. While the paper does not add a dramatically new perspective (or any immediately useful practical insights), it seems to add some valuable contributions into further understanding of in-context learning.

**Justification For Why Not Higher Score:**

There is no crisp new message here, and the new experiments/insights seem incremental.

**Justification For Why Not Lower Score:**

There are a few tidbits of interesting observations that would be valuable to the community. All three reviewers enjoyed reading the paper and I didn't find any obvious flaw or misstatement in the submission

---

### Decision · Program_Chairs · 2024-01-16

Accept (poster)